# MoFO: Momentum-Filtered Optimizer for Mitigating Forgetting in LLM Fine-Tuning

**Yupeng Chen**[*]                                                              *yupengchen1@link.cuhk.edu.cn*
*The Chinese University of Hong Kong, Shenzhen, China*

**Senmiao Wang**[*]                                                           *senmiaowang1@link.cuhk.edu.cn*
*The Chinese University of Hong Kong, Shenzhen, China*

**Yushun Zhang**
*The Chinese University of Hong Kong, Shenzhen, China*
*Shenzhen Research Institute of Big Data*

**Zhihang Lin**
*Shenzhen Research Institute of Big Data*

**Haozhe Zhang**
*Huawei Technologies Co.,Ltd, China*

**Weijian Sun**
*Huawei Technologies Co.,Ltd, China*

**Tian Ding**[†]                                                                      *dingtian@sribd.cn*
*Shenzhen International Center for Industrial and Applied Mathematics*
*Shenzhen Research Institute of Big Data*

**Ruoyu Sun**[†]                                                                 *sunruoyu@cuhk.edu.cn*
*The Chinese University of Hong Kong, Shenzhen, China*
*Shenzhen International Center for Industrial and Applied Mathematics*
*Shenzhen Research Institute of Big Data*

**Reviewed on OpenReview:** *https://openreview.net/forum?id=T1qXIDn9my*

## Abstract

Large language models (LLMs) have demonstrated remarkable capabilities across a wide range of tasks. Typically, LLMs are first pre-trained on large corpora and subsequently fine-tuned on task-specific datasets. However, during fine-tuning, LLMs may forget some knowledge acquired in the pre-training stage, leading to a decline in general capabilities. Existing approaches to mitigate forgetting often rely on access to pre-training data, which may be unavailable in many real-world scenarios—such as fine-tuning checkpoint-only open-source LLMs. To address this challenge, we propose a new fine-tuning algorithm termed Momentum-Filtered Optimizer (MoFO). MoFO is an extension of greedy block coordinate descent (BCD) methods: in each iteration, MoFO only updates the model parameters with the largest momentum magnitudes, while keeping all other parameters fixed. MoFO achieves similar fine-tuning performance to the default fine-tuning algorithm while effectively mitigating knowledge forgetting. We validate MoFO through rigorous convergence analysis and extensive experiments, demonstrating its effectiveness in mitigating forgetting without pre-training data.[1]

---

[*]Equal contribution.
[†]Corresponding author.
[1]Our code is available at `https://github.com/YChen-zzz/MoFO`.

# 1   Introduction

The success of large language models (LLMs) lies in their strong capabilities in language understanding and generation. Typically, LLMs are first pre-trained on extensive corpora to acquire general capabilities, then fine-tuned on smaller, task-specific datasets to adapt to particular tasks or domains (Dai & Le, 2015; Kenton & Toutanova, 2019; Radford et al., 2018). However, it has been observed that during the fine-tuning process, LLMs may forget the knowledge acquired in pre-training, leading to a decline in general capabilities (Lin et al., 2023; Chen et al., 2020; Dong et al., 2021; Korbak et al., 2022; Luo et al., 2023a). Therefore, addressing the forgetting issue in LLM fine-tuning has become an important research direction.

In the field of continual learning, mitigating forgetting has already been a central focus. Continual learning (Wang et al., 2024a) involves training models sequentially on different tasks, which is analogous to the process of pre-training followed by fine-tuning in LLMs: Both involve different stages of training and face the challenge of forgetting previously acquired knowledge when learning new information. To address this forgetting issue, *replay-based methods* (Rolnick et al., 2019; Wang et al., 2020; Ouyang et al., 2022) use a replay buffer to store and revisit past data, in order to reinforce prior knowledge while learning new information. In LLM training, similar replay-based methods are also used to mitigate forgetting (Shi et al., 2024; Roziere et al., 2023; Huang et al., 2024). However, replay-based methods face some practical limitations in LLMs. First, the access to the original pre-training data is often restricted or infeasible. Many open-source LLMs, such as the Llama series (Touvron et al., 2023), do not fully disclose their pre-training datasets. Second, even when pre-training data is available, incorporating it into the fine-tuning process can substantially increase computational and memory costs, as the model must process a much larger and more diverse dataset.

In continual learning, another class of methods focuses on modifying the optimization process of models to mitigate forgetting (Wang et al., 2024a). These methods do not require access to the pre-training data. However, many of them need to store and exploit the pre-training weights and/or intermediate gradients throughout most of the fine-tuning process. For example, some regularization-based methods such as $L_1$ or $L_2$ regularization (Panigrahi et al., 2023; Li et al., 2018) require storing the pre-training weights for regularization computation during the whole fine-tuning process. However, in the context of LLMs, storing additional model weights requires substantial memory due to the large model size, introducing considerable overhead.

Given the limitations of prevalent forgetting-mitigation approaches, we aim to develop an optimization algorithm that neither relies on past data nor introduces additional memory overhead. **In general, our algorithm design goals are twofold: (G1) preservation of pretrained knowledge and (G2) strong performance on the fine-tuning task.** To pursue (G1), we adopt a key insight from continual learning (Kirkpatrick et al., 2017): the closer a fine-tuned model stays to the pretrained parameters, the less forgetting tends to occur. At the same time, enforcing proximity too aggressively can hinder adaptation to the new task, thereby harming (G2). This raises a natural question: *What optimization method might move a small distance from the initial point, but can still find a minimum² of the loss function?*

We notice that the classical block coordinate descent (BCD) method (Tseng, 2001) is a good candidate, since it updates only a subset of parameters at each iteration, thus implicitly biased towards closer solutions. Nevertheless, incorporating BCD into LLM fine-tuning presents some challenges. One challenge is that Adam, the predominant optimizer for LLM training (Radford et al., 2018; Zhang et al., 2024c), differs substantially from the optimizer studied in traditional BCD methods (e.g. GD or SGD) (Nutini et al., 2015; Zhao et al., 2014). It complicates both optimizer design and convergence analysis. Consequently, combining BCD with Adam is not a straightforward task.

In this work, we propose Momentum-Filtered Optimizer (MoFO), a new optimization algorithm that integrates Adam with BCD. At each iteration, MoFO only updates the most effective parameters for reducing the fine-tuning loss—those with large momentum magnitudes, while keeping other parameters fixed. MoFO only modifies the optimizer without the need for pre-training data or introducing additional memory costs, which helps achieve its efficiency and effectiveness during fine-tuning. Our contributions are summarized as follows:

---

²In this paper, we use the term "minimum" (or "minima" in the plural) to refer to a parameter configuration whose fine-tuning loss is near its lowest value in a small neighborhood, while acknowledging that this terminology may not strictly represent a local minimum of the fine-tuning loss function.

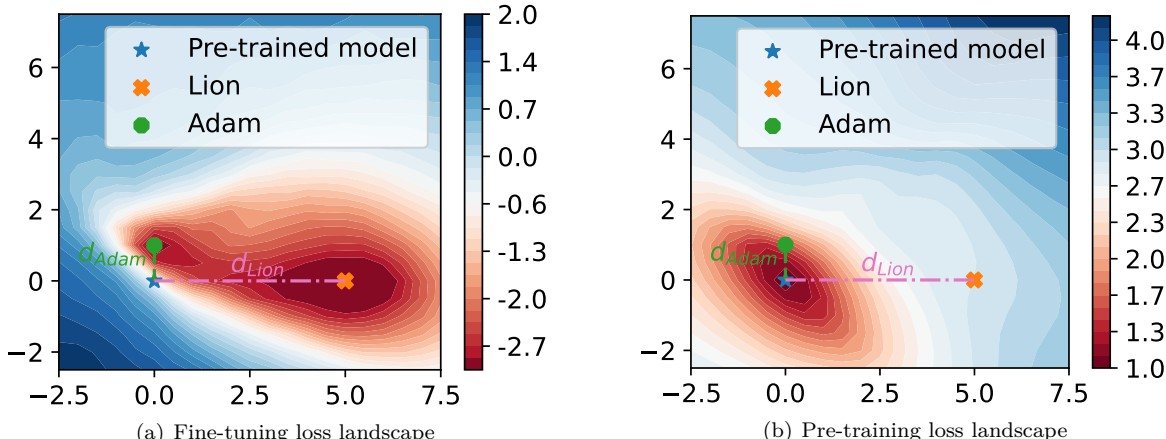

(a) Fine-tuning loss landscape        (b) Pre-training loss landscape

Figure 1: The loss landscapes of Pythia-160M after fine-tuning on a subset of the FLAN dataset using Adam and Lion. We plot the loss landscapes on (a) the fine-tuning dataset and (b) the pre-training dataset (Pile dataset (Gao et al., 2020)). We visualize a 2D weight-space plane spanned by the vector from the pre-trained model to the Lion-tuned model (x-axis) and to the Adam-tuned model (y-axis). Axes are normalized so that one unit equals the length of the pre-trained→Adam vector. The color bar indicates the loss value—(a) fine-tuning loss and (b) pre-training loss. A logarithmic scale is applied to the loss values for better visualization. Two training methods converge to different minima with similar fine-tuning loss. Lion converges to a farther minimum from the pre-trained model and performs more forgetting than Adam.

- We propose MoFO, a new training algorithm designed to mitigate the forgetting of pre-training knowledge during fine-tuning.

- We present a rigorous theoretical convergence result of the MoFO algorithm, providing a solid theoretical foundation that supports its good performance in fine-tuning tasks.

- We conduct experiments on various tasks, demonstrating that MoFO outperforms existing methods both in fine-tuning performance and mitigating forgetting.

## 2 Momentum Filtered Optimizer (MoFO)

### 2.1 Motivation

**Correlation between Distance and Forgetting.**

In LLM fine-tuning, different training methods typically converge to different minima of the loss function. These minima may yield similarly low fine-tuning loss, all achieving reasonably good fine-tuning performance. However, their distances from the pre-trained model can vary significantly. A key observation in traditional continual learning (CL) is that the extent of forgetting increases as the model deviates further from its original state. This insight has influenced the design of many forgetting-mitigation methods (Kirkpatrick et al., 2017; Li et al., 2018).

In this work, we conduct exploratory experiments to investigate this correlation among minima produced by different LLM fine-tuning methods. We conduct two sets of experiments. First, we fine-tune Pythia-160M on a subset of the FLAN dataset[3] using Adam (Kingma & Ba, 2014) and Lion (Chen et al., 2024). As illustrated in Figure 1(a), Adam and Lion converge to distinct minima. Notably, while both optimizers achieve similar fine-tuning losses, the minimum reached by Adam is much closer to the pre-trained model compared to Lion, being only about 20% of Lion's distance. (Specifically we measure the model distance by the average $L_2$ norm difference over parameter blocks; see Appendix C.4 for the formal definition). As shown in Figure 1(b),

---

[3]The subset used is 'definite_pronoun_resolution_10templates,' available at `https://huggingface.co/datasets/Muennighoff/flan`. The learning rate is 2e-5 and the batch size is set as 64.

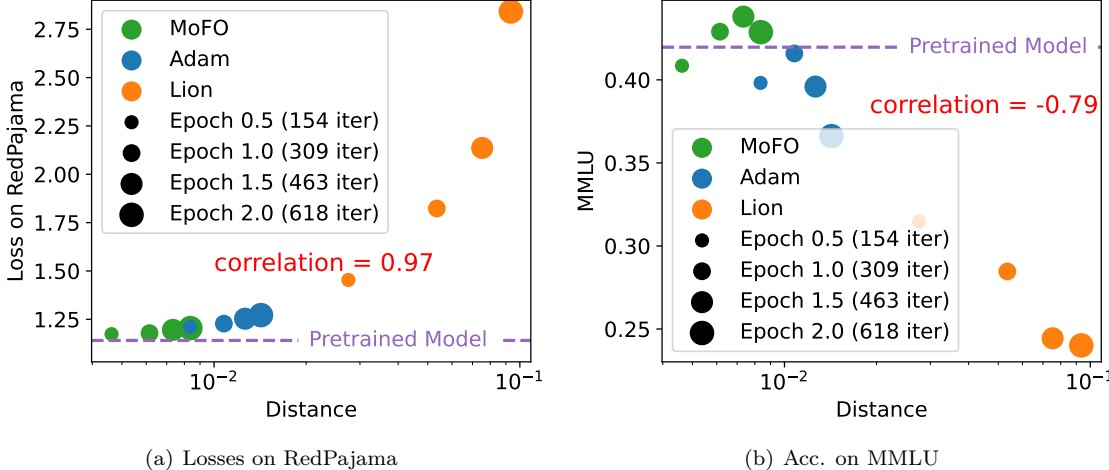

(a) Losses on RedPajama

(b) Acc. on MMLU

Figure 2: (a) Loss changes on the RedPajama dataset and (b) average accuracy changes on MMLU benchmark (measuring the preservation of factual knowledge) of Llama-2-7B after fine-tuning on MetaMathQA using Adam, Lion, and MoFO for 0.5, 1, 1.5, 2 epochs. We note that RedPajama project was explicitly designed as an open-source reproduction of the LLaMA training dataset (Weber et al., 2024). Thus, it serves as a reasonable proxy for the original LLaMA-2 training dataset since the latter has not been publicly released. See Appendix B.1 for the rationale. The results show a strong positive correlation between the distance from the pre-trained model and the extent of forgetting after one epoch. Further discussion of early-training behavior and a comparison of different optimizers are provided in Appendix B.3.

Adam exhibits a smaller increase in pre-training loss than Lion. Further, Adam leads to better preservation in general capability than Lion (see Appendix B.2).

Second, we conduct a larger-scale exploratory experiment to test the generality of this observation. We fine-tune a larger model, LLaMA2-7B, on the MetaMathQA dataset (Yu et al., 2024b) using three optimizers: Adam, Lion, and our proposed MoFO (to be introduced in Section 2.2). To achieve varying distances from the pre-trained model, we fine-tune each model for 0.5, 1, 1.5, and 2 epochs (309 steps per epoch). Details are provided in Appendix C.3. Figure 2(a) demonstrates a strong positive correlation between distance and the increase in pre-training loss. Figure 2(b) indicates a negative correlation with MMLU accuracy (Hendrycks et al., 2021), which measures the preservation of factual knowledge, once the model has been trained for at least one epoch. Taken together, these results suggest that, during LLM fine-tuning, remaining closer to the pre-trained state appears to be associated with reduced forgetting. Further discussion of early-training behavior and a comparison of different optimizers are provided in Appendix B.3.

**Selective Rule Based on Momentum Magnitudes**

Motivated by the correlation between forgetting and the distance from the pre-trained model, we seek to design an optimizer that encourages the fine-tuned model to keep closer to the pre-trained model. To achieve this, we draw inspiration from the classical block coordinate descent (BCD) method (Tseng, 2001), which updates only a subset of parameters during each iteration. We anticipate that restricting updates to a subset of parameters—similar to the BCD approach—will result in smaller overall deviations from the pre-trained model compared to full-parameter updates across all iterations, thereby mitigating the forgetting of pre-training knowledge.

To further accelerate convergence under the BCD framework, we adopt Gauss-Southwell rule, i.e., the greedy rule (Nutini et al., 2015). Gauss-Southwell rule selects the parameters with the largest gradients at each iteration, as those are expected to yield the greatest immediate reduction in the loss. It has also been shown in Nutini et al. (2015) that BCD using the Gauss-Southwell rule—also referred to as greedy BCD—can converge faster than the traditional random BCD. However, BCD algorithms, including greedy BCD, are mostly developed based on the GD or SGD framework, but in LLM training, Adam has replaced SGD as

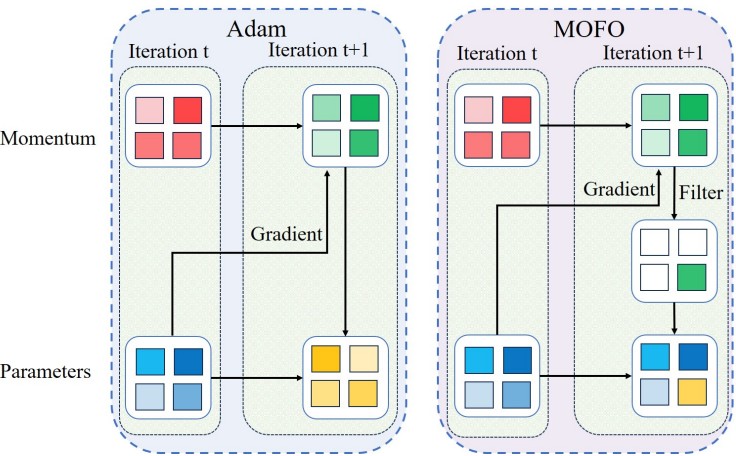

Figure 3: Illustration of MoFO.

the default optimizer (Zhang et al., 2024c). Our experiments in Section 4.4 show that directly following the Gauss–Southwell rule in Adam—that is, always updating parameters with large gradients—does not lead to satisfactory performance on fine-tuning tasks.

Adam inherently incorporates momentum term in parameter updates. Therefore, we propose to modify the Adam optimizer to update only the parameters with the *largest momentum magnitudes*. By focusing on partial yet significant updates, our method, named MoFO, aims to effectively fine-tune models while maintaining closer to their pre-trained state. We first introduce MoFO in the next subsection. Further theoretical analysis and empirical exploration in the selection rule will be provided in Section 3.2 and 4.4, respectively.

## 2.2 Formulation of MoFO

---

**Algorithm 1** Momentum Filtered Optimizer (MoFO)

---

1: Input: Filtering threshold $\alpha$, number of partitions $B$ with the $k$-th partition of size $d_k$, hyperparameters $\beta_1, \beta_2$ of Adam optimizer, learning rate schedule $\{\eta_t\}$.
2: Initialize $m_0, v_0$ as zero tensors.
3: **for** iteration $t$ from $1, 2, \ldots$ until converge **do**
4:     **for** partition $k$ from 1 to $B$ **do**
5:         $g_t^{(k)} = \nabla_{(k)} \mathcal{L}_{finetune}(\theta_{t-1})$
6:         $m_t^{(k)} = \beta_1 m_{t-1}^{(k)} + (1 - \beta_1) g_t^{(k)}$
7:         $v_t^{(k)} = \beta_2 v_{t-1}^{(k)} + (1 - \beta_2) g_t^{(k)} \circ g_t^{(k)}$
8:         $\hat{m}_t^{(k)} = m_t^{(k)} / (1 - \beta_1^t)$
9:         $\hat{v}_t^{(k)} = v_t^{(k)} / (1 - \beta_2^t)$
10:         **for** entry index $i$ from 1 to $d_k$ **do**
11:             $[\texttt{FLT}_\alpha^{(k)}(m_t)]_i = 1$ **if** $|(m_t^{(k)})_i|$ is within the top-$\alpha$ of $|m_t^{(k)}|$'s values **else** 0
12:         **end for**
13:         $\theta_t^{(k)} = \theta_{t-1}^{(k)} - \eta_t \cdot (\hat{m}_t^{(k)} \odot \texttt{FLT}_\alpha^{(k)}(m_t)) / \sqrt{\hat{v}_t^{(k)}}$       *# Momentum Filtering*
14:     **end for**
15:     $\theta_t = \texttt{Concat}(\theta_t^{(1)}, \ldots, \theta_t^{(B)})$
16: **end for**

---

We formally introduce the Momentum-Filtered Optimizer (MoFO) in Algorithm 1. First, all model parameters are partitioned into $B$ blocks. At each iteration, MoFO first computes the gradient and momentum terms for parameters in each block following the standard rule of Adam, as shown in Lines 5-9. Then, MoFO selects and updates the parameter entries with the largest $\alpha$ momentum magnitudes in each parameter block, as

shown in Lines 10-13, where the update fraction $\alpha$ is a pre-determined hyperparameter. This momentum filtering mechanism is illustrated in Figure 3.

Mathematically, the filter can be represented as follows. Consider a momentum vector $m = (m^{(1)}, \ldots, m^{(B)})$, where each $m^{(k)} \in \mathbb{R}^{d_k}$ corresponds to the $k$-th block of parameters with dimensionality $d_k$. The top-$\alpha$ filter, denoted as $\mathtt{FLT}_\alpha(m)$, is defined as $\mathtt{FLT}_\alpha(m) = (\mathtt{FLT}_\alpha^{(1)}(m), \ldots, \mathtt{FLT}_\alpha^{(B)}(m))$, where the $i$-th entry of $\mathtt{FLT}_\alpha^{(k)}(m)$ is given by

$$\left[ \mathtt{FLT}_\alpha^{(k)}(m) \right]_i = \begin{cases} 1 & \text{if } |m_i^{(k)}| \text{ is within the top-}\alpha \text{ of } |m^{(k)}| \text{ values,} \\ 0 & \text{otherwise,} \end{cases}$$

for $i = 1, 2, \cdots, d_k$, $k = 1, 2, \cdots, B$. In our Momentum-Filtered Optimizer (MoFO), this filter $\mathtt{FLT}_\alpha$ is applied to the momentum $m_t$, selecting the entries with the largest magnitudes for updating.

For the parameter partitioning, we note that the network architecture is naturally composed of different modules (e.g., weight matrices, and bias terms). In the PyTorch implementation, the parameters of different modules (along with their gradients and momenta) are naturally stored in separate data tensors. Therefore, we adopt the default partitioning of model parameters as implemented in PyTorch. For Transformers, this means that parameters such as query (Q), key (K), value (V) weights in the attention layers, as well as feed-forward network (FFN) weights, are grouped into distinct partitions following PyTorch's default scheme. This allows us to select and update the top-$\alpha$ parameters in each block without introducing much implementation overhead. See Appendix C.4 for further explanation of the partitioning.

At each iteration, MoFO efficiently selects and updates the most "influential" parameters, as dictated by the momentum's magnitude, while keeping other parameters fixed. We argue that filtering the momentum is more effective than filtering the gradient. In Section 4.4, we will empirically demonstrate that MoFO's momentum-based filtering rule outperforms other filtering rules in fine-tuning tasks.

## 3 Theoretical Analysis

### 3.1 Convergence Result

In this section, we present the convergence result of MoFO for non-convex loss functions. For the simplicity of analysis, we consider the full-batch version of MoFO, with hyperparameters satisfying the following assumption.

**Assumption 1.** *Loss function $\mathcal{L}$ is lower bounded by $\mathcal{L}^*$. The gradient $\nabla\mathcal{L}$ is Lipschitz continuous with constant $L$.*

**Theorem 1** (Convergence of MoFO)**.** *Suppose that the first- and second-order momentum hyperparameters $\beta_1$ and $\beta_2$ satisfy $0 < \beta_1 < \sqrt{\beta_2} < 1$. The learning rate schedule at step $t$ is $\eta_t = \eta/\sqrt{t}$ for some $\eta > 0$. Then, under Assumption 1, MoFO satisfies*

$$\min_{0 \le t \le T-1} \| \nabla\mathcal{L}(\theta_t) \odot FLT_\alpha\big(\nabla\mathcal{L}(\theta_t)\big) \|_1 = \mathcal{O}\left( \frac{\log T}{\sqrt{T}} \right) \quad \text{as } T \to \infty.$$

*Moreover, this bound directly implies*

$$\min_{0 \le t \le T-1} \| \nabla\mathcal{L}(\theta_t) \|_p = \mathcal{O}\left( \frac{\log T}{\sqrt{T}} \right) \quad \text{as } T \to \infty,$$

*for any $p \in [1, \infty]$.*

Although MoFO is designed to mitigate forgetting by updating only a small subset of parameters at each step, it is guaranteed to converge to a critical point of the fine-tuning loss function under the Lipschitz smoothness assumption. This result provides theoretical evidence that MoFO can achieve competitive performance in fine-tuning tasks.

***Proof Sketch of Theorem 1:*** Our proof is inspired by the convergence analysis for full-batch Adam in Shi et al. (2021), but we introduce additional techniques tailored to MoFO's filtering mechanism. We will highlight these additions precisely at the points where they arise below.

Let $g_t = \nabla \mathcal{L}(\theta_{t-1})$. A central step is to establish, for suitable constants $C_1, C_2 > 0$,

$$\frac{C_1}{\sqrt{t}}\|g_t\| \le \mathcal{L}(\theta_{t-1}) - \mathcal{L}(\theta_t) + \frac{C_2}{t}, \tag{1}$$

Summing this inequality from $t = 1$ to $T$ and using $\sum_{t=1}^{T} t^{-1} = \log T + \mathcal{O}(1)$ yields the convergence result for Adam in terms of a diminishing norm of gradient, given by

$$\min_{1 \le t \le T} \|g_t\| = \mathcal{O}\left(\frac{\log T}{\sqrt{T}}\right). \tag{2}$$

**Choice of norm and two subgoals.** In finite-dimensional spaces, all norms are equivalent, so convergence statements like (1)–(2) can be expressed in any fixed norm up to norm-equivalence constants. In practice, however, specific analyses instantiate (1) with a particular norm: Shi et al. (2021) work with the $L_1$ norm $\|g_t\|_1$ for full-batch Adam, whereas our full-batch MoFO analysis will use the $L_{1,\text{top-}\alpha}$ norm $\|g_t \odot \texttt{FLT}_\alpha(g_t)\|_1$, which will be defined in Appendix A.1.

To keep the logic precise, we separate our argument into two subgoals:

**Step I (Key inequality for MoFO).** Show that there exist constants $C_1, C_2 > 0$ (independent of $t$) such that

$$\frac{C_1}{\sqrt{t}} \left\|g_t \odot \texttt{FLT}_\alpha(g_t)\right\|_1 \le \mathcal{L}(\theta_{t-1}) - \mathcal{L}(\theta_t) + \frac{C_2}{t}. \tag{3}$$

Here, we recall that $\texttt{FLT}_\alpha(\cdot)$ preserves the $\alpha$-fraction of largest-magnitude coordinates in each partition and zeros out the rest.

**Step II (Norm property of the left-hand side of (3)).** *This component is one of our new technical ingredients (absent from Shi et al. (2021)).* We first verify that the mapping $x \mapsto \|x \odot \texttt{FLT}_\alpha(x)\|_1$ defines a norm on $\mathbb{R}^d$, which is referred to as $L_{1,\text{top-}\alpha}$ norm. This is proved in Proposition 1 (Appendix A.1) by checking nonnegativity and definiteness, positive homogeneity, and the triangle inequality. Moreover, for the $L_p$ upper bound in Theorem 1, we use the norm equivalence between the $L_{1,\text{top-}\alpha}$ norm and the $L_p$ norm, as shown in Lemma 2.

**With Step II established, it remains to prove the key inequality in Step I.** A direct adaptation of Shi et al. (2021) to MoFO is infeasible due to structural differences. We proceed as follows:

(i) Recap key elements from Shi et al. (2021);

(ii) Identify challenges in extending them to MoFO;

(iii) Resolve these challenges by carefully handling the momentum filter.

***Part (i): Key elements of Shi et al. (2021).*** For full-batch Adam with bias-corrected moments $\hat{m}_t, \hat{v}_t$ and the learning rate schedule $\eta_t = \eta/\sqrt{t}$, the parameter update is

$$\theta_t - \theta_{t-1} = -\frac{\eta}{\sqrt{t}} \cdot \frac{\hat{m}_t}{\sqrt{\hat{v}_t}}.$$

By the $L$-smoothness of the loss and the descent lemma,

$$\frac{\eta}{\sqrt{t}} \sum_{i=1}^{d} g_{i,t} \frac{\hat{m}_{i,t}}{\sqrt{\hat{v}_{i,t}}} \le \mathcal{L}(\theta_{t-1}) - \mathcal{L}(\theta_t) + \frac{L}{2}\|\theta_t - \theta_{t-1}\|_2^2. \tag{4}$$

Lemma 5 in Appendix A.2 lower-bounds the per-coordinate contribution as

$$g_{i,t} \frac{\hat{m}_{i,t}}{\sqrt{\hat{v}_{i,t}}} \ge A|g_{i,t}| - \frac{B}{\sqrt{t}},$$

for some constants $A, B > 0$. Substituting this into (4), summing over coordinates, and controlling the quadratic term in (4) yields the fundamental inequality (1) with the $L_1$-norm:

$$\frac{C_1}{\sqrt{t}}\|g_t\|_1 \leq \mathcal{L}(\theta_{t-1}) - \mathcal{L}(\theta_t) + \frac{C_2}{t},$$

***Part (ii): Challenges in extending to MoFO.*** For the full-batch version of MoFO, the parameter update becomes

$$\theta_t - \theta_{t-1} = -\frac{\eta}{\sqrt{t}}\frac{\hat{m}_t}{\sqrt{\hat{v}_t}} \odot \mathtt{FLT}_\alpha(m_t),$$

i.e., the step is filtered by momentum magnitudes (*NOT gradient magnitudes*). Building upon the convergence analysis of Shi et al. (2021) in Part (i) leads to

$$\frac{C_1}{\sqrt{t}}\|g_t \odot \mathtt{FLT}_\alpha(m_t)\|_1 \leq \mathcal{L}(\theta_{t-1}) - \mathcal{L}(\theta_t) + \frac{C_2}{t}, \tag{5}$$

which brings a notable difference to the target key inequality (3): the inequality (5) applies the momentum filter $\mathtt{FLT}_\alpha(m_t)$, whereas the desired bound applies the gradient filter $\mathtt{FLT}_\alpha(g_t)$. This introduces a non-trivial challenge because the $\|g_t \odot \mathtt{FLT}_\alpha(m_t)\|_1$ being small does not naturally imply that $\|g_t \odot \mathtt{FLT}_\alpha(g_t)\|_1$ is small: large entries of $g_t$ might be excluded if their momenta lags. Consequently, additional analysis is required to bound the discrepancy between them.

***Part (iii): How we overcome the challenge.*** *Here we introduce another new technique to address the discrepancy between* $\|g_t \odot FLT_\alpha(m_t)\|_1$ *and* $\|g_t \odot FLT_\alpha(g_t)\|_1$. We deal with the challenge via Lemma 1 in Appendix A.1, which establishes that the $L_1$-deviation in filtered outputs is *Lipschitz-stable* under input perturbations. Specifically, for any $x, y \in \mathbb{R}^d$:

$$\underbrace{\|x \odot \mathtt{FLT}_\alpha(x)\|_1 - \|x \odot \mathtt{FLT}_\alpha(y)\|_1}_{\text{top-}\alpha \text{ filtered error}} \leq 2\underbrace{\|x - y\|_1}_{\text{input error}}.$$

This result points out that while the filter $\mathtt{FLT}_\alpha(\cdot)$ itself is unstable under perturbation, the filtered output remains controllable through. It effectively "smooths" the discontinuity that would plausibly prevent convergence analysis.

With the Lipschitz stability established, we now deal with the challenge. We first control the $L_1$ distance between the bias-corrected momentum $\hat{m}_t = \frac{m_t}{1-\beta_1^t}$ and the gradient $g_t$ by showing that $\|\hat{m}_t - g_t\|_1 = \mathcal{O}(1/\sqrt{t})$. Second, we apply Lemma 1 with $x = g_t$ and $y = \hat{m}_t$ and yield

$$\|g_t \odot \mathtt{FLT}_\alpha(g_t)\|_1 - \|g_t \odot \mathtt{FLT}_\alpha(\hat{m}_t)\|_1 \leq 2\|\hat{m}_t - g_t\|_1 = \mathcal{O}(1/\sqrt{t}).$$

Since the filtering function $\mathtt{FLT}_\alpha(\cdot)$ is invariant under positive scaling, we can definitely replace $\hat{m}_t$ with the original momentum $m_t$:

$$\|g_t \odot \mathtt{FLT}_\alpha(g_t)\|_1 - \|g_t \odot \mathtt{FLT}_\alpha(m_t)\|_1 \leq 2\|\hat{m}_t - g_t\|_1 = \mathcal{O}(1/\sqrt{t}).$$

Combining it with (5) and subsuming residual $\mathcal{O}(1/t)$ terms, we obtain our target key inequality (3):

$$\frac{C_1}{\sqrt{t}}\|g_t \odot \mathtt{FLT}_\alpha(g_t)\|_1 \leq \mathcal{L}(\theta_{t-1}) - \mathcal{L}(\theta_t) + \frac{C_2}{t}.$$

In conclusion, our statement of **Step I** primarily comprises:

- **Step I.1 (Part (i) and (ii)).** Extending the Adam convergence framework (Shi et al., 2021) to incorporate MoFO's momentum filtering mechanism, yielding (5);

- **Step I.2 (Part (iii)).** Resolving the momentum-gradient filter discrepancy via our Lipschitz stability analysis (Lemma 1), which is a new technique introduced in this work.

Combined with the norm property in **Step II**—another new ingredient not present in Shi et al. (2021)—, these results enable us to establish Theorem 1.

$\square$

We remark that the choice of $\beta_1$ and $\beta_2$ in Assumption 1 aligns with that used in analyzing full-batch Adam (Shi et al., 2021). Furthermore, the use of a diminishing learning rate in Theorem 1 is crucial for ensuring the stability of updates and avoiding divergence in the optimization process.

In summary, Theorem 1 demonstrates that despite updating only a subset of parameters, MoFO maintains the same convergence rate as Adam. This highlights the theoretical robustness of the momentum filter design in MoFO. We believe this result could provide valuable insights into adaptive optimization methods with filtering mechanisms.

### 3.2 Initial Analysis on Forgetting Mitigation

Does MoFO converge to a model that is closer to the pre-trained LLM than Adam, thereby reducing forgetting? In this subsection, we attempt to address this question by providing an initial theoretical analysis on an illustrative example.

**Example 1.** *Suppose the parameter space is $\mathbb{R}^d$, and the updating ratio is $\alpha = 1/d$, i.e., only one coordinate is updated in each iteration. We assume the **pre-training loss** is $\mathcal{L}_{\mathrm{pretrain}}(\theta) = \frac{1}{2}\|\theta\|_2^2$ and that the model has been trained to the global minimum $\theta_{\mathrm{pretrain}} = (0, 0, \ldots, 0)$ during the pre-training phase. The **fine-tuning loss** is given by $\mathcal{L}(\theta) = \prod_{i=1}^{d}(a_i\theta_i - b_i)^2$, where $a_i, b_i > 0$ for any $1 \le i \le d$. In this example, the set of global minima of $\mathcal{L}(\theta)$ is a union of hyperplanes:*

$$S = \bigcup_{i=1}^{d} S_i, \quad \text{where } S_i := \{\theta \in \mathbb{R}^d : \theta_i = b_i/a_i\}.$$

*Here, each $S_i$ represents a hyperplane in $\mathbb{R}^d$.*

**Remark 1.** *We note that the loss landscapes of neural networks are generally non-convex (Liu et al., 2022). Here, we also adopt a non-convex fine-tuning loss $\mathcal{L}(\theta)$. Further, we note that the set of global-minima $S$ consists of infinitely many minima spread across multiple hyperplanes, aligning with the observation on the degenerate structure of minima in neural networks (Lin et al., 2024b).*

**Theorem 2.** *In Example 1, if the learning rates are chosen appropriately, then MoFO converges to a minimum $\theta_{\mathrm{MoFO}}^*$ that is closer to the pre-training state than the minimum $\theta_{\mathrm{Adam}}^*$ obtained by Adam, i.e.,*

$$\|\theta_{\mathrm{MoFO}}^* - \theta_{\mathrm{pretrain}}\|_2 < \|\theta_{\mathrm{Adam}}^* - \theta_{\mathrm{pretrain}}\|_2.$$

*Moreover, MoFO attains a strictly lower pre-training loss, i.e.,*

$$\mathcal{L}_{\mathrm{pretrain}}(\theta_{\mathrm{MoFO}}^*) < \mathcal{L}_{\mathrm{pretrain}}(\theta_{\mathrm{Adam}}^*).$$

Next, we visualize the landscape of this example to provide additional intuition. Consider a simplified two-dimensional case. Specifically, let $\theta \in \mathbb{R}^2$ and $\mathcal{L}(\theta) = (\theta_1 - 1)^2(\theta_2 - 1)^2$.

In Example 1, each hyperplane $S_i$, which forms part of the global minima of $\mathcal{L}$, can be viewed as an attractor for the optimization algorithms. These attractors ($S_i$'s) influence the model's update direction during training. As illustrated in Figure 4, the attractors in this case are two straight lines: $\theta_1 = 1$ and $\theta_2 = 1$. When using Adam, the model is simultaneously pulled by both attractors, causing it to move diagonally along the orange line and converge at $(1, 1)$, with a resulting pre-training loss of 1.

In contrast, MoFO is influenced by only one attractor ($\theta_1 = 1$), leading it to follow the green line and converge to $(1, 0)$, which achieves a lower pre-training loss of $0.5$[4]. We hypothesize that, for full-parameter fine-tuning (Adam), interference among multiple attractors drives convergence toward a "balanced" solution, whereas

---

[4]By symmetry, MoFO may also be influenced by another attractor $\theta_2 = 1$ and converges vertically to $(0, 1)$.

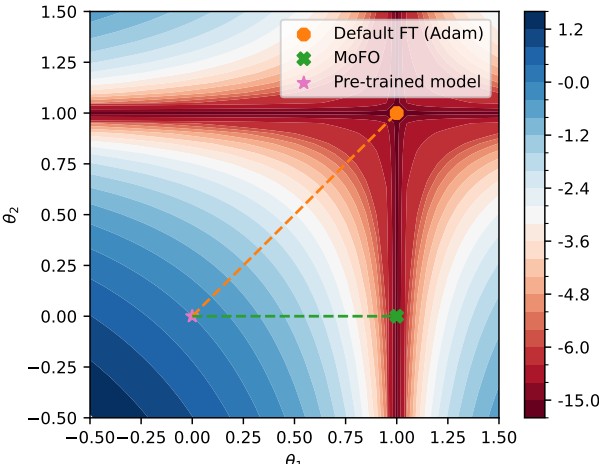

Figure 4: The fine-tuning loss landscape and the training paths of different optimization methods. The color bar indicates the fine-tuning loss value.
A logarithmic scale is applied to the loss values for better visualization. MoFO converges to a minimum closest to the pre-trained model.

MoFO mitigates such interference by selectively updating only parameters with large momentum magnitudes. Such an updating rule helps MoFO move toward a single attractor, thereby converging closer and forgetting less. In addition to Adam and MoFO, we provide convergence paths of some other baseline methods in Appendix B.4.

We believe that the above analysis provides an initial insight into the effectiveness of MoFO. More in-depth analysis of MoFO in forgetting mitigation is left for future work.

## 4 Experiments

### 4.1 Experimental Settings

We verify the effectiveness of MoFO on **instruction fine-tuning** and **continual fine-tuning**. We use Llama-2-7B (Touvron et al., 2023), Gemma-2B-IT (Team et al., 2024), and TinyLlama-1.1B (Zhang et al., 2024b) as our base models. The instruction fine-tuning datasets cover question-answer pairs from different domains like mathematical reasoning and medical knowledge. Specifically, the datasets include: MetaMathQA (Yu et al., 2024b), PMC-LLaMA-Instructions (Wu et al., 2024), Magicoder-Evol-Instruct (Wei et al., 2023). We randomly sample 39.5K and 51K instances from these datasets, respectively, for training the LLMs. Additionally, We investigate the performance of MoFO in the continual fine-tuning scenario by implementing our approach on the TRACE benchmark dataset (Wang et al., 2023b).

**Evaluation metrics for instruction fine-tuning.** We employ widely used benchmarks to assess the performance and potential forgetting effects on the general capabilities of LLMs after instruction fine-tuning. These benchmarks include MMLU (Hendrycks et al., 2021) (0-shot) for factual knowledge; ARC-Challenge, ARC-Easy (Clark et al., 2018), and HellaSwag (Zellers et al., 2019) (0-shot) for commonsense reasoning (CR); GSM8K (Cobbe et al., 2021) (5-shot) for mathematical reasoning; HumanEval (HEval) (Chen et al., 2021) (pass@10) for code generation; PubMedQA (Jin et al., 2019), MedMCQA (Pal et al., 2022), and MedQA (Jin et al., 2021) (0-shot) for medical question answering (MedQ) [5]; IFEval (0-shot) for instruction following.

**Evaluation metrics for continual fine-tuning.** To evaluate the LLM's performance in continual learning, we consider two key metrics in this scenario: Overall Performance (OP) (Chaudhry et al., 2018) and BackWard Transfer (BWT) (Lopez-Paz & Ranzato, 2017).

For more descriptions and implementation details of these metrics and datasets, see Appendix C.

---

[5]For CR and MedQ, we report the average of the benchmarks they comprise.

Table 1: The performance of the fine-tuning task (math), measured by GSM8K, and the general capability scores of Llama-2-7B after fine-tuning on the MetaMathQA dataset. The results show that MoFO achieves comparable performance in the fine-tuning task, while significantly mitigating forgetting of general capabilities. Bold values denote the best results among these methods.

| Method | GSM8K | General Capability | | | |
| --- | --- | --- | --- | --- | --- |
| | | CR | MMLU | HEval | Avg. |
| Llama-2-7B | 13.7 | 65.6 | 42.0 | 24.2 | 43.9 |
| Default FT | 49.4 | 62.3 | 36.6 | 16.1 | 38.3 |
| HFT | 47.5 | 65.5 | 42.3 | 23.6 | 43.8 |
| LoRA | 43.3 | 65.1 | 37.7 | **26.4** | 43.1 |
| MoFO | 47.7 | **65.7** | **42.7** | 24.6 | **44.3** |

## 4.2 Instruction Fine-Tuning

In this section, we investigate the effectiveness of the MoFO algorithm in both preserving general capabilities and learning fine-tuning tasks. The implementation details are provided in Appendix C. The specific hyperparameter settings in each experiment are provided in Appendix C.3.

**LLM Fine-tuning strategy baselines.** We compare MoFO with the default fine-tuning approach and other methods designed to mitigate forgetting. These baselines include: **Default fine-tuning (Default FT)** refers to the full-parameter fine-tuning approach using the Adam optimizer. **Half Fine-tuning (HFT)** (Hui et al., 2024) randomly updates half of the parameter blocks within each transformer layer at each iteration while the other half are frozen. HFT can be considered a specific case of the BCD algorithm. **LoRA** (Hu et al., 2022) is a widely-used, parameter-efficient fine-tuning method. LoRA trains low-rank matrix adaptations on the base model's weights. Recent work (Biderman et al., 2024) demonstrates that LoRA can mitigate forgetting.

**Results of fine-tuning on MetaMathQA.** We fine-tune Llama-2-7B on MetaMathQA using various baseline methods and present the experimental results on mathematical reasoning (GSM8K) and general capabilities in Table 1. We report the experimental results of LoRA under the best-performing hyperparameter configuration on the fine-tuning task. These results demonstrate the effectiveness of our proposed MoFO algorithm in both optimization and mitigating forgetting.

MoFO is compatible to the performance of Default FT and HFT on the math task, yet significantly outperforms these methods in preserving general capability. Specifically, Default FT shows a decline of 5.4% in MMLU accuracy and HFT experiences a drop of 0.6% in HumanEval. In contrast, our MoFO not only maintains but slightly improves these general capability scores by an average of 0.4%.

**Comparison from a Pareto perspective.** Generally, improving performance on the fine-tuning task and reducing forgetting are often a pair of competing objectives. It is intriguing to study how different fine-tuning methods balance this tradeoff. By adjusting the hyperparameters of different methods, we can observe a set of fine-tuned models, each representing a different tradeoff between fine-tuning performance and forgetting. The Pareto frontier formed by these models helps visualize the tradeoffs, and we can identify which method offers the best balance between fine-tuning and forgetting.

In this comparison, we also include traditional regularization methods such as $L_2$-regularization (Li et al., 2018) (denoted as $L_2$ reg) and $L_1$-regularization (Panigrahi et al., 2023) (denoted as $L_1$ reg), which are not specifically designed for large models. These methods modify the original fine-tuning loss $\mathcal{L}_{finetune}(\theta)$ by adding a regularization term. For $L_2$-regularization, the modified loss is $\mathcal{L}_{finetune}(\theta) + \lambda_2 \|\theta - \theta_0\|_2^2$, and for $L_1$-regularization, it is $\mathcal{L}_{finetune}(\theta) + \lambda_1 \|\theta - \theta_0\|_1$, where $\lambda_2$ and $\lambda_1$ are the respective regularization hyperparameters.

We fine-tune the Llama-2-7B model on the MetaMathQA dataset using $L_1$ and $L_2$ regularization, as well as LoRA, and compare their performance with MoFO. We present the results in Figure 5 and plot Pareto optimal fronts[6] for these methods. Details of the hyperparameter configurations for this experiment are provided in Appendix C.3. These results show the effectiveness of the MoFO algorithm in both optimization and mitigating forgetting.

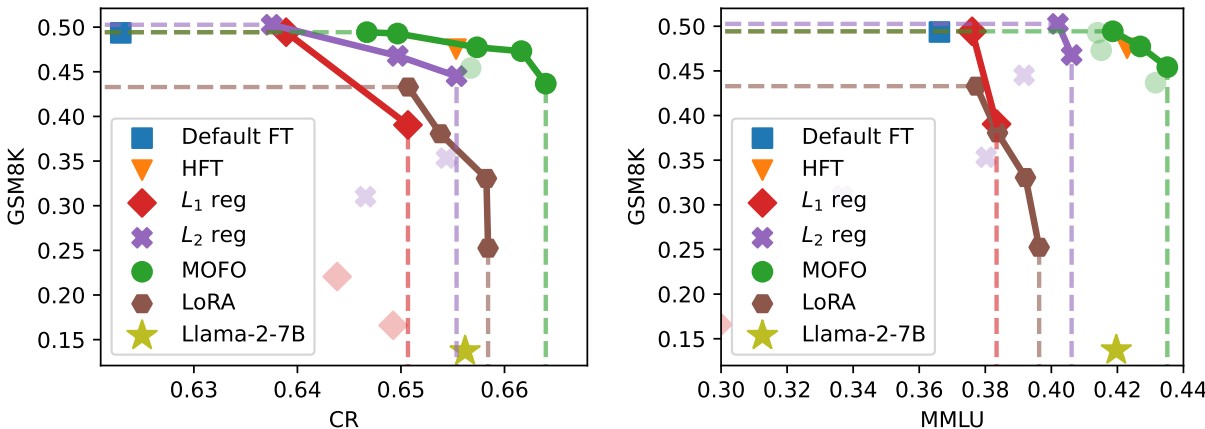

Figure 5: The performance on the math task (GSM8K) and the scores in general capabilities of Llama-2-7B after fine-tuning on the MetaMathQA dataset. Only points on the Pareto front are shown as solid points, while the remaining points are presented as semi-transparent. The results show that compared with $L_1$, $L_2$ regularization, and LoRA across various hyperparameter configurations, the MoFO algorithm achieves a better Pareto front.

The result reveals that MoFO consistently achieves a better Pareto front in comparison to baseline methods. When compared to regularization methods and LoRA, MoFO exhibits less forgetting and can even maintain general capabilities with comparable GSM8K accuracies. Additionally, MoFO outperforms regularization methods in math tasks when the magnitudes of forgetting are similar. We also note that $L_1$ and $L_2$ regularization (Panigrahi et al., 2023; Li et al., 2018) require storing the pre-training weights throughout the entire fine-tuning process for regularization computation, which incurs additional memory overhead. For preliminary analysis on why MoFO might compare favorably to $L_1/L_2$ regularization, see Appendix F.5.

Appendix E reports additional instruction fine-tuning results. Specifically, we evaluate

- LLM variants: Gemma-2B-IT and Llama-2-7B-Chat;
- Domain-specific datasets: medical dataset (PMC-LLaMA-Instruct (Wu et al., 2024)), coding dataset (Magicoder-Evol-Instruct (Wei et al., 2023));
- Different baselines: HMA (Lin et al., 2024a), CoFiTune (Zhang et al., 2024a), Soft-masking (Ke et al., 2023a;b).

**MoFO Converges Closer to the Pre-trained Model.**

In this part, we empirically investigate whether MoFO converges closer to the pre-trained model. Building on the fine-tuned models in Table 1, we compare their distances to the pre-trained model. In addition, for the $L_1$ and $L_2$ regularization baselines in our Pareto analysis above, we select the models that achieve the best performance on the GSM8K benchmark (corresponding to the fine-tuning task).

Figure 6 shows that models fine-tuned with MoFO are closer to the pre-trained model compared to other baseline methods.

---

[6]Since it is impractical to exhaust all hyperparameter configurations in real experiments, we present linear interpolation approximations of the Pareto fronts in Figure 5.

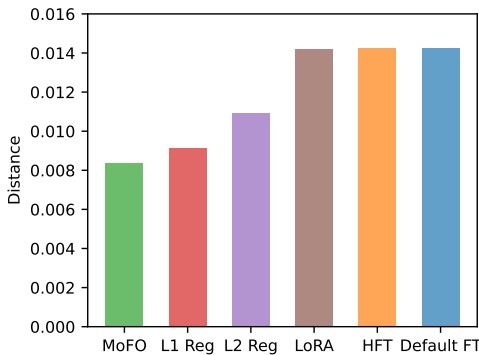

Figure 6: The distances for the fine-tuned Llama2-7B on MetaMathQA. The results show MoFO achieves minima closer to the pre-trained model.

### 4.3 Continual Fine-Tuning

In this section, we explore the performance of our proposed MoFO in continual fine-tuning on the TRACE benchmark (Wang et al., 2023b). We sequentially train TinyLlama-1.1B on the TRACE dataset, which includes the eight tasks from different domains. The implementation details are provided in Appendix C.

**Continual learning baselines.** We consider several traditional methods from the field of continual learning to compare with MoFO. These methods can also be orthogonal combined with MoFO to further enhance performance. **Replay** involves optimizing the model using current data along with a memory buffer containing samples from previous tasks to mitigate forgetting, and we follow the implementation in Wang et al. (2023b). **Gradient of Episodic Memory (GEM)** (Lopez-Paz & Ranzato, 2017) mitigates forgetting by using gradients from old tasks to adjust the parameter updates during the training of new tasks. **Elastic weight consolidation (EWC)** (Kirkpatrick et al., 2017) uses a diagonal approximation of Fisher information matrix, which can be calculated by gradients from previous tasks, to regularize parameter updates.

Table 2: The OP and BWT scores of TinyLlama-1.1B after fine-tuning on TRACE benchmark. The results show that MoFO outperforms Default FT, HFT, GEM, and EWC in continual learning and can combine well with continual learning methods. Bold values denote the best results among these methods in each group.

|  | OP | BWT |
| :---: | :---: | :---: |
| Default FT | 38.4 | -10.3 |
| HFT | 39.9 | -10.1 |
| MoFO | **41.3** | **-5.4** |
| GEM | 40.8 | -8.5 |
| GEM + MoFO | **41.7** | **-6.7** |
| EWC | 41.1 | -8.3 |
| EWC + MoFO | **43.2** | **-4.4** |
| Replay | 45.5 | 4.7 |
| Replay + MoFO | **47.0** | **4.8** |

**Results of continual fine-tuning.** We present the experimental results of sequentially fine-tuning TinyLlama-1.1B on the TRACE benchmark with various methods in Table 2. The results indicate that in continual fine-tuning, MoFO not only outperforms other fine-tuning baselines but also surpasses GEM and EWC. Moreover, MoFO combines well with the Replay method, offering a 1.5% performance gain on the OP metric compared to using Replay alone. Moreover, MoFO works well in combination with EWC, yielding at least a 2.1% improvement in the OP metric over using EWC alone. Additionally, when combined with the GEM method, MoFO provides a 0.9% improvement on the OP metric compared to using GEM alone.

In summary, these results underscore the superior performance of MoFO in continual fine-tuning and its effectiveness in alleviating forgetting.

Table 3: The performance on the math reasoning task (GSM8K) and general capability scores of Llama-2-7B after fine-tuning on MetaMathQA using different updating strategies in MoFO. Bold values denote the best results among the BCD methods.

| Method | GSM8K | General Capability | | | |
| --- | --- | --- | --- | --- | --- |
| | | CR | MMLU | HEval | Avg. |
| Llama-2-7B | 13.7 | 65.6 | 42.0 | 24.2 | 43.9 |
| Default FT | 49.4 | 62.3 | 36.6 | 16.1 | 38.3 |
| Random BCD | 35.0 | 65.8 | 41.1 | 25.1 | 44.0 |
| Grad BCD | 40.2 | **66.0** | 41.6 | **28.0** | 45.2 |
| MV BCD | 42.2 | **66.0** | 40.0 | 27.6 | 44.5 |
| MoFO | **45.4** | 65.7 | **43.5** | 27.4 | **45.5** |

### 4.4 Impact of Update Strategy in MoFO

In addition to MoFO, we consider three other BCD methods with different filtering strategies: **random BCD**, **gradient-filtered BCD**, and **MV-filtered BCD**. **Random BCD** updates a random subset of parameters at each iteration. **Gradient-filtered BCD** replaces MoFO's filter $\texttt{FLT}_\alpha(m_t)$ with $\texttt{FLT}_\alpha(g_t)$, while **MV-filtered BCD** uses $\texttt{FLT}_\alpha(m_t/\sqrt{v_t})$.

We fine-tune Llama-2-7B on MetaMathQA using these four methods with 10% parameter update fraction and present the results in Table 3. Experimental results show that all four BCD methods exhibit significantly less forgetting compared to Default FT, demonstrating the effectiveness of BCD algorithms in mitigating forgetting.

In terms of GSM8K performance, our proposed MoFO method significantly surpasses Random BCD, Gradient-filtered BCD, and MV-filtered BCD, indicating that updating parameters with the largest momentum leads to strong optimization power. Additional comparative experiments on BCD filtering strategies are presented in Appendix F.3. More insights towards this result are provided in Appendix F.4.

### 4.5 Furthur Analysis

**Guidelines for setting $\alpha$.** Experiments show that setting the updating fraction $\alpha = 15\%$ works the best for most of our experiments; and $5\% - 15\%$ all work quite well. We provide a more detailed guideline for determining $\alpha$ in Appendix D. The guideline involves randomly sampling a small proxy subset and performing a grid search over possible $\alpha$ values.

**Efficiency Analysis.** We provide an efficiency analysis on MoFO in Appendix F.2. The results show that MoFO requires only around $4\% - 5\%$ additional training time compared with Default FT throughout the entire training process.

## 5 Related Works

### 5.1 Forgetting in Continual Learning

Catastrophic forgetting, a significant issue where models forget previously learned information upon learning new data, has received considerable attention in machine learning (McCloskey & Cohen, 1989; Goodfellow et al., 2013; Kemker et al., 2018; Ramasesh et al., 2021; Verwimp et al., 2023; Liu et al., 2024). Traditional continual learning primarily focuses on addressing catastrophic forgetting in *sequential-task learning* scenarios. In addition to investigating the forgetting of pre-training knowledge during fine-tuning, Section 4.3 conducts

experimental studies on catastrophic forgetting in sequential-task fine-tuning processes, which aligns more closely with conventional continual learning paradigms.

**Replay-based methods**. In sequential-task learning, these methods leverage past experiences to facilitate the learning of new tasks. The most classical scheme is experience replay, which involves replaying data of past tasks during incremental training (Rolnick et al., 2019) (Aljundi et al., 2019a; Hayes et al., 2019; Cha et al., 2021; Chaudhry et al., 2019b; Riemer et al., 2019b). Other variants utilize gradient information from old tasks (Lopez-Paz & Ranzato, 2017; Riemer et al., 2019a; Chaudhry et al., 2019a; Farajtabar et al., 2020; Aljundi et al., 2019b; Chaudhry et al., 2021; Tiwari et al., 2022). In LLMs, Yin et al. (2023); Wang et al. (2024b); Ouyang et al. (2022) propose replay-based methods to mitigate forgetting. While MoFO is a replay-free method, MoFO can be combined with replay strategies.

**Regularization-based methods**. These methods introduce constraints to the training process to preserve past knowledge, such as adding regularization to the loss functions (Kirkpatrick et al., 2017; Aljundi et al., 2018; Zenke et al., 2017; Li et al., 2018; Ritter et al., 2018; Kumar et al., 2023) or the embedding/output changes (Li & Hoiem, 2017; Rannen et al., 2017; Buzzega et al., 2020; Huang et al., 2021; Cha et al., 2020). Some regularization-based approaches still rely on partial information from previous models (Kirkpatrick et al., 2017). In contrast, MoFO does not require past information and does not alter the original loss function, making it inherently orthogonal to regularization-based methods. To improve generalization and robustness to noise after instruction-tuning, several studies introduce explicit regularization (Li & Zhang, 2021; Zhang et al., 2023). In particular, Zhang et al. (2023) proposes a Hessian-based penalty that encourages convergence to flatter minima. It is an interesting direction for future research to evaluate whether MoFO's momentum filtering mechanism implicitly favors more stable minima, thereby further enhancing generalization and noise robustness.

**Optimization-based methods**. These methods focus on modifying the training algorithm to mitigate forgetting. In traditional continual learning, optimization-based methods commonly include, but are not limited to, gradient projection techniques (Wang et al., 2023a; Lopez-Paz & Ranzato, 2017), meta-learning approaches (Beaulieu et al., 2020; Javed & White, 2019), and strategies leveraging the structure of the loss landscapes (Mirzadeh et al., 2020a;b). When it comes to forgetting-mitigation in LLM training, recent studies have explored optimization strategies that update only a subset of parameters at each iteration. For instance, Hui et al. (2024) randomly freezes half of the model's parameter modules and updatets the rest at each iteration. Ke et al. (2023b;a) introduce a soft-masking mechanism that selects parameters for update based on their importance values. Further, Zhang et al. (2024a) combines selective module updating with soft-masking. MoFO, which also falls into this category, updates parameters with largest momentum magnitudes at each iteration. Compared to these works, our study provides a theoretical convergence guarantee of our proposed method, thereby establishing its effectiveness in LLM fine-tuning.

**Model merging methods**. These methods balance learning new knowledge and retaining old knowledge by merging the new and past models. One line of research focuses on model averaging, which interpolates between the weights of different LLMs (Wortsman et al., 2022a;b; Eeckt et al., 2022; Yadav et al., 2024; Lin et al., 2023; 2024a). Another line of research relies on the observation that task-specific knowledge largely resides in a subspace of the weight space (Ilharco et al., 2023; Panigrahi et al., 2023; Gueta et al., 2023; Zhu et al., 2024; He et al., 2024), and leverage task vectors or task localization to preserve pre-training knowledge in the fine-tuned models (Panigrahi et al., 2023; Yadav et al., 2024; Yu et al., 2024a).

**Architecture-based methods**. These methods modify the model's architecture in training. LoRA (Hu et al., 2022), as the most popular parameter-efficient fine-tuning (PEFT) method, freezes the pre-training weights and introduces low-rank trainable matrices. Variants of LoRA are applied in continual learning for LLMs (Ren et al., 2024; Wang et al., 2023a). However, LoRA is observed to forget less but also learn less than default fine-tuning (Biderman et al., 2024). Apart from LoRA, Adapters (Houlsby et al., 2019) and BitFit (Zaken et al., 2021) are also well-known PEFT methods. In Appendix E.4, we include them as baselines for comparison and find that, while they are slightly less effective than MoFO in mitigating forgetting, their performance on the fine-tuning task is substantially worse than that of MoFO.

Other approaches adaptively expand model capacity or isolate partial weights to mitigate interference between new and old tasks (Wang et al., 2023a; Razdaibiedina et al., 2023). In contrast, MoFO updates a subset of parameters at each iteration, but does not alter the total trainable parameters.

## 5.2 Block Coordinate Descent

Block Coordinate Descent (BCD) involves iteratively optimizing over a block of coordinates while holding the others constant. The foundational work of Tseng (2001) provides a comprehensive analysis of the convergence properties of BCD under certain conditions. Subsequent research has explored various BCD variants (Hong et al., 2017), including random BCD (Nesterov, 2012; Richtárik & Takáč, 2014; Lu & Xiao, 2015), cyclic BCD (Sun & Hong, 2015; Razaviyayn et al., 2013), and greedy BCD (Nutini et al., 2015). Among these, the greedy variant, also known as Gauss-Southwell BCD method, has drawn attention due to its ability to prioritize coordinates that yield the most substantial improvement in each iteration, thereby potentially accelerating convergence.

In the realm of machine learning, BCD has also found applications (Nutini et al., 2022). For example, Luo et al. (2024) leverages BCD to perform memory-efficient fine-tuning of LLM and Xu & Zhang (2024) uses random masking to perform this. In federated learning, Rothchild et al. (2020) adopts top-$k$ momentum value unsketch rather than our top-$k$ momentum filtering to tackle communication bottleneck and convergence issues. In LLMs, some concurrent works propose BCD-based algorithms leveraging task vectors to enhance fine-tuning performance (Li et al., 2024) and mitigate catastrophic forgetting in multi-task learning (Panda et al., 2024). Our approach can be regarded as a type of greedy BCD adapted to Adam, achieving good performance in fine-tuning tasks and alleviating forgetting.

## 6 Conclusion and Limitations

This paper presents the Momentum-Filtered Optimizer (MoFO), a new approach designed to mitigate the crucial issue of pre-training knowledge forgetting in LLMs during fine-tuning. By selectively updating the parameters with the largest momentum magnitudes in each parameter block, MoFO converges to a point closer to the pre-trained model compared to full-parameter fine-tuning and effectively preserves pre-trained knowledge. Our experimental results demonstrate that MoFO not only achieves comparable performance to default fine-tuning but also effectively alleviates forgetting.

While this work provides a preliminary exploration of applying traditional block coordinate descent methods to mitigate forgetting in LLM training, several avenues remain open for further investigation. First, the current framework uses a uniform and consistent update fraction across all parameter blocks throughout training, whereas future work may explore adaptive update fractions and block-wise dynamic adjustments. Second, although our focus centers on forgetting mitigation during supervised fine-tuning, extending this methodology to downstream phases such as RLHF represents a promising direction for improving LLM development pipelines.

## Acknowledgement

The authors would like to express our sincere gratitude to the reviewers for their insightful feedback during the discussion phase. The authors also thank Congliang Chen and Ziniu Li for their helpful suggestions. This paper is supported by NSFC (No. 12326608 and No. 12401409); Hetao Shenzhen-Hong Kong Science and Technology Innovation Cooperation Zone Project (No.HZQSWS-KCCYB-2024016); University Development Fund UDF01001491, the Chinese University of Hong Kong, Shenzhen; Guangdong Provincial Key Laboratory of Mathematical Foundations for Artificial Intelligence (2023B1212010001).

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

# A    Theoretical Analysis

Appendix A is organized into three self-contained parts. A roadmap is provided to guide the reader in locating and understanding each theoretical result efficiently.

**Roadmap to Appendix A (Quick Reference).**

**Appendix A.1: Supplementary Analysis on the Top-$\alpha$ Filter**

This section formalizes the top-$\alpha$ filter $\mathrm{FLT}_\alpha(\cdot)$ and the induced quantity $\|z \odot \mathrm{FLT}_\alpha(z)\|_1$ that will serve as our working norm in the analysis. Proposition 1 verifies that this is indeed a norm. Lemma 1 establishes a *Lipschitz stability* property for filtered outputs, which later lets us pass from the momentum-defined filtering to the gradient-defined filtering. Lemma 2 relates the $L_{1,\text{top-}\alpha}$ norm to standard $L_p$ norms, which is used at the very end to translate the convergence bound to any $p \in [1, \infty]$.

**Appendix A.2: Proof of Theorem 1 (Convergence of MoFO)**

For a high-level narrative, see the proof sketch accompanying Theorem 1 in the main text; here we provide a quick-reference map to the technical steps.

Lemma 3 bounds each step size and the $\ell_2$-movement of parameters using the number of active coordinates. Lemma 4 controls the drift of gradients across iterations via $L$-smoothness. Lemma 5 lower-bounds the per-coordinate alignment term $g_{i,t}\hat{m}_{i,t}/\sqrt{\hat{v}_{i,t}}$, which is the driver in the descent inequality. Lemma 6 shows that the bias-corrected momentum $\hat{m}_t$ tracks $g_t$ in $\ell_1$ at rate $O(t^{-1/2})$. Combining these with the descent lemma yields a key inequality in which the filter is $\mathrm{FLT}_\alpha(m_t)$:

$$\frac{C_1}{\sqrt{t}}\|g_t \odot \mathrm{FLT}_\alpha(m_t)\|_1 \le \mathcal{L}(\theta_{t-1}) - \mathcal{L}(\theta_t) + \frac{C_2}{t}.$$

Finally, Lemma 1 help convert it to the desired inequality with the gradient filter $\mathrm{FLT}_\alpha(g_t)$; summing over $t$ gives $\min_{0 \le t \le T-1}\|\nabla\mathcal{L}(\theta_t) \odot \mathrm{FLT}_\alpha(\nabla\mathcal{L}(\theta_t))\|_1 = O(\log T/\sqrt{T})$. By Lemma 2, this implies the same $O(\log T/\sqrt{T})$ rate for any $L_p$ norm $\min_{0 \le t \le T-1}\|\nabla\mathcal{L}(\theta_t)\|_p$.

**Appendix A.3: Proof of Theorem 2 (Illustrative example: forgetting mitigation of MoFO)**

In an illustrative example with updating ratio $\alpha = 1/d$, MoFO converges to a single-attractor minimum and is strictly closer to the pre-training state than Adam, thus attaining a lower pre-training loss. We prove this by mathematical induction.

**Appendix A.4: Challenges and Potential Extensions of Theorem 1 to Nonsmooth Objectives**

This subsection explains why extending the convergence analysis of Theorem 1 to nonsmooth objectives is nontrivial and outlines several possible future directions.

### A.1 Supplementary Analysis on the Top-$\alpha$ Filter

In this section, we provide supplementary analysis on our top-$\alpha$ filter, which serves as a preliminary for proving Theorem 1 in Appendix A.2.

As introduced in Section 2.2, the entire parameter space is divided into $B$ parts, with the $k$-th part having a dimension of $d_k$. We assume the parameter space is $\mathbb{R}^d$, which can be expressed as the product $\mathbb{R}^d \cong \mathbb{R}^{d_1} \times \mathbb{R}^{d_2} \times \cdots \times \mathbb{R}^{d_B}$. For any $z \in \mathbb{R}^d$, we represent it as:

$$z = \text{Concat}(z^{(1)}, z^{(2)}, \ldots, z^{(B)}),$$

where $z^{(k)} \in \mathbb{R}^{d_k}$ for each $1 \leq k \leq B$.

**Definition 1.** *For any $z \in \mathbb{R}^d$, we define the top-$\alpha$ filter of $z$ as*

$$FLT_\alpha(z) := \text{Concat}(\mathbf{e}_{S_1}^{(1)}; \mathbf{e}_{S_2}^{(2)}; \ldots; \mathbf{e}_{S_B}^{(B)}) \in \mathbb{R}^d,$$

*where*

$$S_k = \{i \in [d_k] : |z_i^{(k)}| \text{ ranks within the top-}\alpha \text{ of all } |z^{(k)}|\text{'s entries } (|z_1^{(k)}|, |z_2^{(k)}|, \ldots, |z_{d_k}^{(k)}|)\}$$

*and $\mathbf{e}_{S_k}^{(k)}$ is a $d_k$-dimensional vector where the $i$-th entry is 1 if $i \in S_k$, and 0 otherwise.*

**Remark 2.** *To ensure that the top-$\alpha$ filter $FLT_\alpha(z)$ is well-defined, when multiple entries share identical absolute values and including all of them in the set $S_k$ would result in exceeding the $\alpha$ threshold of set size, the construction of $S_k$ prioritizes the entries with the smallest indices among those with the same absolute values.*

**Definition 2.** *For any $z \in \mathbb{R}^d$, we define the $L_{1,\text{top-}\alpha}$ norm of $z$ as*

$$\|z\|_{1,\text{top-}\alpha} := \|z \odot FLT_\alpha(z)\|_1.$$

**Proposition 1.** *$\|\cdot\|_{1,\text{top-}\alpha}$ is indeed a norm in $\mathbb{R}^d$.*

*Proof.* By Definition 1, we get

$$\|z\|_{1,\text{top-}\alpha} = \|z \odot \text{FLT}_\alpha(z)\|_1 = \sum_{k=1}^{B} \|z^{(k)} \odot \mathbf{e}_{S_k}^{(k)}\|_1. \tag{6}$$

First, if $\|z\|_{1,\text{top-}\alpha} = 0$, then by (6), $\|z^{(k)} \odot \mathbf{e}_{S_k}^{(k)}\|_1 = 0$ for any $1 \leq k \leq B$. Thus,

$$\|z^{(k)}\|_\infty = \underset{1 \leq i \leq d_k}{\arg\max} |z_i^{(k)}| \leq \|z^{(k)} \odot \mathbf{e}_{S_k}^{(k)}\|_1 = 0.$$

So $z^{(k)}$ is a zero vector for any $1 \leq k \leq B$ and then $z$ is a zero vector.

Second, for any given $c \in \mathbb{R}_+$, $\{|z_i^{(k)}|\}_{1 \leq i \leq d_k}$ and $\{|cz_i^{(k)}|\}_{1 \leq i \leq d_k}$ have the same order. So $z$ and $cz$ share the same filter $\text{FLT}_\alpha(z)$ and

$$\|cz\|_{1,\text{top-}\alpha} = \|cz \odot \text{FLT}_\alpha(cz)\|_1 = c\|z \odot \text{FLT}_\alpha(z)\|_1 = c\|z\|_{1,\text{top-}\alpha}.$$

Third, for any $x, y \in \mathbb{R}^d$, we let

$$S_k' = \{i \in [d_k] : |x_i^{(k)}| \text{ ranks within the top-}\alpha \text{ of all } |x^{(k)}|\text{'s entries } (|x_1^{(k)}|, |x_2^{(k)}|, \ldots, |x_{d_k}^{(k)}|)\},$$

$$S_k'' = \{i \in [d_k] : |x_i^{(k)} + y_i^{(k)}| \text{ ranks within the top-}\alpha \text{ of all}$$
$$|x^{(k)} + y^{(k)}|\text{'s entries } (|x_1^{(k)} + y_1^{(k)}|, |x_2^{(k)} + y_2^{(k)}|, \ldots, |x_{d_k}^{(k)} + y_{d_k}^{(k)}|)\}.$$

Then we have

$$\text{FLT}_\alpha(x) = \text{Concat}(\mathbf{e}_{S_1'}^{(1)}; \mathbf{e}_{S_2'}^{(2)}; \ldots; \mathbf{e}_{S_B'}^{(B)}) \quad \text{and} \quad \text{FLT}_\alpha(x + y) = \text{Concat}(\mathbf{e}_{S_1''}^{(1)}; \mathbf{e}_{S_2''}^{(2)}; \ldots; \mathbf{e}_{S_B''}^{(B)}).$$

By the construction of $S_k'$, for any $1 \leq k \leq B$, we have

$$\|x^{(k)} \odot \mathbf{e}_{S_k''}^{(k)}\|_1 \leq \|x^{(k)} \odot \mathbf{e}_{S_k'}^{(k)}\|_1.$$

So

$$\|x \odot \text{FLT}_\alpha(x+y)\|_1 = \sum_{k=1}^{B} \|x^{(k)} \odot \mathbf{e}_{S_k''}^{(k)}\|_1 \leq \sum_{k=1}^{B} \|x^{(k)} \odot \mathbf{e}_{S_k'}^{(k)}\|_1 = \|x \odot \text{FLT}_\alpha(x)\|_1.$$

Similarly, it holds that

$$\|y \odot \text{FLT}_\alpha(x+y)\|_1 \leq \|y \odot \text{FLT}_\alpha(y)\|_1.$$

Thus, we have

$$
\begin{aligned}
\|x+y\|_{1,\text{top-}\alpha} &= \|(x+y) \odot \text{FLT}_\alpha(x+y)\|_1 \\
&= \|x \odot \text{FLT}_\alpha(x+y) + y \odot \text{FLT}_\alpha(x+y)\|_1 \\
&\leq \|x \odot \text{FLT}_\alpha(x+y)\|_1 + \|y \odot \text{FLT}_\alpha(x+y)\|_1 \\
&\leq \|x \odot \text{FLT}_\alpha(x)\|_1 + \|y \odot \text{FLT}_\alpha(y)\|_1 \\
&= \|x\|_{1,\text{top-}\alpha} + \|y\|_{1,\text{top-}\alpha}.
\end{aligned}
$$

$\square$

We propose a lemma which is useful for the proof of Theorem 1.

**Lemma 1.** *For any $x, y \in \mathbb{R}^d$, it holds that*

$$\|x \odot FLT_\alpha(x)\|_1 - \|x \odot FLT_\alpha(y)\|_1 \leq 2\|x-y\|_1.$$

*Proof.* By Proposition 1, $\|\cdot\|_{1,\text{top-}\alpha}$ is a norm in $\mathbb{R}^d$, so we have

$$
\begin{aligned}
&\|x \odot \text{FLT}_\alpha(x)\|_1 - \|x \odot \text{FLT}_\alpha(y)\|_1 \\
={}& \|x \odot \text{FLT}_\alpha(x)\|_1 - \|y \odot \text{FLT}_\alpha(y)\|_1 + \|y \odot \text{FLT}_\alpha(y)\|_1 - \|x \odot \text{FLT}_\alpha(y)\|_1 \\
={}& \|x\|_{1,\text{top-}\alpha} - \|y\|_{1,\text{top-}\alpha} + \|y \odot \text{FLT}_\alpha(y)\|_1 - \|x \odot \text{FLT}_\alpha(y)\|_1 \\
\leq{}& \|x-y\|_{1,\text{top-}\alpha} + \|(y-x) \odot \text{FLT}_\alpha(y)\|_1 \\
\leq{}& \|x-y\|_1 + \|y-x\|_1 \\
={}& 2\|x-y\|_1.
\end{aligned}
$$

$\square$

The lemma below quantifies the relationship between $L_{1,\text{top-}\alpha}$ and $L_p$ norms for $p \in [1, +\infty]$.

**Lemma 2.** *Assume the parameter space $\mathbb{R}^d$ is decomposed into $B$ blocks $\mathbb{R}^{d_1} \times \cdots \times \mathbb{R}^{d_B}$ and $z = \text{Concat}(z^{(1)}, \ldots, z^{(B)})$ with $z^{(k)} \in \mathbb{R}^{d_k}$. Then for any $z \in \mathbb{R}^d$ and any $p \in [1, \infty]$, it holds that*

$$\alpha\|z\|_p \leq \|z\|_{1,top\text{-}\alpha} \leq (d\alpha + B)^{1-\frac{1}{p}}\|z\|_p.$$

*Proof.* We first prove the $p = 1$ case. Fix any block $k$. Write the absolute values in nonincreasing order $a_1^{(k)} \geq \cdots \geq a_{d_k}^{(k)} \geq 0$, where $a_i^{(k)} := |z_i^{(k)}|$. Let $m_k := |S_k(z)|$; by the definition of the top-$\alpha$ filter (Definition 1) and the tie-breaking rule in Remark 2, $m_k = \lceil \alpha d_k \rceil$. Since the average of the top $m_k$ numbers is at least the overall average, we have

$$\sum_{i \in S_k(z)} |z_i^{(k)}| \geq \frac{m_k}{d_k} \sum_{i=1}^{d_k} |z_i^{(k)}| \geq \alpha\|z^{(k)}\|_1.$$

Summing over $k$ yields $\|z\|_{1,\text{top-}\alpha} \geq \alpha \sum_k \|z^{(k)}\|_1 = \alpha\|z\|_1$. The upper bound $\|z\|_{1,\text{top-}\alpha} \leq \|z\|_1$ is immediate since $\text{FLT}_\alpha(z)$ is a $\{0, 1\}$ filter.

Next, let $p \in (1, +\infty]$. The lower bound follows from

$$\|z\|_{1,\text{top-}\alpha} \geq \alpha\|z\|_1 \geq \alpha\|z\|_p,$$

since $\|z\|_1 \geq \|z\|_p$. For the upper bound, fix $k$ and apply Hölder's inequality on the subset $S_k(z)$:

$$\sum_{i \in S_k(z)} |z_i^{(k)}| \leq \Big( \sum_{i \in S_k(z)} 1^q \Big)^{1/q} \Big( \sum_{i=1}^{d_k} |z_i^{(k)}|^p \Big)^{1/p} = m_k^{1/q}\|z^{(k)}\|_p = m_k^{1-\frac{1}{p}}\|z^{(k)}\|_p,$$

where $q = \frac{p}{p-1}$ and we used $1/q = 1 - 1/p$. Summing over $k$ and applying Hölder again to the finite sum $\sum_{k=1}^{B} m_k^{1-1/p}\|z^{(k)}\|_p$ gives

$$\|z\|_{1,\text{top-}\alpha} = \sum_{k=1}^{B} \sum_{i \in S_k(z)} |z_i^{(k)}| \leq \sum_{k=1}^{B} m_k^{1-\frac{1}{p}}\|z^{(k)}\|_p \leq \Big( \sum_{k=1}^{B} m_k \Big)^{1-\frac{1}{p}} \Big( \sum_{k=1}^{B} \|z^{(k)}\|_p^p \Big)^{1/p}.$$

We set $m := \sum_{k=1}^{B} m_k$. Since $\sum_{k=1}^{B} \|z^{(k)}\|_p^p = \|z\|_p^p$ (blocks are disjoint) and $m = \sum_k m_k = \sum_k \lceil \alpha d_k \rceil \leq \alpha d + B$, we obtain

$$\|z\|_{1,\text{top-}\alpha} \leq m^{1-\frac{1}{p}}\|z\|_p \leq (\alpha d + B)^{1-\frac{1}{p}}\|z\|_p.$$

For $p = +\infty$ the same argument applies with $q = 1$ and yields $\sum_{i \in S_k(z)} |z_i^{(k)}| \leq m_k\|z^{(k)}\|_\infty$, hence $\|z\|_{1,\text{top-}\alpha} \leq m\|z\|_\infty$; the lower bound $\|z\|_{1,\text{top-}\alpha} \geq \alpha\|z\|_\infty$ follows from $\|z\|_1 \geq \|z\|_\infty$. This completes the proof. $\square$

## A.2 Proof of Theorem 1 (Convergence of MoFO)

**Notation recap (Appendix A.2).**

- $g_t = \nabla \mathcal{L}(\theta_{t-1}) \in \mathbb{R}^d$ is the full-batch gradient at step $t$.

- Step size: $\eta_t = \eta/\sqrt{t}$; hyperparameters satisfy $\beta_1 < \sqrt{\beta_2} < 1$.

- First/second moments: $m_t, v_t \in \mathbb{R}^d$ with updates

$$(m_{i,t}, v_{i,t}) = \left((1-\beta_1)g_{i,t} + \beta_1 m_{i,t-1},\ (1-\beta_2)g_{i,t}^2 + \beta_2 v_{i,t-1}\right);$$

bias-corrected first/second moments: $\hat{m}_t = m_t/(1-\beta_1^t)$, $\hat{v}_t = v_t/(1-\beta_2^t)$.

- MoFO's filter $\mathtt{FLT}_\alpha(\cdot) \in \{0,1\}^d$: in each partition $k \in [B]$ of size $d_k$ (with $\sum_k d_k = d$), we keep $\lceil d_k \alpha \rceil$ entries and zero out the others. We write $\|z \odot \mathtt{FLT}_\alpha(z)\|_1$ for the $\ell_1$-norm of the kept coordinates and $\|x\|_{1,\text{top-}\alpha} \triangleq \|x \odot \mathtt{FLT}_\alpha(x)\|_1$.

- MoFO's update: $\theta_{i,t} - \theta_{i,t-1} = -\eta_t\, \hat{m}_{i,t}/\sqrt{\hat{v}_{i,t}}$ if $\mathtt{FLT}_\alpha(m_t)_i = 1$, and $0$ otherwise.

Our proof of Theorem 1 follows the convergence analysis of the full-batch Adam optimizer in Shi et al. (2021), with novel adaptations to address the unique aspects of MoFO.

To maintain consistency with the notation used in MoFO (Algorithm 1 in Section 2.2), we denote

$$z_t = \mathtt{Concat}(z_t^{(1)}, \ldots, z_t^{(B)}),$$

where $z$ represents the model parameter $\theta$, the gradient $g$, the first moment estimate $m$, or the second moment estimate $v$. Notably, each of these variables belongs to $\mathbb{R}^d$. Thus, for any $1 \le i \le d$, we can denote $z_{i,t}$ as the $i$-th entry of $z_t$ when $z$ represents $\theta$, $g$, $m$, or $v$.

By the update rules of the first and second moment estimates, we have

$$m_{i,t} = (1-\beta_1)g_{i,t} + \beta_1 m_{i,t-1}, \quad m_{i,0} = 0,$$
$$v_{i,t} = (1-\beta_2)g_{i,t}^2 + \beta_2 v_{i,t-1}, \quad v_{i,0} = 0.$$

So by mathematical induction, for any $1 \le i \le d$, we have

$$m_{i,t} = (1-\beta_1)\sum_{s=1}^{t} \beta_1^{t-s} g_{i,s} \tag{7}$$

and

$$v_{i,t} = (1-\beta_2)\sum_{s=1}^{t} \beta_2^{t-s} g_{i,s}^2. \tag{8}$$

We will frequently use Equation (7) and (8) in the proofs of the subsequent lemmas and theorems.

**Lemma 3.** *For the full-batch version of MoFO with hyperparameters satisfying $\beta_1 < \sqrt{\beta_2} < 1$, $\epsilon = 0$, it holds that*

$$|\theta_{i,t} - \theta_{i,t-1}| \le \frac{1}{\sqrt{1-\beta_2}(1-\beta_1/\sqrt{\beta_2})} \cdot \eta_t \cdot FLT_\alpha(m_t)_i, \quad \text{for any coordinate } 1 \le i \le d.$$

*Moreover, it holds that*

$$\|\theta_t - \theta_{t-1}\|_2 \le C\eta_t,$$

*where $C = \frac{\sqrt{d\alpha + B}}{\sqrt{1-\beta_2}(1-\beta_1/\sqrt{\beta_2})}$.*

*Proof.* When the $i$-th entry is not in our filter at iteration $t$, i.e. $\text{FLT}_\alpha(m_t)_i = 0$, we have $\theta_{i,t} = \theta_{i,t-1}$. Then

$$|\theta_{i,t} - \theta_{i,t-1}| = 0 = \frac{1}{\sqrt{1 - \beta_2}(1 - \beta_1/\sqrt{\beta_2})} \cdot \eta_t \cdot \text{FLT}_\alpha(m_t)_i.$$

When the $i$-th entry is in our filter, i.e. $\text{FLT}_\alpha(m_t)_i = 1$, by the weight updating rule of MoFO, we have $\theta_{i,t} - \theta_{i,t-1} = -\eta_t \hat{m}_{i,t}/\sqrt{\hat{v}_{i,t}}$. We first analyze $m_{i,t}$ and $v_{i,t}$.

By Equation (7) and (8), the first/second moments $m_{i,t}, v_{i,t}$ satisfy

$$|m_{i,t}| \le (1 - \beta_1) \sum_{s=1}^{t} \beta_1^{t-s} |g_{i,s}|,$$

$$v_{i,t} = (1 - \beta_2) \sum_{s=1}^{t} \beta_2^{t-s} g_{i,s}^2 \ge (1 - \beta_2) \beta_2^{t-s} g_{i,s}^2, \quad \text{for any } 1 \le s \le t.$$

So we get

$$
\begin{aligned}
|\theta_{i,t} - \theta_{i,t-1}| &= \left| -\eta_t \frac{\hat{m}_{i,t}}{\sqrt{\hat{v}_{i,t}}} \right| = \eta_t \frac{\sqrt{1 - \beta_2^t}}{1 - \beta_1^t} |m_{i,t}|/\sqrt{v_{i,t}} \\
&\le \eta_t \frac{\sqrt{1 - \beta_2^t}}{1 - \beta_1^t} \sum_{s=1}^{t} \frac{(1 - \beta_1)\beta_1^{t-s}|g_{i,s}|}{\sqrt{(1 - \beta_2)\beta_2^{t-s}}|g_{i,s}|} = \eta_t \frac{1 - \beta_1}{1 - \beta_1^t} \sqrt{\frac{1 - \beta_2^t}{1 - \beta_2}} \sum_{s=1}^{t} (\beta_1/\sqrt{\beta_2})^{t-s} \\
&\le \frac{\eta_t}{\sqrt{1 - \beta_2}} \sum_{s=0}^{t-1} (\beta_1/\sqrt{\beta_2})^s \\
&\le \frac{\eta_t}{\sqrt{1 - \beta_2}(1 - \beta_1/\sqrt{\beta_2})}.
\end{aligned}
$$

Here, the last inequality holds because of the assumption $\beta_1 < \sqrt{\beta_2} < 1$.

The parameter vector is partitioned into $B$ blocks with sizes $\{d_k\}_{k=1}^{B}$ and MoFO actually choose $\lceil d_k \alpha \rceil$ entries to update in each part $k$ of parameters. Then for any $z \in \mathbb{R}^d$, we have

$$\#\{1 \le i \le d : \text{FLT}_\alpha(z)_i = 1\} = \sum_{k=1}^{B} \lceil d_k \alpha \rceil \le \sum_{k=1}^{B} (d_k \alpha + 1) = d\alpha + B.$$

Then for the $L_2$-norm of the parameter update, we have

$$
\begin{aligned}
\|\theta_t - \theta_{t-1}\|_2 &= \left( \sum_{k=1}^{d} |\theta_{i,t} - \theta_{i,t-1}|^2 \cdot \text{FLT}_\alpha(m_t)_i \right)^{\frac{1}{2}} \\
&\le \left( \frac{\eta_t^2}{(\sqrt{1 - \beta_2}(1 - \beta_1/\sqrt{\beta_2}))^2} \cdot \#\{1 \le i \le d : \text{FLT}_\alpha(z)_i = 1\} \right)^{\frac{1}{2}} \\
&\le \frac{\sqrt{d\alpha + B}}{\sqrt{1 - \beta_2}(1 - \beta_1/\sqrt{\beta_2})} \cdot \eta_t \\
&= C\eta_t.
\end{aligned}
$$

$\square$

**Lemma 4.** *Suppose that the gradient $\nabla \mathcal{L}$ is Lipschitz continuous with constant $L$. Suppose that the full-batch version of MoFO has the hyperparameters satisfying $\beta_1 < \sqrt{\beta_2} < 1$, $\epsilon = 0$ and the learning rate schedule $\eta_t = \eta/\sqrt{t}$. For any iteration steps $t \ge s \ge 1$ and any coordinate $i$, it holds that*

$$|g_{i,t} - g_{i,s}| \le \|g_t - g_s\|_2 \le \frac{2\sqrt{2}LC\eta(t-s)}{\sqrt{t}},$$

where $C = \frac{\sqrt{d\alpha+B}}{\sqrt{1-\beta_2}(1-\beta_1/\sqrt{\beta_2})}$.

*Proof.* Fix the iterations steps $t$ and $s$ with $t \geq s \geq 1$. Since $\nabla\mathcal{L}$ has Lipschitz constant $L$, the gradient difference between the step $t$ and $s$ satisfies

$$|g_{i,t} - g_{i,s}| \leq \|g_t - g_s\|_2 = \|\nabla\mathcal{L}(\theta_{t-1}) - \nabla\mathcal{L}(\theta_{s-1})\|_2 \leq L\|\theta_{t-1} - \theta_{s-1}\|_2. \tag{9}$$

By Lemma 3, for any $t > s \geq 1$, the parameter difference satisfies

$$\begin{aligned}
\|\theta_{t-1} - \theta_{s-1}\|_2 &\leq \sum_{u=s}^{t-1} \|\theta_u - \theta_{u-1}\|_2 \leq C \sum_{u=s}^{t-1} \eta_u \\
&\leq C\eta \sum_{u=s}^{t-1} \frac{1}{\sqrt{u}} \leq C\eta \sum_{u=s}^{t-1} \frac{2}{\sqrt{u-1}+\sqrt{u}} \leq 2C\eta \sum_{u=s}^{t-1} (\sqrt{u} - \sqrt{u-1}) \\
&= 2C\eta(\sqrt{t-1} - \sqrt{s-1}) = \frac{2C\eta(t-s)}{\sqrt{t-1}+\sqrt{s-1}} \\
&\leq \frac{2C\eta(t-s)}{\sqrt{t-1}} \leq \frac{2C\eta(t-s)}{\sqrt{t/2}} \\
&= \frac{2\sqrt{2}C\eta(t-s)}{\sqrt{t}}.
\end{aligned}$$

When $t = s > 1$, it is obvious that

$$\|\theta_{t-1} - \theta_{s-1}\|_2 = 0 \leq \frac{2\sqrt{2}C\eta(t-s)}{\sqrt{t}}.$$

Combining it with (9), for any $t \geq s \geq 1$, we have

$$|g_{i,t} - g_{i,s}| \leq \|g_t - g_s\|_2 \leq \frac{2\sqrt{2}LC\eta(t-s)}{\sqrt{t}}.$$

$\square$

**Lemma 5.** *Under the assumptions in Lemma 4, for any iteration step $t \geq 1$ and any coordinate $i$, it holds that*

$$g_{i,t} \frac{\hat{m}_{i,t}}{\sqrt{\hat{v}_{i,t}}} \geq \sqrt{1-\beta_2}\left(|g_{i,t}| - \left[\frac{2\sqrt{2}\beta_1}{(1-\beta_1)^2} + \frac{4}{1-\beta_2}\right]\frac{LC\eta}{\sqrt{t}}\right).$$

*Proof.* By Lemma 4, we get

$$g_{i,t}g_{i,s} = g_{i,t}^2 - g_{i,t}(g_{i,t} - g_{i,s}) \geq g_{i,t}^2 - |g_{i,t}| \cdot |g_{i,t} - g_{i,s}| \geq g_{i,t}^2 - \frac{2\sqrt{2}LC\eta(t-s)}{\sqrt{t}}|g_{i,t}|.$$

Then for the product of gradient and momentum, we have

$$\begin{aligned}
g_{i,t}m_{i,t} &= (1-\beta_1)\sum_{s=1}^{t} \beta_1^{t-s} g_{i,t}g_{i,s} \\
&\geq g_{i,t}^2 \cdot (1-\beta_1)\sum_{s=1}^{t} \beta_1^{t-s} - \frac{2\sqrt{2}LC\eta}{\sqrt{t}}|g_{i,t}| \cdot (1-\beta_1)\sum_{s=1}^{t} \beta_1^{t-s} \cdot (t-s) \tag{10} \\
&\geq g_{i,t}^2 \cdot (1-\beta_1)\sum_{s=0}^{t-1} \beta_1^{s} - \frac{2\sqrt{2}LC\eta}{\sqrt{t}}|g_{i,t}| \cdot (1-\beta_1)\sum_{s=1}^{t-1} s\beta_1^{s}.
\end{aligned}$$

Since we have

$$\sum_{s=0}^{t-1} \beta_1^s = \frac{1-\beta_1^t}{1-\beta_1}, \quad \sum_{s=1}^{t-1} s\beta_1^{s-1} \leq \sum_{s=1}^{\infty} s\beta_1^{s-1} = \frac{d}{d\beta_1}\left(\sum_{s=1}^{\infty} \beta_1^s\right) = \frac{d}{d\beta_1}\left(\frac{\beta_1}{1-\beta_1}\right) = \frac{1}{(1-\beta_1)^2}, \quad (11)$$

it holds that

$$g_{i,t}m_{i,t} \geq \text{RHS of (10)} \geq (1-\beta_1^t)g_{i,t}^2 - \frac{2\sqrt{2}\beta_1 LC\eta}{(1-\beta_1)\sqrt{t}}|g_{i,t}|. \quad (12)$$

For the second moment $v_{i,t}$, we have

$$
\begin{aligned}
v_{i,t} &= (1-\beta_2)\sum_{s=1}^{t} \beta_2^{t-s} g_{i,s}^2 \leq (1-\beta_2)\sum_{s=1}^{t} \beta_2^{t-s}(|g_{i,t}| + |g_{i,s} - g_{i,t}|)^2 \\
&\leq (1-\beta_2)\sum_{s=1}^{t} \beta_2^{t-s}\left(|g_{i,t}| + \frac{2\sqrt{2}LC\eta(t-s)}{\sqrt{t}}\right)^2 = (1-\beta_2)\sum_{s=0}^{t-1} \beta_2^s\left(|g_{i,t}| + \frac{2\sqrt{2}LC\eta s}{\sqrt{t}}\right)^2 \\
&= |g_{i,t}|^2 \cdot (1-\beta_2)\left(\sum_{s=0}^{t-1} \beta_2^s\right) + |g_{i,t}| \cdot \frac{4\sqrt{2}LC\eta}{\sqrt{t}}(1-\beta_2)\left(\sum_{s=1}^{t-1} s\beta_2^s\right) \\
&\quad + \frac{8L^2C^2\eta^2}{t}(1-\beta_2)\left(\sum_{s=1}^{t-1} s^2\beta_2^s\right).
\end{aligned}
\quad (13)
$$

Since we have

$$
\begin{aligned}
\sum_{s=0}^{t-1} \beta_2^s &= \frac{1-\beta_2^t}{1-\beta_2} \leq \frac{1}{1-\beta_2}, \\
\sum_{s=0}^{t-1} s\beta_2^{s-1} &\leq \sum_{s=0}^{\infty} s\beta_2^{s-1} = \frac{d}{d\beta_2}\left(\sum_{s=0}^{\infty} \beta_2^s\right) = \frac{d}{d\beta_2}\left(\frac{1}{1-\beta_2}\right) = \frac{1}{(1-\beta_2)^2}, \\
\sum_{s=0}^{t-1} s^2\beta_2^{s-1} &\leq \sum_{s=0}^{\infty} s^2\beta_2^{s-1} = \beta_2\left(\sum_{s=0}^{\infty} s(s-1)\beta_2^{s-2}\right) + \sum_{s=0}^{\infty} s\beta_2^{s-1} \\
&= \beta_2 \cdot \frac{d^2}{d\beta_2^2}\left(\sum_{s=0}^{\infty} \beta_2^s\right) + \frac{1}{(1-\beta_2)^2} = \beta_2 \cdot \frac{d^2}{d\beta_2^2}\left(\frac{1}{1-\beta_2}\right) + \frac{1}{(1-\beta_2)^2} \\
&= \frac{2\beta_2}{(1-\beta_2)^3} + \frac{1}{(1-\beta_2)^2} \\
&= \frac{1+\beta_2}{(1-\beta_2)^3},
\end{aligned}
$$

it holds that

$$
\begin{aligned}
v_{i,t} \leq \text{RHS of (13)} &\leq |g_{i,t}|^2 + |g_{i,t}| \cdot \frac{4\sqrt{2}\beta_2 LC\eta}{(1-\beta_2)\sqrt{t}} + \frac{8(1+\beta_2)\beta_2 L^2C^2\eta^2}{(1-\beta_2)^2 t} \\
&\leq |g_{i,t}|^2 + |g_{i,t}| \cdot \frac{8LC\eta}{(1-\beta_2)\sqrt{t}} + \frac{16L^2C^2\eta^2}{(1-\beta_2)^2 t} \\
&= \left(|g_{i,t}| + \frac{4LC\eta}{(1-\beta_2)\sqrt{t}}\right)^2.
\end{aligned}
$$

Thus, we get

$$\sqrt{v_{i,t}} \leq |g_{i,t}| + \frac{4LC\eta}{(1-\beta_2)\sqrt{t}}.$$

Recalling (12), we have

$$
\begin{aligned}
g_{i,t}m_{i,t} &\geq (1-\beta_1^t)\left(|g_{i,t}| + \frac{4LC\eta}{(1-\beta_2)\sqrt{t}}\right)\left(|g_{i,t}| - \frac{2\sqrt{2}\beta_1 LC\eta}{(1-\beta_1^t)(1-\beta_1)\sqrt{t}} - \frac{4LC\eta}{(1-\beta_2)\sqrt{t}}\right) \\
&\quad + (1-\beta_1^t)\cdot\frac{4LC\eta}{(1-\beta_2)\sqrt{t}}\left(\frac{2\sqrt{2}\beta_1 LC\eta}{(1-\beta_1^t)(1-\beta_1)\sqrt{t}} + \frac{4LC\eta}{(1-\beta_2)\sqrt{t}}\right) \\
&\geq (1-\beta_1^t)\left(|g_{i,t}| + \frac{4LC\eta}{(1-\beta_2)\sqrt{t}}\right)\left(|g_{i,t}| - \frac{2\sqrt{2}\beta_1 LC\eta}{(1-\beta_1^t)(1-\beta_1)\sqrt{t}} - \frac{4LC\eta}{(1-\beta_2)\sqrt{t}}\right) \\
&\geq (1-\beta_1^t)\sqrt{v_{i,t}}\left(|g_{i,t}| - \frac{2\sqrt{2}\beta_1 LC\eta}{(1-\beta_1^t)(1-\beta_1)\sqrt{t}} - \frac{4LC\eta}{(1-\beta_2)\sqrt{t}}\right).
\end{aligned}
$$

Therefore, for the bias-corrected first/second moments $\hat{m}_{i,t}$ and $\hat{v}_{i,t}$, it holds that

$$
\begin{aligned}
g_{i,t}\frac{\hat{m}_{i,t}}{\sqrt{\hat{v}_{i,t}}} = \frac{\sqrt{1-\beta_2^t}}{1-\beta_1^t}g_{i,t}\frac{m_{i,t}}{\sqrt{v_{i,t}}} &\geq \sqrt{1-\beta_2^t}\left(|g_{i,t}| - \frac{2\sqrt{2}\beta_1 LC\eta}{(1-\beta_1^t)(1-\beta_1)\sqrt{t}} - \frac{4LC\eta}{(1-\beta_2)\sqrt{t}}\right) \\
&\geq \sqrt{1-\beta_2}\left(|g_{i,t}| - \left[\frac{2\sqrt{2}\beta_1}{(1-\beta_1)^2} + \frac{4}{1-\beta_2}\right]\frac{LC\eta}{\sqrt{t}}\right).
\end{aligned}
$$

$\square$

**Lemma 6.** *Under the assumptions in Lemma 4, for any iteration step $t \geq 1$ and any coordinate $i$, it holds that*

$$
\|\hat{m}_t - g_t\|_1 \leq \frac{2\sqrt{2}\beta_1\sqrt{d}LC\eta}{(1-\beta_1)^2\sqrt{t}}.
$$

*Proof.* Recalling the calculation of the momentum $m_t$ in (7), we get

$$
m_t = (1-\beta_1)\sum_{s=1}^{t}\beta_1^{t-s}g_s,
$$

and

$$
m_t - (1-\beta_1^t)g_t = (1-\beta_1)\sum_{s=1}^{t}\beta_1^{t-s}(g_t - g_s).
$$

By Lemma 4 and Equation (11) in the proof of Lemma 5, we get

$$
\begin{aligned}
\|\hat{m}_t - g_t\|_2 = \left\|\frac{m_t}{1-\beta_1^t} - g_t\right\|_2 &\leq \frac{1-\beta_1}{1-\beta_1^t}\sum_{s=1}^{t}\beta_1^{t-s}\|g_t - g_s\|_2 \leq \sum_{s=1}^{t}\beta_1^{t-s}\|g_t - g_s\|_2 \\
&\leq \frac{2\sqrt{2}LC\eta}{\sqrt{t}}\sum_{s=1}^{t}\beta_1^{t-s}(t-s) = \frac{2\sqrt{2}LC\eta}{\sqrt{t}}\sum_{s=0}^{t-1}s\beta_1^s \\
&\leq \frac{2\sqrt{2}\beta_1 LC\eta}{(1-\beta_1)^2\sqrt{t}}.
\end{aligned}
$$

By Cauchy-Schwarz's inequality, we have

$$
\|\hat{m}_t - g_t\|_1 \leq \sqrt{d}\|\hat{m}_t - g_t\|_2 \leq \frac{2\sqrt{2}\beta_1\sqrt{d}LC\eta}{(1-\beta_1)^2\sqrt{t}}.
$$

$\square$

Now we will complete the proof of Theorem 1.

*Proof of Theorem 1.* By the descent lemma, since $\nabla\mathcal{L}$ is Lipschitz with constant $L$, we have

$$
\begin{aligned}
\mathcal{L}(\theta_t) - \mathcal{L}(\theta_{t-1}) &\leq \nabla\mathcal{L}(\theta_{t-1})^\top (\theta_t - \theta_{t-1}) + \frac{L}{2}\|\theta_t - \theta_{t-1}\|_2^2 \\
&\leq g_t^\top (\theta_t - \theta_{t-1}) + \frac{L}{2}\|\theta_t - \theta_{t-1}\|_2^2.
\end{aligned}
\tag{14}
$$

By Lemma 3 and Lemma 5, we have

$$
\begin{aligned}
\mathcal{L}(\theta_t) - \mathcal{L}(\theta_{t-1}) &\leq \text{RHS of (14)} \leq -\eta_t \left( \sum_{i=1}^{d} g_{i,t} \frac{\hat{m}_{i,t}}{\sqrt{\hat{v}_{i,t}}} \cdot \mathtt{FLT}_\alpha(m_t)_i \right) + \frac{LC^2\eta_t^2}{2} \\
&\leq \frac{LC^2\eta^2}{2t} - \frac{\eta}{\sqrt{t}} \sum_{i=1}^{d} \sqrt{1-\beta_2} \left( |g_{i,t}| - \left[ \frac{2\sqrt{2}\beta_1}{(1-\beta_1)^2} + \frac{4}{1-\beta_2} \right] \frac{LC\eta}{\sqrt{t}} \right) \cdot \mathtt{FLT}_\alpha(m_t)_i \\
&= -\frac{\sqrt{1-\beta_2}\cdot\eta}{\sqrt{t}}\|g_t \odot \mathtt{FLT}_\alpha(m_t)\|_1 + \left[ \frac{2\sqrt{2}\beta_1\sqrt{1-\beta_2}}{(1-\beta_1)^2} + \frac{4}{\sqrt{1-\beta_2}} + \frac{C}{2} \right] \frac{LC\eta^2}{t} \cdot \|\mathtt{FLT}_\alpha(m_t)\|_1 \\
&\leq -\frac{\sqrt{1-\beta_2}\cdot\eta}{\sqrt{t}}\|g_t \odot \mathtt{FLT}_\alpha(m_t)\|_1 + \left[ \frac{2\sqrt{2}\beta_1\sqrt{1-\beta_2}}{(1-\beta_1)^2} + \frac{4}{\sqrt{1-\beta_2}} + \frac{C}{2} \right] \frac{LC\eta^2(d\alpha + B)}{t}.
\end{aligned}
\tag{15}
$$

By Lemma 1 and Lemma 6, we have

$$
\begin{aligned}
\|g_t \odot \mathtt{FLT}_\alpha(g_t)\|_1 - \|g_t \odot \mathtt{FLT}_\alpha(m_t)\|_1 &= \|g_t \odot \mathtt{FLT}_\alpha(g_t)\|_1 - \left\| g_t \odot \mathtt{FLT}_\alpha\left( \frac{m_t}{1-\beta_1^t} \right) \right\|_1 \\
&= \|g_t \odot \mathtt{FLT}_\alpha(\hat{m}_t)\|_1 \\
&\leq 2\|g_t - \hat{m}_t\|_1 \\
&\leq \frac{4\sqrt{2}\beta_1\sqrt{d}LC\eta}{(1-\beta_2)^2\sqrt{t}}.
\end{aligned}
$$

Thus,

$$
\begin{aligned}
\mathcal{L}(\theta_t) - \mathcal{L}(\theta_{t-1}) &\leq \text{RHS of (15)} \\
&\leq -\frac{\sqrt{1-\beta_2}\cdot\eta}{\sqrt{t}}\|g_t \odot \mathtt{FLT}_\alpha(g_t)\|_1 + \left[ \frac{2\sqrt{2}\beta_1\sqrt{1-\beta_2}}{(1-\beta_1)^2} + \frac{4}{\sqrt{1-\beta_2}} + \frac{C}{2} \right] \frac{LC\eta^2(d\alpha + B)}{t} \\
&\qquad + \frac{4\sqrt{2}\beta_1\sqrt{d}LC\eta^2}{(1-\beta_2)^{\frac{3}{2}}t} \\
&= -\frac{C_1}{\sqrt{t}}\|g_t\|_{1,\text{top-}\alpha} + \frac{C_2}{t} \leq -\frac{C_1}{\sqrt{t}}\min_{1\leq t\leq T}\|g_t\|_{1,\text{top-}\alpha} + \frac{C_2}{t},
\end{aligned}
\tag{16}
$$

where

$$
\begin{aligned}
C_1 &= \sqrt{1-\beta_2}\cdot\eta, \\
C_2 &= LC\eta^2 \cdot \left\{ \left[ \frac{2\sqrt{2}\beta_1\sqrt{1-\beta_2}}{(1-\beta_1)^2} + \frac{4}{\sqrt{1-\beta_2}} + \frac{C}{2} \right] (d\alpha + B) + \frac{4\sqrt{2}\beta_1\sqrt{d}}{(1-\beta_2)^{\frac{3}{2}}} \right\}.
\end{aligned}
$$

Taking the summation of (15) from 1 to $T$, we get

$$
\begin{aligned}
\mathcal{L}^* - \mathcal{L}(\theta_0) &\leq \mathcal{L}(\theta_T) - \mathcal{L}(\theta_0) = \sum_{t=1}^{T} \mathcal{L}(\theta_t) - \mathcal{L}(\theta_{t-1}) \\
&\leq -C_1 \left( \sum_{t=1}^{T} \frac{1}{\sqrt{t}} \right) \cdot \min_{1\leq t\leq T}\|g_t \odot \mathtt{FLT}_\alpha(g_t)\|_1 + C_2 \sum_{t=1}^{T} \frac{1}{t}.
\end{aligned}
$$

Since

$$\sum_{t=1}^{T} \frac{1}{\sqrt{t}} \geq \sum_{t=1}^{T} \frac{2}{\sqrt{t} + \sqrt{t+1}} = \sum_{t=1}^{T} 2(\sqrt{t+1} - \sqrt{t}) = 2(\sqrt{T+1} - 1),$$

$$\sum_{t=1}^{T} \frac{1}{t} = 1 + \sum_{t=1}^{T-1} \frac{1}{t+1} \leq 1 + \sum_{t=1}^{T-1} \int_{t}^{t+1} \frac{1}{u} \, du \leq 1 + \int_{1}^{T} \frac{1}{u} \, du = 1 + \log T,$$

we get

$$\min_{0 \leq t \leq T-1} \|\nabla \mathcal{L}(\theta_t)\|_{1,\text{top-}\alpha} = \min_{1 \leq t \leq T} \|g_t\|_{1,\text{top-}\alpha} = \min_{1 \leq t \leq T} \|g_t \odot \text{FLT}_\alpha(g_t)\|_1$$

$$\leq \frac{\mathcal{L}(\theta_0) - \mathcal{L}^* + C_2 \sum_{t=1}^{T} \frac{1}{t}}{C_1 \sum_{t=1}^{T} \frac{1}{\sqrt{t}}}$$

$$\leq \frac{\mathcal{L}(\theta_0) - \mathcal{L}^* + C_2(1 + \log T)}{2C_1(\sqrt{T+1} - 1)}.$$

Thus, we have

$$\min_{0 \leq t \leq T-1} \|\nabla \mathcal{L}(\theta_t)\|_{1,\text{top-}\alpha} = \min_{1 \leq t \leq T} \|\mathcal{L}(\theta_t) \odot \text{FLT}_\alpha(\mathcal{L}(\theta_t))\|_1 = \mathcal{O}\left(\frac{\log T}{\sqrt{T}}\right).$$

By the relationship between $L_{1,\text{top-}\alpha}$ norm and $L_p$ norm in Lemma 2, for any $p \in [1, +\infty]$,

$$\min_{0 \leq t \leq T-1} \|\nabla \mathcal{L}(\theta_t)\|_p \leq \frac{1}{\alpha} \min_{0 \leq t \leq T-1} \|\nabla \mathcal{L}(\theta_t)\|_{1,\text{top-}\alpha}.$$

Therefore,

$$\min_{0 \leq t \leq T-1} \|\nabla \mathcal{L}(\theta_t)\|_p = \mathcal{O}\left(\frac{\log T}{\sqrt{T}}\right).$$

$\square$

### A.3 Proof of Theorem 2 (Illustrative example: forgetting mitigation of MoFO)

*Proof of Theorem 2.* Let the set $U := \{\theta \in \mathbb{R}^d : \theta_i < b_i/a_i, \ \forall 1 \le i \le d\}$. We note that:

1. The boundary of $U$ is the subset of $S = \cup_{i=1}^d S_i$, which is the collection all global minima of the fine-tuning loss.

2. The pre-training state $\theta_{\text{pretrain}} = (0, 0, \ldots, 0)$, which is also the starting point of fine-tuning, lies in $U$.

We may as well assume that with proper learning rates, the parameter $\theta$ remains within $U$ during training, unless it converges to a minimum on the boundary. If it goes across the boundary at a certain iteration before converging, the learning rate can be adjusted to ensure that it remains within $U$. For any $\theta \in U$ and coordinate $i \in \{1, 2, \ldots, d\}$, we have

$$\frac{\partial \mathcal{L}}{\partial \theta_i} = 2a_i(a_i\theta_i - b_i)\prod_{j \neq i}(a_j\theta_j - b_j)^2 = \frac{2\mathcal{L}(\theta)}{\theta_i - \frac{b_i}{a_i}} < 0. \tag{17}$$

For clarity of definition, we let $\theta_t$, represent the parameter at iteration $t$. We let $\theta_{i,t}$, $g_{i,t}$, $m_{i,t}$ denote the $i$-th coordinate of $\theta_t$, $g_t$, $m_t$ at iteration $t$, respectively.

**Analysis of MoFO.** We will use mathematical induction to show the following results: *If MoFO selects the coordinate $i_0$ at the first iteration step, it will always select $i_0$ at any iteration $t$. Moreover, for any coordinate $i \neq i_0$, we have*

- $m_{i_0,1} \le m_{i,1} \le 0$, *and* $m_{i_0,t} < m_{i,t} \le 0$ *for any iteration step* $t \ge 2$.

- $0 < \frac{b_{i_0}}{a_{i_0}} - \theta_{i_0,t} < \frac{b_i}{a_i} - \theta_{i,t}$ *for any iteration step* $t \ge 1$ *if the algorithm has not reached the minimum.*

**Base case (1st iteration step).** At the first iteration step, the momentum $m_1 = (1 - \beta_1)g_1$. So for any $1 \le i \le d$, we get

$$m_{1,t} = (1 - \beta_1)g_{1,t} = (1 - \beta_1)\left.\frac{\partial \mathcal{L}}{\partial \theta_i}\right|_{\theta_{\text{pretrain}}} < 0.$$

According to the momentum filtering mechanism of MoFO, we have

$$i_0 \in \arg\max_{1 \le i \le d} |m_{i,t}| = \arg\max_{1 \le i \le d} |g_{i,t}| = \arg\max_{1 \le i \le d} \left|\frac{\partial \mathcal{L}}{\partial \theta_i}\right| = \arg\max_{1 \le i \le d} \frac{1}{\frac{b_i}{a_i} - \theta_{i,\text{pretrain}}} = \arg\min_{1 \le i \le d} \left\{\frac{b_i}{a_i}\right\}.$$

Obviously, we get $m_{i_0,1} \le m_{i,1} \le 0$ for any coordinate $i \neq i_0$.

The parameter updates at the first iteration are:

$$\theta_{i_0,1} = \theta_{i_0,0} - \frac{\eta_1\sqrt{1 - \beta_2}m_{i_0,1}}{(1 - \beta_1)\sqrt{v_{i_0,t}}} = \theta_{i_0,0} - \frac{\eta_1 g_{i,0}}{|g_{i,0}|} = \theta_{i_0,0} - \eta_1\text{sign}\left(\frac{\partial \mathcal{L}}{\partial \theta_i}(\theta_{\text{pretrain}})\right) > \theta_{i_0,0},$$

$$\theta_{i,1} = \theta_{i,0}, \quad \forall i \neq i_0.$$

If the algorithm has not converged at the first iteration, then we have $\theta_{i_0,1} < b_{i_0}/a_{i_0}$. Moreover, for any $i \neq i_0$,

$$0 < \frac{b_i}{a_i} - \theta_{i_0,1} < \frac{b_i}{a_i} = \frac{b_i}{a_i} - \theta_{i_0,\text{pretrain}} = \frac{b_i}{a_i} - \theta_{i,\text{pretrain}} = \frac{b_i}{a_i} - \theta_{i,1}.$$

**Induction step.** Suppose that the induction hypothesis holds up to iteration $t$. Then, for any coordinate $i \neq i_0$, we have

- $m_{i_0,t} \le m_{i,t} \le 0$.

- $0 \leq \frac{b_{i_0}}{a_{i_0}} - \theta_{i_0,t} < \frac{b_i}{a_i} - \theta_{i,t}$.

So for the gradient,

$$g_{i_0,t+1} = \frac{2\mathcal{L}(\theta_t)}{\theta_{i_0,t} - \frac{b_i}{a_i}} < \frac{2\mathcal{L}(\theta_t)}{\theta_{i_0,t} - \frac{b_i}{a_i}} = g_{i,t+1} < 0,$$

and

$$m_{i_0,t+1} = \beta_1 m_{i_0,t} + (1 - \beta_1)g_{i_0,t+1}$$
$$< \beta_1 m_{i,t} + (1 - \beta_1)g_{i,t+1} = m_{i_0,t+1} < 0.$$

Thus, $i_0$ is the only coordinate in $\arg\max_{1 \leq i \leq d} |m_{i,t+1}|$ and MoFO still chooses the coordinate $i_0$ to update. In addition,

$$\theta_{i_0,t+1} = \theta_{i_0,t} - \frac{\eta_{t+1}\hat{m}_{i,t+1}}{\sqrt{\hat{v}_{i,t+1}}} > \theta_{i,t},$$

$$\theta_{i,t+1} = \theta_{i,t}, \quad \forall i \neq i_0.$$

If the algorithm has not converged at iteration step $t+1$, then we have $\theta_{i_0,t+1} < b_{i_0}/a_{i_0}$. Moreover, for any $i \neq i_0$,

$$0 < \frac{b_i}{a_i} - \theta_{i_0,t+1} < \frac{b_i}{a_i} - \theta_{i_0,t} \leq \frac{b_i}{a_i} - \theta_{i,t} = \frac{b_i}{a_i} - \theta_{i,t+1}.$$

**Conclusion.** MoFO consistently updates $\theta_{i_0}$ and eventually converges to $\theta^*_{\text{MoFO}} = (0, \ldots, 0, \frac{b_{i_0}}{a_{i_0}}, 0, \ldots, 0)$, with pre-training loss

$$\mathcal{L}_{\text{pretrain}}(\theta^*_{\text{MoFO}}) = \frac{b_{i_0}^2}{2a_{i_0}^2}.$$

**Analysis of Adam.** Unlike MoFO, Adam updates all the parameters. By Inequality (17), we have $g_{i,t} < 0$. By the momentum update rule of Adam:

$$m_{i,t+1} = \beta_1 m_{i,t} + (1 - \beta_1)g_{i,t+1},$$

we get that $m_{i,t} < 0$ for any $1 \leq i \leq d$ and any iteration $t$. Therefore, it holds for Adam that

$$\theta_{i,t+1} = \theta_{i,t} - \frac{\eta_t \mathcal{L}(\theta_t)}{\theta_{i,t} - \frac{b_i}{a_i}} > \theta_{i,t}.$$

Assuming that Adam converges to $\theta^*_{\text{Adam}}$, we have

- $\theta^*_{\text{Adam},i} > 0$ for any $1 \leq i \leq d$,

- There exists $j_0$ such that $\theta^*_{\text{GD},j_0} = b_{j_0}/a_{j_0}$.

Recall that at iteration 1, MoFO selects

$$i_0 \in \arg\min_{1 \leq i \leq d} \left\{ \frac{b_i}{a_i} \right\}.$$

Thus, the pre-training loss for Adam is

$$\mathcal{L}_{\text{pretrain}}(\theta^*_{\text{Adam}}) = \frac{b_{j_0}^2}{2a_{j_0}^2} + \sum_{i \neq j_0} \theta^{*2}_{\text{Adam},i} > \frac{b_{j_0}^2}{2a_{j_0}^2} \geq \frac{b_{i_0}^2}{2a_{i_0}^2} = \mathcal{L}_{\text{pretrain}}(\theta^*_{\text{MoFO}}).$$

In other words,

$$\|\theta^*_{\text{MoFO}} - \theta_{\text{pretrain}}\|_2^2 = 2\mathcal{L}_{\text{pretrain}}(\theta^*_{\text{MoFO}}) < 2\mathcal{L}_{\text{pretrain}}(\theta^*_{\text{Adam}}) = \|\theta^*_{\text{Adam}} - \theta_{\text{pretrain}}\|_2^2.$$

In conclusion, MoFO converges to a minimum closer to the pre-training state than Adam, preserving a lower pre-training loss. $\qquad \square$

### A.4 Challenges and Potential Extensions of Theorem 1 to Nonsmooth Objectives

This subsection outlines challenges and possible future directions for extending our convergence analysis to nonsmooth objectives; a complete extension is left for future work.

**Why the extension is challenging.**

Our Theorem 1 relies on the standard $L$-smoothness assumption—i.e., the gradient of $\mathcal{L}$ is $L$-Lipschitz—to invoke the descent lemma and to derive an upper bound on $\min_{0 \leq t \leq T-1} \|\nabla \mathcal{L}(\theta_t)\|$. When $\mathcal{L}$ is nonsmooth,

(i) $\nabla \mathcal{L}(\theta)$ may not exist at nondifferentiable points, which requires working with generalized gradients (e.g., subgradients) rather than classical gradients in the derivation.

(ii) The upper bound on $\min_{0 \leq t \leq T-1} \|\nabla \mathcal{L}(\theta_t)\|$ in the proof of Theorem 1 (Appendix A.2) scales with the smoothness constant $L$; when $L$ is unbounded (or effectively very large), these inequalities become non-informative.

**Possible extensions.**

We outline below several plausible directions; full development is left for future work.

(i) **Subgradient analysis.** Replace classical gradients with subgradients and assess convergence via a subgradient-based stationarity criterion.

(ii) **Smoothing.** Introduce a family of smoothed surrogates $\mathcal{L}_\mu$, analyze the MoFO algorithm under these surrogates to obtain $\mu$-dependent bounds, and then let $\mu \downarrow 0$ to recover convergence results for the original loss $\mathcal{L}$.

(iii) **Algorithmic modifications.** Incorporate techniques such as gradient clipping or stochastic subgradient steps (Xiao et al., 2024), and then analyze the convergence of the modified algorithm.

# B    Supplemental Figures and Explanations

## B.1    Reason for using RedPajama to approximate LLaMA-2's training data

We note that original LLaMA-2 training dataset has not been publicly released. Thus, we can only rely on public datasets to approximate LLaMA-2's original training data.

RedPajama project was explicitly designed as an open-source reproduction of the LLaMA training dataset (Weber et al., 2024; Together, 2023). It closely mirrors the data sources outlined in the original LLaMA paper and adopts similar strategies for data collection, mixture, and preprocessing. We believe it serves as a reasonable proxy for approximating LLaMA-2's training dataset.

## B.2    Supplementary Figures for Figure 1

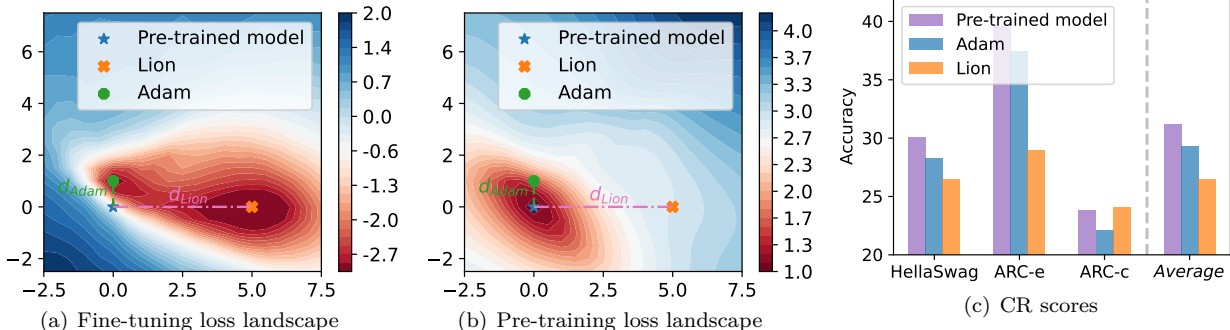

(a) Fine-tuning loss landscape       (b) Pre-training loss landscape       (c) CR scores

Figure 7: The loss landscapes of Pythia-160M after fine-tuning on a subset of the FLAN dataset using Adam and Lion. We plot the loss landscapes on (a) the fine-tuning dataset and (b) the pre-training dataset (Pile dataset (Gao et al., 2020)) and (c) the accuracies on CR tasks, including HellaSwag, ARC-c, and ARC-e. We visualize a 2D weight-space plane spanned by the vector from the pre-trained model to the Lion-tuned model (x-axis) and to the Adam-tuned model (y-axis). Axes are normalized so that one unit equals the length of the pre-trained→Adam vector. The color bar indicates the loss value—(a) fine-tuning loss and (b) pre-training loss. A logarithmic scale is applied to the loss values for better visualization. Two training methods converge to different minima with similar fine-tuning loss. Lion converges to a farther minimum from the pre-trained model and performs more forgetting than Adam.

## B.3    Supplementary Experiments on the Correlation between Distance and Forgetting

In this subsection, we augment Figure 2(b) by probing the relationship between a model's parameter distance from its pre-trained state and evaluation accuracy under additional training budgets and optimizers. Concretely, under the same settings as Figure 2(b), we add runs at 0.1 and 0.2 epochs and extend training beyond 3 epochs, using Adam and MoFO. We report results on MMLU (as in the main text, measuring preservation of factual knowledge) and newly include HumanEval (measuring preservation of code-generation ability). Since the fine-tuning task is math, both can serve as forgetting mitigation metrics. The corresponding scatter plots are shown in Figure 8 (MMLU) and Figure 9 (HumanEval). When examining the points **for each optimizer separately**, we make the following observations:

- **Observation 1 (sufficient training).** Once training exceeds approximately 1 epoch, MMLU and HumanEval scores show a consistent strong negative correlation with the parameter distance to the pre-trained state.

- **Observation 2 (early training).** For Adam or MoFO at less than 1 epoch, we may observe a mild correlation with the parameter distance to the pre-trained state. The correlation may be unstable and can be positive or negative.

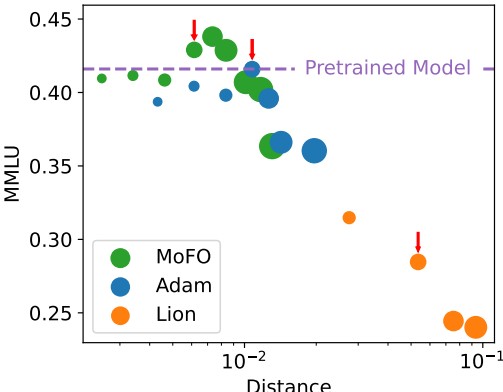

Figure 8: Average accuracy on the MMLU benchmark (measuring preservation of factual knowledge) for Llama-2-7B after fine-tuning on MetaMathQA with Adam, Lion, and MoFO. Building on Figure 2(b), we add points for runs where Llama-2-7B was trained for 0.1, 0.2, and >3 epochs using both Adam and MoFO. The marker size encodes the number of training epochs (larger means more epochs). Red arrows indicate the points obtained after exactly 1 epoch for each optimizer.

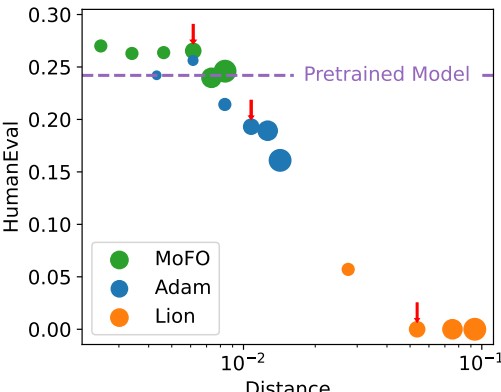

Figure 9: Scores on HumanEval benchmark for Llama-2-7B after fine-tuning on MetaMathQA with Adam, Lion, and MoFO. Building on Figure 2(b), we add points for runs where Llama-2-7B was trained for 0.1, 0.2, and >3 epochs using both Adam and MoFO. The marker size encodes the number of training epochs (larger means more epochs). Red arrows indicate the points obtained after exactly 1 epoch for each optimizer.

We speculate that this short-lived positive trend may be related to benchmark alignment. The MMLU benchmark (used to measure the preservation of factual knowledge) might share partial overlap with the patterns of our math fine-tuning task (measured by GSM8K benchmark); in the early training steps, the model might incidentally acquire features that also benefit MMLU, resulting in a temporary gain. By contrast, as for HumanEval in Figure 9, which measures code generation and differs from our math fine-tuning in both domain and output format, it may exhibit an unstable correlation whose sign can be either negative or positive. Another possible factor could stochasticity, since at less than 1 epoch the dataset has not yet been fully traversed.

Overall, the negative correlation becomes clear and consistent after sufficient training; while the early training presents a mild, benchmark-dependent relationship.

In addition, we emphasize an empirical point: the negative relationship between parameter distance and the preservation of pre-trained knowledge is evident **across optimizers**. In Figure 2(b), 8, and 9, the parameter distances roughly follow the ordering **Lion > Adam > MoFO**, whereas the scores measuring preservation of pre-trained knowledge follow the inverse ordering **MoFO > Adam > Lion**. For a broader comparison,

we evaluate five optimizers—MoFO, NAdam (Dozat, 2016), Adam, RMSProp (Tieleman, 2012), and Lion. After two epochs of training, we report (i) their parameter distance to the pre-trained state and (ii) their forgetting-mitigation performance on MMLU and HumanEval, shown in Figure 10(a) and Figure 10(b), respectively. The results show a consistent negative correlation between distance and performance. Notably, MoFO remains closer to the pre-trained state and achieves higher scores compared with the other optimizers. **Therefore, this cross-optimizer rank-order correlation is sufficient to motivate our algorithm design**: favor optimizers that converge closer to the pre-trained state, so as to better preserve the pre-trained knowledge.

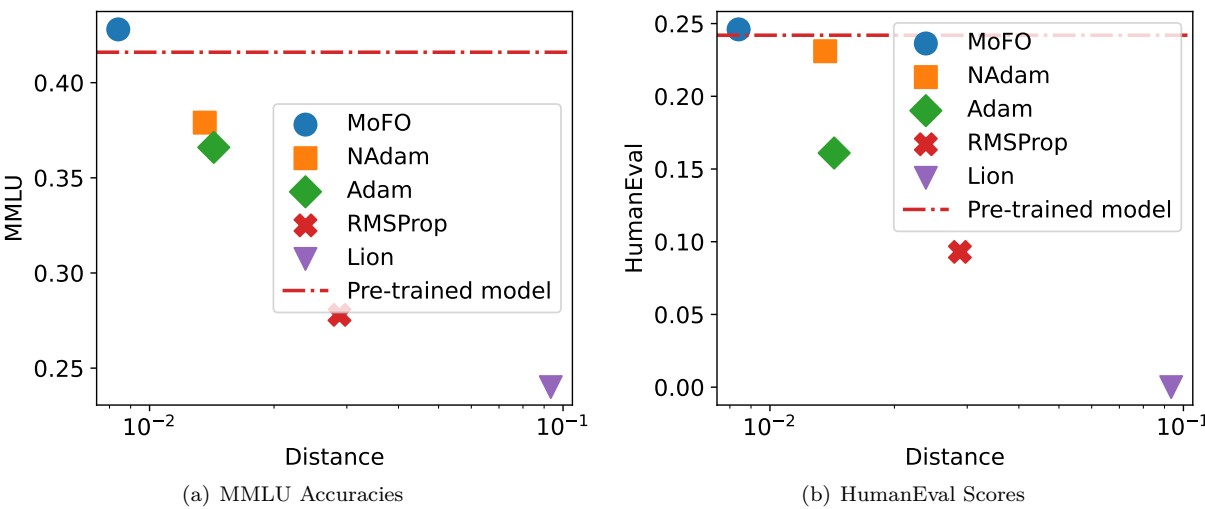

(a) MMLU Accuracies

(b) HumanEval Scores

Figure 10: (a) MMLU accuracies and (b) HumanEval scores of Llama-2-7B after fine-tuning on the Meta-MathQA dataset using different optimizers. All experiments are run for 2 epochs.

### B.4 Supplemental Explanation of Example 1

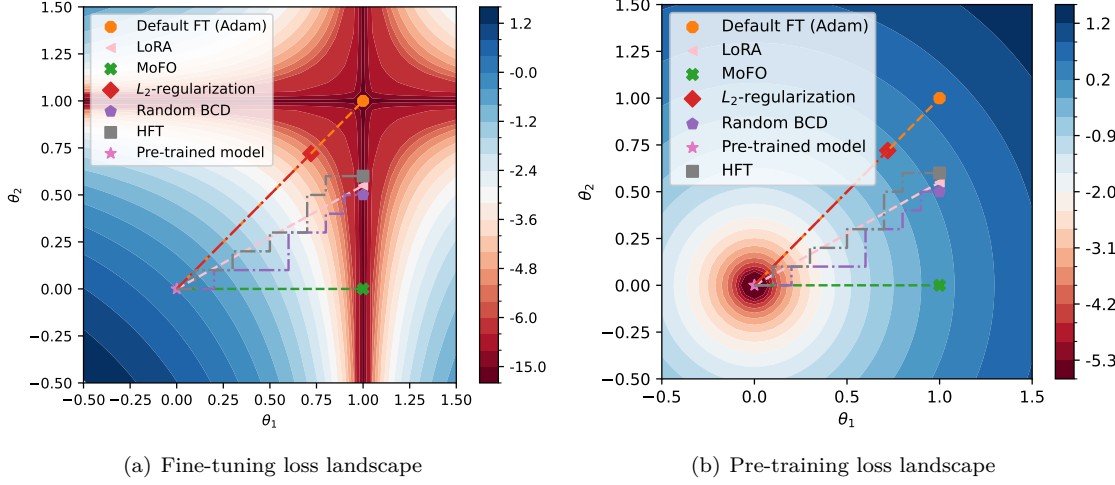

(a) Fine-tuning loss landscape

(b) Pre-training loss landscape

Figure 11: The loss landscapes of the example. We plot the landscapes on (a) the fine-tuning loss and (b) the pre-training loss. The color bar indicates the loss value—(a) fine-tuning loss and (b) pre-training loss. A logarithmic scale is applied to the loss values for better visualization. In Example 1, MoFO converges to a minimum closest to the pre-trained model, with a low pre-training loss.

In addition to Default FT and MoFO, we also analyze four other optimization methods in Example 1, namely $L_2$ regularization (Li et al., 2018), Half Fine-tuning (HFT) (Hui et al., 2024), Random BCD (Nesterov, 2012), and Low-Rank Adaptation (LoRA) (Hu et al., 2022). These methods are also introduced in Section 4.

For the $L_2$ regularization method, we add a regularization term $\lambda\|\theta - \theta_{\text{pretrain}}\|_2^2$ to the fine-tuning loss to encourage the model to stay closer to the pre-trained state. As shown in Figure 11(b), the $L_2$ regularization approach remains closer to the pre-trained model, thereby achieving a smaller pre-training loss. However, since the fine-tuning objective is modified, Figure 11(a) shows that $L_2$ regularization does not reach the minimum of the fine-tuning loss.

We note that HFT operates in a manner similar to Random BCD, which randomly selects a subset of coordinates (e.g., $\theta_1$ or $\theta_2$) to update. Figure 11 further illustrates that both HFT and LoRA do not converge to minima as close to the pre-trained model as MoFO does, indicating that they may undergo higher levels of forgetting compared to MoFO.

For LoRA, we make the following modelling in the landscape visualization example. The core principle of LoRA (Low-Rank Adaptation) involves approximating the original training space by a low-rank subspace. Since we consider a two-dimensional training space for visualizing the landscape, we set the rank of LoRA space to 1. Specifically, the parameters $\theta_1$ and $\theta_2$ exhibit a linear relationship. Given that the pre-trained model is $(0, 0)$, the parameters under LoRA are set to satisfy $\theta_2 = \beta\theta_1$, where $\beta$ is a hyperparameter we set to 0.5. Figure 11 shows that LoRA converges to a closer local minimum than Default FT.

### B.5 Synthetic Experiment for Example 1

In this subsection, we conduct a synthetic experiment to provide a more concrete illustration of Example 1. Specifically, we set the parameter dimension to $d = 10$, with parameters $\theta = (\theta_1, \ldots, \theta_{10})$. The pretraining loss is defined as the squared $L_2$ norm of the parameters: $L_{\text{pretrain}}(\theta) = \frac{1}{2}\|\theta\|_2^2$, with the pre-trained model given by $\theta_{\text{pretrain}} = (0, 0, \ldots, 0)$. Starting from the pre-trained model, we optimize the parameters with respect to the fine-tuning loss $\mathcal{L}(\theta) = \prod_{i=1}^{d}(a_i\theta_i - b_i)^2$, where $a_i, b_i > 0$ for any $1 \le i \le d$. The coefficients $a_i$ and $b_i$ are sampled from a standard normal distribution; to ensure positivity, we take their absolute values and add 0.3 and 0.1, respectively.

In this experiment, we compare two optimizers: Adam and MoFO. For each, we perform a grid search for the optimal learning rate over the set $\{10^{-2}, 10^{-3}, 10^{-4}\}$. We consider the fine-tuning process to have converged to a minimum when the fine-tuning loss drops below $10^{-8}$ within 10000 iterations. At convergence, we record two metrics: the Euclidean distance from the fine-tuned model to the original pre-trained model, and the value of the pretraining loss. The entire experiment is repeated across three different random seeds for robustness.

As presented in Table 4, the results indicate that the minimum found by MoFO is at roughly half the distance from the pre-trained model compared to the one found by Adam. Concurrently, MoFO achieves a lower pre-training loss. These findings provide strong evidence that MoFO can converge to minima closer to the pre-trained model in Example 1, thereby supporting Theorem 2.

Table 4: The Euclidean distance from the fine-tuned model to the original pre-trained model, and the pretraining loss of parameters after optimizing the fine-tuning loss using Adam and MoFO. The results show that MoFO finds a fine-tuning minimum that is closer to the pre-trained model compared to the one found by Adam.

|                              | Adam  | MoFO  |
| ---------------------------- | ----- | ----- |
| Distance to pre-trained model | 0.549 | 0.275 |
| Pre-training loss            | 0.151 | 0.038 |

# C  Implementation Details

## C.1  Datasets for Fine-Tuning.

**MetaMathQA** (Yu et al., 2024b). This dataset comprises 395K math question-answer pairs. Numerous studies indicate that LLMs significantly enhance performance metrics on mathematical benchmarks such as GSM8K after fine-tuning on this dataset. We randomly select 10% of this dataset for training LLMs, which includes 39.5K question-answer pairs.

**PMC-LLaMA-Instructions** (Wu et al., 2024). This dataset comprises 514K instruction-response pairs. Fine-tuning LLMs on this dataset has been shown to enhance performance on medical NLP tasks, such as PubMedQA (Jin et al., 2019), MedMCQA (Pal et al., 2022), and MedQA (Jin et al., 2021). We randomly sampled 51K instances with prompt lengths less than 750 characters for training our models.

**Magicoder-Evol-Instruct** (Wei et al., 2023). This dataset comprises 110K instruction-response pairs related to coding, and it is decontaminated and redistributed from the Evol-CodeAlpaca-V1 dataset (Luo et al., 2023b) We randomly select 39.5K question-answer pairs from this dataset to fine-tune LLM and enhance their coding capability.

**TRACE benchmark dataset** (Wang et al., 2023b). TRACE benchmark is designed with a comprehensive set of 8 distinct tasks across various domains, including domain-specific knowledge, multilingual proficiency, code generation, and mathematical reasoning.

## C.2  Evaluation Metrics for Instruction Fine-Tuning

We employ a comprehensive suite of widely used benchmarks to assess the performance and potential catastrophic forgetting effects on the general capabilities of LLMs after instruction fine-tuning. The benchmarks are as follows:

- **Factual knowledge (MMLU)**: We use the Massive Multitask Language Understanding (MMLU) benchmark (Hendrycks et al., 2021) to evaluate factual knowledge across 57 diverse subjects, ranging from STEM fields and the humanities to social sciences. Evaluations are performed using 8-bit precision with the open-instruct implementation, and by following the setup of (Hui et al., 2024), we report the 0-shot accuracy.

- **Common sense reasoning (CR)**: To measure the commonsense reasoning capabilities of LLMs, we employ the widely recognized benchmarks ARC-Challenge (ARC-C), ARC-Easy (ARC-E) (Clark et al., 2018), and HellaSwag (Zellers et al., 2019), collectively referred to as the Commonsense benchmark. We use the average of their metrics as the evaluation, conducting assessments using the LM Eval Harness framework (Gao et al., 2023) and reporting the 0-shot accuracy based on the "acc_norm, none" metric.

- **Mathematical Reasoning (GSM8K)**: We assess mathematical reasoning capability using GSM8K (Cobbe et al., 2021), which consists of 8.5K high-quality grade school math problems. Evaluations are conducted on the test set using the LM Eval Harness framework prompting in a 5-shot setting, reporting the "exact_match, flexible-extract" metric.

- **Code Generation (HumanEval)**: We adopt HumanEval (Chen et al., 2021), comprising 164 unique programming problems, to evaluate the coding capabilities of LLMs. For chat experiments, we report the pass@10 performance.

- **Medical Question Answering (MedQ)**: To assess medical knowledge, we utilize three benchmarks—PubMedQA (Jin et al., 2019), MedMCQA (Pal et al., 2022), and MedQA (Jin et al., 2021). Evaluations are performed using the LM Eval Harness framework. For PubMedQA, we report the "acc, none" metric; for MedMCQA and MedQA, we report the "acc_norm, none" metric.

- **Instruction Following (IFEval)**: We evaluate the instruction-following ability of LLMs using the IFeval benchmark. Evaluations are conducted with the LM Eval Harness implementation, and we report the "inst_level_strict_acc, none" metric.

All benchmarks—including CommonSense, GSM8K, PubMedQA, MedMCQA, MedQA, and IFeval—are evaluated using the LM Eval Harness framework (Gao et al., 2023), following their default settings unless specified otherwise.

### C.3 Hyperparameter Configurations

**Instruction fine-tuning.** In our instruction fine-tuning experiments, we follow the implementation of Ivison et al. (2023). For instruction fine-tuning, we set the maximum sequence length to 1024, the global batch size to 128, and we train the model for 2 epochs. For the Llama-2-7B model, we use a learning rate of 2e-5 and 0 warm-up ratio, with a cosine decay learning rate scheduler. The learning rate is set to 2e-5 for fine-tuning both the Llama-2-7B-Chat model on the MetaMathQA dataset and the Gemma-2B-IT model, while a learning rate of 1e-5 is used for fine-tuning the Llama-2-7B-Chat model on the PMC-LLaMA-Instruct dataset; all these settings employ a warm-up ratio of 0.03 and a cosine decay learning rate scheduler. For LoRA, we set the learning rate as 1e-4. The other hyperparameters in the experiments are as follows.

**Fine-tuning Llama-2-7B on MetaMathQA.**

- Learning rate: 2e-5.

- Update fraction of MoFO: $\alpha = 15\%$.

- LoRA: $r = 4, 16, 64, 256$. We report the best-performing hyperparameter configuration for the fine-tuning task in Table 1, which, in this case, is $r = 256$.

**Fine-tuning Llama-2-7B-Chat on PMC-LLaMA-Instruct.**

- Learning rate: 1e-5.

- Update fraction of MoFO: $\alpha = 15\%$.

- LoRA: $r = 16, 256$. We report the best-performing hyperparameter configuration for the fine-tuning task in Table 5, which, in this case, is $r = 256$.

**Fine-tuning Llama-2-7B-Chat on MetaMathQA.**

- Learning rate: 2e-5.

- Update fraction of MoFO: $\alpha = 15\%$.

- LoRA: $r = 16, 256$. We report the best-performing hyperparameter configuration for the fine-tuning task in Table 7, which, in this case, is $r = 256$.

**Fine-tuning Gemma-2B-IT on MetaMathQA.**

- Learning rate: 2e-5.

- Update fraction of MoFO: $\alpha = 5\%$.

- LoRA: $r = 16, 256, 512$. We report the best-performing hyperparameter configuration for the fine-tuning task in Table 6, which, in this case, is $r = 512$.

**Fine-tuning Llama-2-7B-Chat on Magicoder-Evol-Instruct.**

- Learning rate: 1e-5.

- Update fraction of MoFO: $\alpha = 20\%$.

- LoRA: $r = 16, 256$. We report the best-performing hyperparameter configuration for the fine-tuning task in Table 6, which, in this case, is $r = 256$.

**Hyperparameters in the Pareto comparison.** To provide a comprehensive comparison, we explore various hyperparameter settings for $\lambda_1$, $\lambda_2$, LoRA's rank, and the update fraction $\alpha$ in MoFO in Figure 5. Specifically, we set $\lambda_1$ as 1e-4, 1e-5, 1e-6, 1e-7, while $\lambda_2$ is set as 1e-2, 5e-3, 1e-3, 5e-4, and 1e-4. The update fraction $\alpha$ in MoFO is set as 5%, 10%, 15%, 20%, 40%, 80%. The rank of LoRA is set as 4, 16, 64, 256.

**Continual fine-tuning.** In our continual fine-tuning experiments, we follow the default settings of the TRACE benchmark. We sequentially train TinyLlama-1.1B on the TRACE benchmark datasets: C-STANCE, FOMC, MeetingBank, Py150, ScienceQA, NumGLUE-cm, NumGLUE-ds, and 20Minuten for 5, 3, 7, 5, 3, 5, 5, and 7 epochs, respectively. We use a learning rate of 1e-5 with a cosine decay schedule and a batch size of 64. The parameter update fraction for MoFO is set to 5%.

All experiments are conducted on four A800 (80GB) GPUs.

### C.4 More Explanation on the partitioning and Calculation of distance

**Partitioning.** We use the default partitioning scheme in PyTorch's Transformer implementation. Different types of parameters within the Transformer, such as query (Q), key (K), value (V) weights for attention heads, and feed-forward network (FFN) weights, are divided into separate partitions. Notably, in the default PyTorch implementation, within a layer, the query (Q) weights of all attention heads are grouped into a single partition. The same applies to the key (K) and value (V) weights. Our momentum-based filtering mechanism is applied to each partition individually. A different parameter partition scheme, along with its corresponding experiments, is presented in Appendix F.6.

**Calculation of distance.** Following the notation in Section 2.2, we suppose that the parameter parameters are partitioned into

$$\theta = (\theta^{(1)}, \theta^{(2)}, \ldots, \theta^{(B)}).$$

Denote the pre-trained model by $\theta_0$ and the fine-tuned model by $\theta$.

First, we calculate the relative change of parameters $\frac{\|\theta^{(k)} - \theta_0^{(k)}\|}{\|\theta_0^{(k)}\|}$ in each partition $k \in \{1, 2, \ldots, B\}$. Second, we compute the distance from the pre-trained model $\theta_0$ to the fine-tuned model $\theta$ by averaging the relative changes across all partitions, defined as:

$$D(\theta, \theta_0) = \frac{1}{B} \sum_{k=1}^{B} \frac{\|\theta^{(k)} - \theta_0^{(k)}\|}{\|\theta_0^{(k)}\|}.$$

# D    Guideline for Setting $\alpha$

Given a pre-trained LLM and a dataset for fine-tuning, we recommend the following procedure:

1. Random Sampling: Randomly sample a small subset of the dataset to serve as a proxy.

2. Grid Search: Perform a grid search over candidate values of $\alpha$ using this proxy subset.

3. Selection: Choose the $\alpha$ configuration that strikes a good balance between fine-tuning performance on the target dataset and preserving the model's general capability.

To illustrate this procedure, we use Llama-2-7B as an example. We randomly sample 10% of the instances from the current training set of MetaMathQA dataset (39.5k) for fine-tuning, and then perform a grid search over $\alpha$ values of 5%, 10%, 15%, 20%, 40%, and 80%. As shown by the green line in Figure 12(a), the fine-tuning performance is relatively stable across these values of $\alpha$. However, the green line in Figure 12(b) indicates that $\alpha = 15\%$ best preserves the model's general capability. Therefore, we set $\alpha = 15\%$.

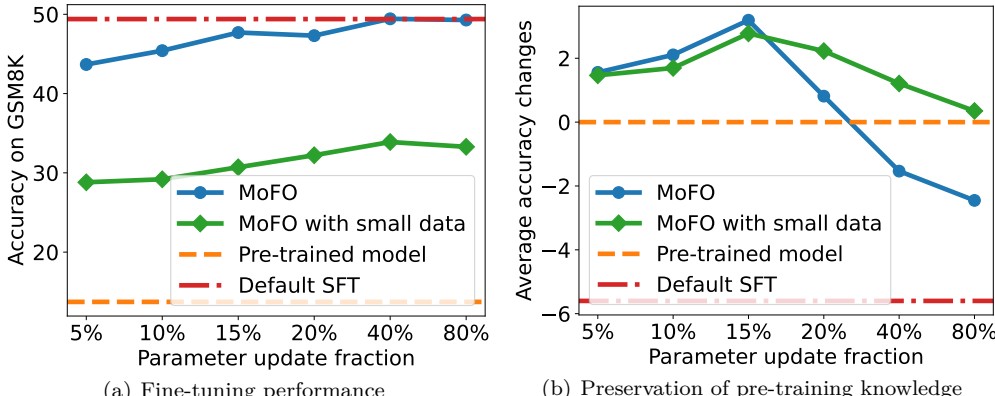

(a) Fine-tuning performance

(b) Preservation of pre-training knowledge

Figure 12: **(a) Fine-tuning performance:** Accuracy on the GSM8K math reasoning task for LLMs of different sizes, fine-tuned via MoFO with varying update fractions ($\alpha$). **(b) Preservation of pre-training knowledge:** Average accuracy changes on MMLU, HumanEval, and commonsense reasoning benchmarks relative to the original pre-trained LLMs, illustrating how much pre-training knowledge is retained. All results are obtained by fine-tuning Llama2-7B on MetaMathQA and its proxy subset. The performance trends under different update fractions on the proxy subset align with those observed on the full dataset.

Furthermore, by comparing the blue and green lines in Figure 12, we observe that the trend of model performance with respect to $\alpha$ on the small proxy subset is consistent with the trend observed when fine-tuning on the full dataset. This implies that a small, randomly sampled subset is sufficient to guide the selection of a suitable $\alpha$.

Empirically, the optimal $\alpha$ often lies between 5% and 20%. A more fine-grained grid search within this range can be performed if needed. Designing more refined and efficient strategies for tuning $\alpha$ is left for future work.

# E Additional Experiments on Instruction Fine-tuning

This section begins with a comparison of MoFO and baseline methods across additional datasets and models in E.1. In E.2, we explore the combination of LoRA and MoFO to assess their performance. Finally, in E.3, we compare MoFO with several algorithms designed to mitigate forgetting.

## E.1 More Experimental Results in Instruction Fine-Tuning

Table 5: The performance on the fine-tuning task (medical QA task), measured by MedQ, and general capability scores of Llama-2-7B-Chat after fine-tuning on the PMC-LLaMA-Instruct dataset. The figure on the right visualizes both MedQ accuracy and general capability scores. The results show that MoFO achieves comparable performance in the MedQ while significantly mitigating forgetting of general capabilities. Bold values denote the best results among these methods.

| Method | MedQ | General Capability | | | |
| | | CR | IFEval | HumanEval | Avg. |
|---|---|---|---|---|---|
| Llama-2-7B-Chat | 49.8 | 65.6 | 41.4 | 24.3 | 43.8 |
| Default FT | 54.3 | 64.6 | 32.1 | 20.6 | 39.1 |
| HFT | **54.4** | 65.2 | 33.5 | 23.1 | 40.6 |
| LoRA | 54.2 | 64.4 | 33.9 | 23.5 | 40.6 |
| MoFO | 54.3 | **65.6** | **38.6** | **25.0** | **43.1** |

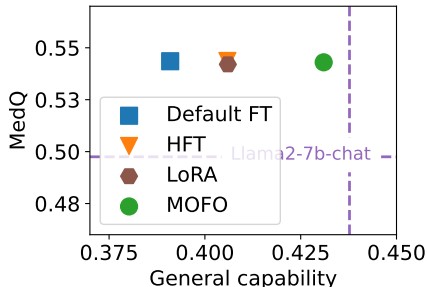

**Results of fine-tuning on PMC-LLaMA-Instruct.** We fine-tune Llama-2-7B-Chat on the PMC-LLaMA-Instructions dataset using various baseline methods and present the experimental results on medical question answering (MedQ) and general capabilities in Table 5. Since the MMLU benchmark already contains medical-related instances (Hendrycks et al., 2021), which may lead to improved performance after fine-tuning, we instead use IFEval to assess general capabilities.

MoFO performs well on the fine-tuning task of medical QA. It achieves compatible performance compared to Default FT and HFT. In terms of general capabilities, MoFO demonstrates the least degradation compared to other baselines, with an average accuracy reduction of only 0.2%. Specifically, on the IFEval benchmark, our method only exhibits a minor reduction of 0.3%, while Default FT, HFT, and LoRA experience significant degradations ranging from 7.5% to 9.3%. On code generation (HumanEval) tasks and commonsense reasoning (CR) benchmarks, our method also only exhibits a minor reduction less than 0.2%.

Table 6: The performance of the fine-tuning task (math), measured by GSM8K, and the general capability scores of Gemma-2B-IT after fine-tuning on the MetaMathQA dataset. The figure on the right visualizes both GSM8K accuracy and general capability scores. The results show that MoFO achieves comparable performance in the fine-tuning task, while significantly mitigating forgetting of general capabilities. Bold values denote the best results among these methods.

| Method | GSM8K | General Capability | | | |
| | | CR | IFeval | HumanEval | Avg. |
|---|---|---|---|---|---|
| Gemma-2B-IT | 11.4 | 57.6 | 33.6 | 31.5 | 40.9 |
| Default FT | 42.0 | 52.1 | 24.3 | 20.6 | 32.3 |
| HFT | 41.5 | 53.9 | 24.1 | 21.2 | 33.1 |
| LoRA | 40.6 | 54.4 | 26.1 | **29.8** | 36.8 |
| MoFO | **42.1** | **55.0** | **28.7** | 29.1 | **37.6** |

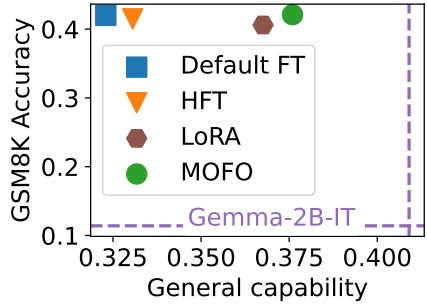

**Results of fine-tuning Gemma-2B-IT on MetaMathQA.** We also explore how MoFO performs in other LLMs. Specifically, we fine-tune Gemma-2B-IT on MetaMathQA using various baseline methods and present the experimental results on mathematical reasoning (GSM8K) and general capabilities in Table 6. The experimental results demonstrate that MoFO achieves comparable performance of the fine-tuning task to Default FT and HFT across different models. In terms of general capabilities, MoFO exhibits significantly less forgetting compared to other baselines. This result demonstrates the versatility of the MoFO algorithm.

Table 7: The performance of the fine-tuning task (math), measured by GSM8K, and the general capability scores of Llama-2-7B-chat after fine-tuning on the MetaMathQA dataset. The figure on the right visualizes both GSM8K accuracy and general capability scores. The results show that MoFO achieves comparable performance in the fine-tuning task, while significantly mitigating forgetting of general capabilities. Bold values denote the best results among these methods.

| Method | GSM8K | General Capability | | | |
| | | CR | IFeval | HumanEval | Avg. |
|---|---|---|---|---|---|
| Llama-2-7B-Chat | 13.7 | 65.6 | 41.4 | 24.3 | 43.8 |
| Default FT | **48.4** | 62.8 | 30.7 | 15.6 | 36.4 |
| HFT | 46.9 | 63.4 | 31.8 | 20.0 | 38.4 |
| LoRA | 45.3 | 63.9 | 35.6 | 21.0 | 40.2 |
| MoFO | 47.1 | **64.0** | **37.1** | **21.7** | **40.9** |

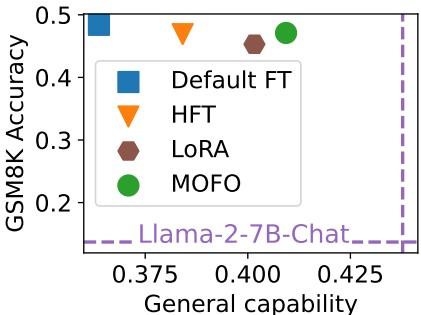

**Results of fine-tuning Llama-2-7B-Chat on MetaMathQA.** We also fine-tune the Llama-2-7B-Chat on the MetaMathQA dataset. The results are presented in Table 7. The results demonstrate that our approach achieves performance comparable to Default FT and HFT while exhibiting less forgetting compared to baseline methods.

Table 8: The performance of the fine-tuning task (coding), measured by HumanEval, and the general capability scores of Llama-2-7B-Chat after fine-tuning on the Magicoder-Evol-Instruct dataset. Here we choose the three benchmarks exhibiting the most significant forgetting. We set the rank of LoRA as 256, and $\alpha$ of MoFO is set as 20%. The results show that MoFO achieves comparable performance in the fine-tuning task, while mitigating forgetting of general capabilities. Bold values denote the best results among these methods.

| Method | HumanEval | General Capability | | | |
| | | ARC-E | ARC-C | IFEval | Avg. |
|---|---|---|---|---|---|
| Llama-2-7B-Chat | 24.2 | 74.5 | 46.3 | 41.1 | 54.0 |
| Default FT | 56.2 | 71.2 | 45.2 | 33.5 | 50.0 |
| HFT | 50.7 | 71.5 | 45.6 | **36.3** | 51.1 |
| LoRA | 48.6 | **72.1** | 45.1 | 33.4 | 50.2 |
| MoFO | 53.3 | **72.1** | **46.0** | 36.1 | **51.4** |

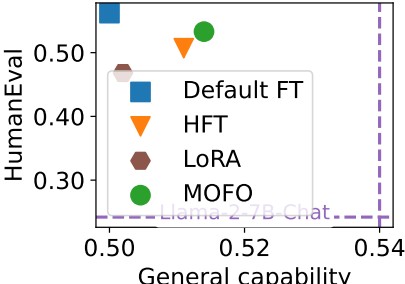

**Results of fine-tuning Llama-2-7B-Chat on Magicoder-Evol-Instruct.** We also fine-tune the Llama-2-7B-Chat on the Magicoder-Evol-Instruct dataset. We use ARC-Easy (ARC-E) and ARC-Challenge (ARC-C) scores to measure general capability. The results in Table 8 demonstrate that our approach outperforms the baselines in the fine-tuning tasks and exhibits less forgetting compared to baseline methods.

In summary, our MoFO algorithm shows competitive performance in instruction fine-tuning while preserving the general capabilities, effectively alleviating forgetting.

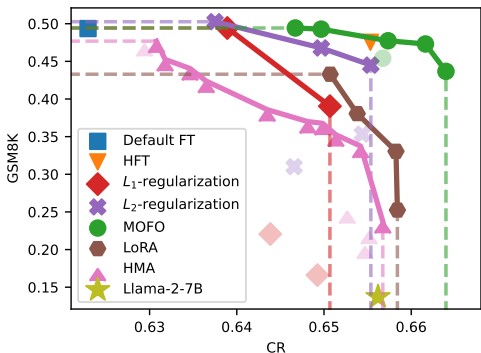

Figure 13: The performance on the math task (GSM8K) and the scores in Commonsense Reasoning of Llama-2-7B after fine-tuning on the MetaMathQA dataset. The results show that the MoFO algorithm achieves a better Pareto front. The pink triangle represents the model obtained through HMA.

## E.2   Experiment on the Combination of MoFO and LoRA

In addition to using LoRA as a baseline for comparison, we can also view it as an orthogonal method that can be integrated with MoFO. In the fine-tuning stage, LoRA restricts the trainable parameter space to a low-rank subspace. We note that the comparison of LoRA and MoFO (PEFT version) essentially evaluates MoFO and Adam within this same low-rank subspace defined by LoRA. To investigate this further, we conduct a comparative experiment following the setup in Table 1 of Section 4.2. Table 9 implies that MoFO + LoRA effectively mitigates the forgetting issue that arises when using LoRA alone.

Table 9: The performance of the fine-tuning task (math), measured by GSM8K, and the general capability scores of Llama-2-7B after fine-tuning on the MetaMathQA dataset. The results show that MoFO + LoRA preserve more pre-training knowledge than using LoRA alone.

| Method | GSM8K | General Capability | | | Avg. |
| --- | --- | --- | --- | --- | --- |
| | | CR | MMLU | HumanEval | |
| LoRA | 43.3 | 65.1 | 37.7 | 26.4 | 43.1 |
| LoRA + MoFO | **43.4** | **65.5** | **39.4** | **26.7** | **43.9** |

## E.3   Comparison with More Fine-Tuning Methods

### Experiments on Heterogeneous Model Averaging (HMA)

We compare our proposed method with the Heterogeneous Model Averaging (HMA) (Lin et al., 2024a). HMA approach evenly divides the LLM into three parts—the input part, the middle part, and the output part—and averages these parts with different ratios. To facilitate a comprehensive comparison, following the setting in Section 4.2, we evaluate the fine-tuning and forgetting mitigation performance for different HMA strategies. We select 15 different combinations of averaging ratios for different parts as follows: {(0.05, 0.2, 0.35), (0.1, 0.2, 0.3), (0.2, 0.2, 0.2), (0.3, 0.2, 0.1), (0.35, 0.2, 0.05), (0.3, 0.5, 0.7), (0.4, 0.5, 0.6), (0.5, 0.5, 0.5), (0.6, 0.5, 0.4), (0.7, 0.5, 0.3), (0.65, 0.8, 0.95), (0.7, 0.8, 0.9), (0.8, 0.8, 0.8), (0.9, 0.8, 0.7), (0.95, 0.8, 0.65)}. We plot the results to construct a Pareto front in Figure 13.

Results show that our proposed method, MoFO achieves a more effective Pareto front compared to the baselines.

### Experiments on CoFiTune and Soft-masking

Zhang et al. (2024a) introduces CoFiTune, a coarse-to-fine framework that balances specificity and versatility in LLMs by selectively updating specific modules and employing a soft-masking mechanism, which is introduced

by Ke et al. (2023b;a). We have compared MoFO with CoFiTune (with and without soft-masking) and the vanilla soft-masking method alone, following the setting in Table 1 of Section 4.2. The results, presented in Table 10 below, demonstrate that MoFO outperforms these methods in both fine-tuning performance and mitigating forgetting. The results demonstrate that

- CoFiTune achives similar forgetting mitigation performance as MoFO, but underperforms MoFO on fine-tuning tasks.

- Vanilla Soft-masking exhibits slightly reduced performance in both fine-tuning tasks and mitigating forgetting than MoFO. These findings underscore the advantages of our proposed method.

Table 10: The performance of the fine-tuning task (math), measured by GSM8K, and the general capability scores of Llama-2-7B after fine-tuning on the MetaMathQA dataset. The results show that MoFO achieves comparable performance in the fine-tuning task, while significantly mitigating forgetting of general capabilities.

| Method | GSM8K | General Capability | | | |
|---|---|---|---|---|---|
| | | CR | MMLU | HumanEval | Avg. |
| MoFO | **47.7** | 65.7 | 42.7 | 24.6 | 44.3 |
| Vanilla-SoftMask | 46.4 | 65.6 | 42.9 | 23.2 | 43.9 |
| CoFiTune w/o SoftMask | 37.7 | 65.4 | 42.1 | 25.8 | **44.4** |
| CoFiTune w/ SoftMask | 34.4 | 65.0 | 41.5 | 25.6 | 44.0 |

From the results, we can see that MoFO achieves higher scores on the fine-tuning tasks while effectively reducing knowledge forgetting, demonstrating its superiority over these methods.

## E.4 Comparison with More Parameter-Efficient Fine-Tuning Methods

Parameter-Efficient Fine-Tuning (PEFT) encompasses a collection of fine-tuning methods designed to reduce the computational cost required for model training. In this subsection, we mainly focus on three famous PEFT methods:

- Adapter (Houlsby et al., 2019): This approach involves inserting trainable "adapter" modules into every layer of model. During fine-tuning, the parameters of these adapter modules are updated for downstream tasks, while the majority of the original model's parameters remain frozen.

- BitFit (Zaken et al., 2021): This method reduces the number of trainable parameters by fine-tuning the bias terms of the model on a given downstream task, keeping other weights frozen.

- LoRA (Hu et al., 2022): is a widely-used, parameter-efficient fine-tuning method. LoRA trains low-rank matrix adaptations on the base model's weights. Recent work (Biderman et al., 2024) demonstrates that LoRA can mitigate forgetting.

We compared MoFO with Adapter, BitFit, and LoRA, following the setup in Table 1 of Section 4.2. For the Adapter method, we set the learning rate to 1e-4 and performed a grid search for the adapter size over the values {16, 64, 128}. We report the best-performing hyperparameter configuration for each method on the fine-tuning task.

As shown in Table 11, all three PEFT baselines (BitFit, Adapter, and LoRA) achieve higher average general capability scores than Default FT, indicating that they are indeed effective at mitigating catastrophic forgetting. However, they still lag behind MoFO in this regard.

Moreover, compared to MoFO, the three PEFT methods perform markedly worse on the fine-tuning task (GSM8K), with the gap being especially pronounced for Adapters and BitFit. We conjecture that this may stem from the fact that Adapters and BitFit were originally proposed and evaluated on relatively simple

classification or QA benchmarks (e.g., GLUE) with masked language models. While these methods were effective in such settings, they may be less suited to today's more challenging domain-specific LLM tasks, leading to the weaker performance observed here.

Table 11: The performance of the fine-tuning task (math), measured by GSM8K, and the general capability scores of Llama-2-7B after fine-tuning on the MetaMathQA dataset. The results show that MoFO outperforms three PEFT methods.

| Method | GSM8K | General Capability | | | |
|---|---|---|---|---|---|
| | | CR | MMLU | HumanEval | Avg. |
| Llama-2-7B | 13.7 | 65.6 | 42.0 | 24.2 | 43.9 |
| Default FT | 49.4 | 62.3 | 36.6 | 16.1 | 38.3 |
| MoFO | **47.7** | **65.7** | **42.7** | 24.6 | **44.3** |
| BitFit | 15.1 | 64.8 | 36.1 | 24.4 | 41.8 |
| Adapter | 24.4 | 63.2 | 32.3 | 21.8 | 39.1 |
| LoRA | 43.3 | 65.1 | 37.7 | **26.4** | 43.1 |

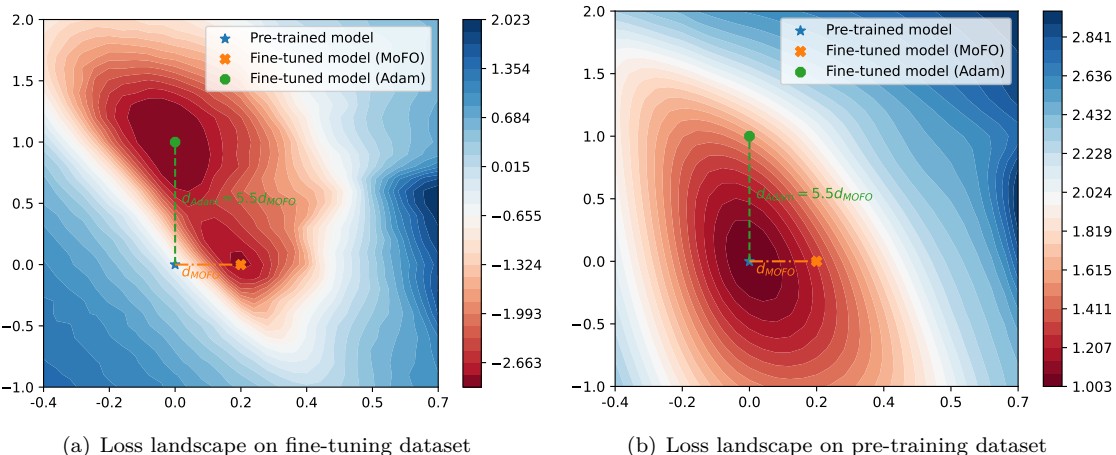

(a) Loss landscape on fine-tuning dataset        (b) Loss landscape on pre-training dataset

Figure 14: The loss landscapes of Pythia-160m after fine-tuning on a subset of the FLAN dataset using Adam optimizer and MoFO. We plot the loss landscapes on (a) the fine-tuning dataset and (b) the pre-training dataset (Pile). We visualize a 2D weight-space plane spanned by the vector from the pre-trained model to the MoFO-tuned model (x-axis) and to the Adam-tuned model (y-axis). Axes are normalized so that one unit equals the length of the pre-trained→ Adam vector. The color bar indicates the loss value—(a) fine-tuning loss and (b) pre-training loss. A logarithmic scale is applied to the loss values for better visualization. We find that MoFO, reaching a closer point to the pre-trained model, has minimal fine-tuning loss and lower pre-training loss, compared to Adam.

## F    More Explorations on MoFO

This section aims to provide a deeper understanding of MoFO through a series of experiments. In Appendix F.2, we conduct an efficiency analysis of MoFO. In Appendix F.3, we present additional comparative experiments on different filtering strategies. In Appendix F.4, we investigate why the momentum-filtered update rule in MoFO achieves optimal fine-tuning performance compared to other update strategies. In Appendix F.5, we present a preliminary analysis on why MoFO might compare favorably to $L_1/L_2$ regularization. In Appendix F.6, we investigate the performance of MoFO under an alternative parameter partitioning strategy. Finally, in Appendix F.7, we explore the application of MoFO to the Lion optimizer.

### F.1    Validating MoFO's Impact on Preserving Pre-training Knowledge through Proximity

In this section, we empirically examine whether MoFO achieves its intended goal of converging to a minimum closer to the pre-trained model and mitigating forgetting mentioned in Section 2.

Our exploratory experiment shows that MoFO indeed converges to a minimum closer to the pre-training model. As shown in Figure 14(a), both MoFO and the Adam optimizer achieve minimal fine-tuning loss, indicating that switching from Adam to MoFO does not lead to performance degradation. Moreover, the distance from the pre-trained model to the minimum reached by MoFO is approximately 20% of that reached by the default Adam optimizer.

Our experiment demonstrates that the reduced parameter movement achieved by MoFO effectively mitigates the forgetting of pre-training knowledge. As shown in Figure 14(b), the fine-tuned model using MoFO experiences a smaller increase in pre-training loss. Additionally, Table 12 shows that MoFO achieves higher accuracy on commonsense reasoning tasks, indicating less forgetting.

Table 12: Pythia-160m's performance on common sense tasks, after being fine-tuned with the Adam optimizer and MoFO. The results indicate that MoFO significantly mitigates catastrophic forgetting. Bold values denote the best results among these optimizers.

|  | HellaSwag | ARC-easy | ARC-challenge | Average |
|---|---|---|---|---|
| Pythia-160m | 30.1 | 39.6 | 23.8 | 31.2 |
| Adam | 28.3 | 37.4 | 22.1 | 29.3 |
| MoFO | **29.9** | **42.0** | **22.9** | **31.6** |

### F.2 Efficiency Analysis on MoFO

We claim that MoFO does not lead to significant reduced fine-tuning efficiency. We provide an efficiency analysis by comparing the total training time between MoFO and Default FT on three LLMs with different sizes. The parameter update fraction of MoFO is set as 10%. The experimental results show that although MoFO requires computing a filter to select parameters with the largest momentum magnitudes, the additional computational overhead is minor. As shown in Table 13, the additional training time incurred by MoFO is approximately 4%–5%, which is relatively minor and manageable in practical applications.

Table 13: Comparison of total training time for Default FT and MoFO on various LLaMA models and datasets. The additional time incurred by MoFO is around 4–5%, which is relatively minor in practical applications. LLaMA3-8B is fine-tuned on the UltraFeedback dataset (Cui et al., 2024).

| Model | Default FT | MoFO | Additional Training Time |
|---|---|---|---|
| LLaMA3.2-1B | 49m22s | 51m24s | 4.1% |
| LLaMA3.2-3B | 1h30m18s | 1h34m24s | 4.5% |
| LLaMA3-8B | 5h2m0.69s | 5h17m34.01s | 5.0% |

### F.3 Further Comparative Experiments on Filtering Strategies

To further substantiate the claim in Section 4.4, we conduct an additional comparison of filtering strategies—complementary to the results in Table 3—using a different dataset and model. Specifically, we fine-tune Gemma-2B-IT on the IFEval-like dataset[7] using several BCD variants. The IFEval-like dataset contains instruction–response pairs in the style of the IFEval benchmark. For these experiments, we randomly sample 39.5k instances from the dataset for training, set the learning rate to $1\times10^{-5}$, and train the model for 2 epochs. We evaluate fine-tuning performance with the IFEval benchmark and assess general capabilities with CR (common-sense reasoning), HumanEval, and BBH (0-shot) (Suzgun et al., 2022).

As shown in Table 14, all BCD variants mitigate forgetting: their average general-capability score exceeds that of Default fine-tuning by at least 2.6%. Although MoFO is lower than MV BCD by only 0.1% in average general capability, it achieves the strongest performance on the target fine-tuning task (IFEval), outperforming Random BCD, Grad BCD, and MV BCD by 5.8%, 2.6%, and 1.3%, respectively.

Taken together, these results indicate that while all filtering strategies are broadly effective at mitigating forgetting, they exhibit distinct differences in task-specific fine-tuning performance. In our setting, MoFO offers a favorable balance—matching the strongest methods on forgetting mitigation while delivering the highest IFEval score among the tested strategies.

---

[7]The "filtered" subset from `https://huggingface.co/datasets/argilla/ifeval-like-data`.

Table 14: The performance on the instruction-following task (IFEval) and general capability scores of Gemma-2B-IT after fine-tuning on IFEval-like dataset using different updating strategies in MoFO. Here we choose the three benchmarks exhibiting the most significant forgetting. For all the update strategies, we set the parameter update fraction $\alpha$ as 10%. Bold values denote the best results among the BCD methods.

| Method | IFEval | General Capability | | | |
| --- | --- | --- | --- | --- | --- |
| | | CR | HumanEval | BBH | Avg. |
| Gemma-2B-IT | 33.6 | 57.6 | 31.5 | 32.7 | 40.6 |
| Default FT | 56.7 | 56.9 | 22.9 | 32.5 | 37.4 |
| Random BCD | 51.4 | **57.5** | 28.0 | 33.9 | 39.8 |
| Grad BCD | 54.6 | 57.3 | 28.1 | 34.1 | 39.8 |
| MV BCD | 55.9 | **57.5** | 28.4 | **34.6** | **40.2** |
| MoFO | **57.2** | 57.3 | **28.5** | 34.4 | 40.1 |

## F.4 Insights of the Choice of Filtering Strategy

In this section, we attempt to address the question: *What makes the momentum-filtered update rule optimal among the candidates?*

We hypothesize that the good performance of the momentum-filtered updating rule arises from its ability to promote more stable and consistent updates throughout training. Specifically, we hypothesize that:

1. **Utilizing momentum instead of gradient filtering leads to more stable updates.** Momentum accumulates historical gradients, so it promotes stability by smoothing out fluctuations in the gradient updates. Thereby, during the training process, the momentum-filtering mechanism chooses updating parameters in a more stable manner.

2. **Excluding the introduction of in the filtering mechanism contributes to more stable updates.** The 2nd-order moment $v_t$ may normalize gradients based on their magnitudes, potentially averaging out the importance of individual parameters within the filtering mechanism. Thereby, during the training process, not incorporating $v$ may help choose updating parameters in a more stable manner.

For the ablation study, we add the GV-filtered BCD methods as a baseline, which replaces MoFO's filter by $\texttt{FLT}_\alpha(g_t/\sqrt{v_t})$. Following the setting of experiments in Appendix B.2 and Figure 1, we run all four methods with the updating fraction $\alpha = 3\%$ over approximately 200 steps. To assess how many parameters change significantly during training, we calculate the percentage of weight parameters whose absolute change exceeds a threshold of 2e-6.

Table 15: Percentage of weight parameters with significant changes (absolute change $> 2 \times 10^{-6}$) during training.

| Method | Percentage of Significant Updates |
| --- | --- |
| MoFO | **29.8%** |
| Gradient BCD | 35.7% |
| MV-filtered BCD | 83.6% |
| GV-filtered BCD | 87.1% |

Table 15 indicates that the parameter updating process of MoFO is more stable, which we think may contribute to its better fine-tuning performance.

### F.5 Preliminary Analysis on Why MoFO Might Compare Favorably to $L_1/L_2$ Regularization

As shown in Figure 5 (Section 4.2), when fine-tuning Llama-2-7B on MetaMathQA, MoFO yields a more favorable Pareto front—balancing fine-tuning performance and forgetting mitigation—than $L_1/L_2$ regularization and LoRA across hyperparameter configurations. This section presents a preliminary analysis offering one possible explanation for why MoFO may compare favorably to $L_1/L_2$ regularization. Importantly, we reuse the exact checkpoints from Figure 5; no additional training runs are introduced.

**Setup.** We reuse the Llama-2-7B MetaMathQA sweeps underlying Figure 5—including $L_1$, $L_2$, and MoFO across the same hyperparameters. For each checkpoint, we compute the (unregularized) fine-tuning loss and the $\ell_2$ norm of the gradient on 512 examples. To factor out severe forgetting, we report results for the subset whose CR scores fall within an acceptable range (same CR metric as in the main text). Figure 15(a) and Figure 15(b) plot fine-tuning loss and gradient norm versus CR, respectively, for this subset of checkpoints drawn from Figure 5.

**Observations.** For comparable CR scores, MoFO checkpoints generally achieve lower fine-tuning losses than those trained with $L_1$ or $L_2$; moreover, they also exhibit smaller gradient norms. Together, these appear to suggest that MoFO converges better than $L_1/L_2$ regularization.

**One plausible explanation.** Penalty methods promote proximity to the pretrained parameters, but they also **modify the training objective and its gradient field**, thereby possibly hindering convergence to a local minimum of the original fine-tuning loss. MoFO instead constrains how updates are applied (via momentum filtering) while continuing to optimize the original fine-tuning loss, which may allow the optimizer to follow task-relevant directions more effectively.

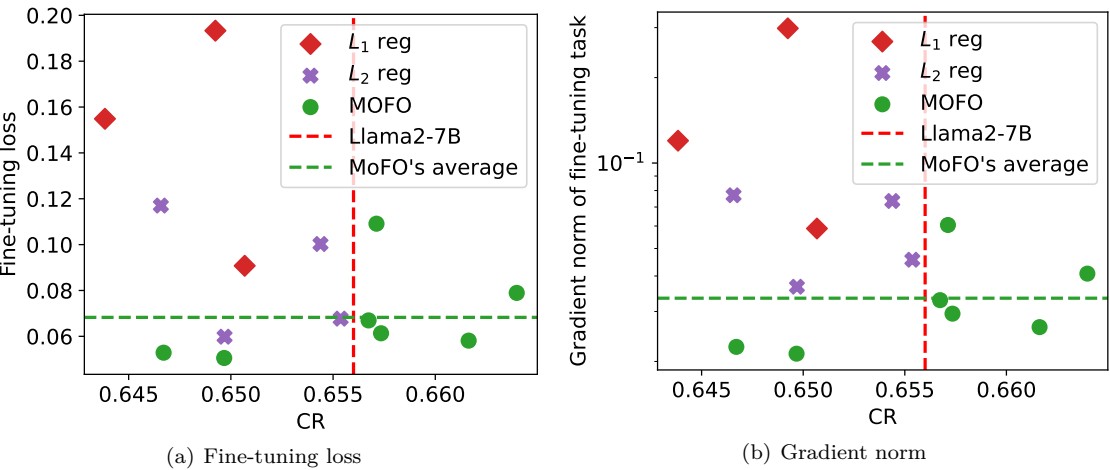

(a) Fine-tuning loss            (b) Gradient norm

Figure 15: (a) Fine-tuning loss and (b) gradient norm versus CR score for Llama2-7B fine-tuned on MetaMathQA with MoFO, $L_1$, and $L_2$ regularization. The results show that MoFO generally achieves a lower fine-tuning loss and gradient norm than $L_1$ and $L_2$ while effectively resisting forgetting.

### F.6 A Different Parameter Partition Scheme

In this section, we propose a different strategy for partitioning model parameters. Different from the default partitioning scheme in PyTorch's Transformer implementation (Appendix C.4), our alternative approach partitions parameters at the granularity of individual attention heads. Recent studies (Zhang et al., 2024d;c) show that the Hessian matrix in Transformers is nearly block-diagonal, with several dense principal sub-blocks. In particular, Zhang et al. (2024c) finds that within the same layer, distinct Q (or K) heads form different blocks in the Hessian.

Motivated by this finding, we treat each head's Q, K, and V weights as separate partitions and apply our momentum-based filtering mechanism to these finer-grained partitions individually. To evaluate our method, we conduct experiments on the LLaMA2-7B models, comparing the default partitioning approach against our alternative head-level partitioning scheme.

Table 16: Performance on the fine-tuning task (GSM8K) and general capabilities of **Llama-2-7B** after fine-tuning. Bold values denote the best results among these methods. The updating fraction $\alpha$ is 15%.

| Method | GSM8K | General Capability | | | |
|---|---|---|---|---|---|
| | | CR | MMLU | HumanEval | Avg. |
| MoFO with default partition | **47.7** | 65.7 | **42.7** | 24.6 | **44.3** |
| MoFO with individual head partition | 46.6 | 65.7 | 41.6 | 24.1 | 43.8 |

---

**Algorithm 3** MoFO + Lion

---

1: Input: Filtering threshold $\alpha$, number of partitions $B$ with the $k$-th partition of size $d_k$, hyperparameters $\beta_1, \beta_2, \lambda$ of Lion optimizer, learning rate schedule $\{\eta_t\}$.
2: Initialize $m_0$ as zero tensors.
3: **for** iteration $t$ from $1, 2, \ldots$ until converge **do**
4:      **for** partition $k$ from 1 to $B$ **do**
5:          $g_t^{(k)} = \nabla_{\theta^{(k)}} \mathcal{L}_{finetune}(\theta_{t-1})$
6:          $c_t^{(k)} = \beta_1 m_{t-1}^{(k)} + (1 - \beta_1) g_t^{(k)}$
7:          **for** entry index $i$ from 1 to $d_k$ **do**
8:             $[\texttt{FLT}_\alpha^{(k)}(m_{t-1})]_i = 1$ **if** $|(m_{t-1}^{(k)})_i|$ is within the top-$\alpha$ of $|m_{t-1}^{(k)}|$'s values **else** 0
9:          **end for**
10:          $\theta_t^{(k)} = \theta_{t-1}^{(k)} - \eta_t(\text{sign}(c_t^{(k)} \odot \texttt{FLT}_\alpha^{(k)}(m_{t-1})) + \lambda \theta_{t-1}^{(k)})$        *# Momentum Filtering*
11:          $m_t^{(k)} = \beta_2 m_{t-1}^{(k)} + (1 - \beta_2) g_t^{(k)}$
12:      **end for**
13:      $\theta_t = \texttt{Concat}(\theta_t^{(1)}, \ldots, \theta_t^{(B)})$
14: **end for**

---

Table 16 shows that both partition schemes yield similar performance.

From the above initial experiment results, we believe that there is room for further exploration in better partitioning strategies.

### F.7 Momentum Filtering Mechanism on the Lion Optimizer

---

**Algorithm 2** Lion Optimizer

---

1: Input: Number of partitions $B$ with the $k$-th partition of size $d_k$, hyperparameters $\beta_1, \beta_2, \lambda$ of Lion optimizer, learning rate schedule $\{\eta_t\}$.
2: Initialize $m_0$ as zero tensors.
3: **for** iteration $t$ from $1, 2, \ldots$ until converge **do**
4:      **for** partition $k$ from 1 to $B$ **do**
5:          $g_t^{(k)} = \nabla_{\theta^{(k)}} \mathcal{L}_{finetune}(\theta_{t-1})$
6:          $c_t^{(k)} = \beta_1 m_{t-1}^{(k)} + (1 - \beta_1) g_t^{(k)}$
7:          $m_t^{(k)} = \beta_2 m_{t-1}^{(k)} + (1 - \beta_2) g_t^{(k)}$
8:      **end for**
9:      $\theta_t = \texttt{Concat}(\theta_t^{(1)}, \ldots, \theta_t^{(B)})$
10: **end for**

---

In this section, we extend our investigation by integrating our proposed MoFO into the Lion optimizer (Chen et al., 2024). We first present the original formulation of the Lion optimizer in Algorithm 2.

Building upon this foundation, we propose Algorithm 3, where momentum filtering is applied to refine the parameter update in every iteration (Lines 7-10).

Following the experimental setup in Appendix B.2 and Figure 1, we conduct comparative experiments to evaluate the effectiveness of 'MoFO + Lion' in mitigating forgetting. As shown in Table 17, incorporating MoFO into Lion improves the average accuracy of preserving old knowledge by 2.6% compared to using Lion alone. However, 'MoFO + Lion' still lags behind 'MoFO + Adam' by 2.5%.

Table 17: Performance comparison on HellaSwag, ARC-easy, and ARC-challenge, along with their average.

| Method | HellaSwag | ARC-easy | ARC-challenge | Average |
|---|---|---|---|---|
| Pythia-160m | 30.1 | 39.6 | 23.8 | 31.2 |
| Adam | 28.3 | 37.4 | 22.1 | 29.3 |
| MoFO (for Adam) | 29.9 | 42.0 | 22.9 | 31.6 |
| Lion | 26.5 | 29.0 | 24.1 | 26.5 |
| Lion + MoFO | 27.5 | 36.7 | 23.2 | 29.1 |

