# OpenReview forum: "MoFO: Momentum-Filtered Optimizer for Mitigating Forgetting in LLM Fine-Tuning"
_TMLR — Accepted by TMLR_

### Review · Reviewer_3rAc · 2025-08-02

**Summary Of Contributions:**

This paper presented a novel Momentum-Filtered Optimizer (MoFO) framework for mitigating forgetting in LLM fine-tuning. Specifically, MoFo only updated the model parameters with the largest momentum magnitudes. The convergence of MoFo-based algorithm was analyzed. Experimental results demonstrated that MoFo achieved better performance than baselines.

**Audience:**

Yes

**Broader Impact Concerns:**

No concerns on the ethical implications of the work need to be addressed.

**Claims And Evidence:**

No

**Requested Changes:**

(1) The caption of Figure 2 states that "Given that LLaMA2’s original training data is not publicly available, we use the RedPajama dataset (Computer, 2023) as a comparable alternative". It is confusing why RedPajama is used here.

(2) It states that "Figure 2(b) indicates a strong negative correlation with MMLU accuracy". However, it seems that MoFO involves a strong positive correlation. This phenomenon needs more explanations.

(3) It is unclear how all model parameters can be partitioned into $B$ blocks in Algorithm 1.

(4) The running efficiency of MoFo can be analyzed compared to baselines.

**Strengths And Weaknesses:**

Strengths:

(S1) A MoFo framework was proposed to only update the model parameters with the largest momentum magnitudes.

(S2) The theoretical convergence result of the MoFO algorithm was derived.

(S3) Experimental results demonstrated that MoFo achieved better performance than baselines in LLM fine-tuning.

Weaknesses:

(W1) One major concern is that the related works on parameter-efficient fine-tuning algorithms are not discussed. The parameter-efficient fine-tuning framework also aims to adjust a small number of parameters. It might be more convincing to explain how the proposed method outperforms parameter-efficient fine-tuning algorithms (e.g., LoRA, Adapters, BitFit etc.) in mitigating forgetting.

(W2) The motivation that "filtering the momentum is more effective than filtering the gradient" is not well justified. Table 3 shows that Gradient-filtered BCD has comparable performance as MoFo. The benefits of MoFo can be further explained both theoretically and empirically.

(W3) The forgetting mitigation in Subsection 3.2 involves the specifical cases of learning functions. It is more convincing to provide the theoretical analysis in general cases.

---

> ### Author Response · Authors · 2025-08-15
> **Response to W1**
>
> ## Comment:
> *One major concern is that the related works on parameter-efficient fine-tuning algorithms are not discussed. The parameter-efficient fine-tuning framework also aims to adjust a small number of parameters. It might be more convincing to explain how the proposed method outperforms parameter-efficient fine-tuning algorithms (e.g., LoRA, Adapters, BitFit etc.) in mitigating forgetting.*
>
> ## Response:
>
> Thanks sincerely for the reviewer's comment.
>
> ### **Part 1: Existing PEFT baseline: LoRA**
>
> First, we would like to note that we have already included a parameter-efficient fine-tuning (PEFT) method, **LoRA**, as one of our baselines (see [[Table 1 in our manuscript]](https://anonymous.4open.science/r/MoFO-TMLR-Rebuttal-C117/main-exp.png) for instance).
>
>
> ### **Part 2: Newly added PEFT baselines: Adapters & BitFit**
>
> In addition, we have supplemented two more PEFT baselines, **Adapters** and **BitFit**, under the same experimental setting as Table 1 (i.e., fine-tuning LLaMA2-7B on the GSM8K task).
>
> As shown in [[this table]](https://anonymous.4open.science/r/MoFO-TMLR-Rebuttal-C117/PEFT-baselines.png), all three PEFT baselines (BitFit, Adapter, and LoRA) achieve higher average general capability scores than Default FT, indicating that they are indeed effective at mitigating catastrophic forgetting. However, they still lag behind MoFO in this regard.
>
> Moreover, compared to MoFO, **the three PEFT methods perform markedly worse on the fine-tuning task** (GSM8K), with the gap being especially pronounced for Adapters and BitFit. We conjecture that this may stem from the fact that Adapters and BitFit were originally proposed and evaluated on relatively simple classification or QA benchmarks (e.g., GLUE) with masked language models. While these methods were effective in such settings, they might be less suited to today’s more challenging domain-specific LLM tasks, leading to the weaker performance observed here.
>
> These additional comparison experiments on PEFT methods have been included in **Appendix E.4** of the revised manuscript.
>
>
> ### **Part 3: More discussion in the related work section**
>
> Moreover, we have added a discussion of parameter-efficient fine-tuning algorithms in the **Related Work** section, with the new content highlighted in blue:
>
>
> Apart from LoRA, Adapters and BitFit are also well-known PEFT methods. In Appendix E.4, we include them as baselines for comparison and find that, while they are slightly less effective than MoFO in mitigating forgetting, their performance on the fine-tuning task is substantially worse than that of MoFO.

---

> ### Author Response · Authors · 2025-08-15
> **Response to W2**
>
> ## Comments:
>
> *The motivation that "filtering the momentum is more effective than filtering the gradient" is not well justified. Table 3 shows that Gradient-filtered BCD has comparable performance as MoFo. The benefits of MoFo can be further explained both theoretically and empirically.*
>
> ## Response:
>
> ### **Part 1: Clarifying the evidence already in Table 3**
>
> Thanks for the reviewer's comment. We kindly clarify the possible confusion: **[[Table 3]](https://anonymous.4open.science/r/MoFO-TMLR-Rebuttal-C117/math-filter-comparison.png) in fact already demonstrates a clear advantage of MoFO (i.e. momentum-filtered BCD) over gradient-filtered BCD.** For a fine-tuned model, our work focuses on two equally important criteria: **forgetting mitigation** and **performance on fine-tuning task**.
>
> Table 3 shows that, while MoFO (momentum-filtered BCD) and Gradient-filtered BCD preserve **comparable general capabilities (measures of forgetting mitigation)**, MoFO attains 5.2\% **higher accuracy on the fine-tuning task** (GSM8K, math task). This result suggests that gradient-filtered BCD, although also effective at reducing forgetting, appears to sacrifice fine-tuning performance.
>
>
> ### **Part 2: Additional Empirical Evidence Demonstrating the Superiority of MoFO**
>
> To further assess MoFO, we added a new comparison between gradient-filtered BCD and MoFO in an additional setting: fine-tuning Gemma-2B-IT on an IFEval-like dataset.
>
> As shown in [[the table]](https://anonymous.4open.science/r/MoFO-TMLR-Rebuttal-C117/IFEval-filter-comparison.png), all BCD variants mitigate forgetting: their average general-capability score exceeds that of standard fine-tuning by at least 2.6\%. Although MoFO is lower than MV BCD by only 0.1\% in average general capability, it achieves the strongest performance on the target fine-tuning task (IFEval), outperforming Random BCD, Grad BCD, and MV BCD by 5.8\%, 2.6\%, and 1.3\%, respectively.
>
> Taken together, these results indicate that while all filtering strategies are broadly effective at mitigating forgetting, they exhibit distinct differences in task-specific fine-tuning performance. In our setting, MoFO offers a favorable balance—matching the strongest methods on forgetting mitigation while delivering the highest IFEval score among the tested strategies.
>
> We have incorporated this experiment and discussion into **Appendix F.3** of the revised manuscript.
>
> ### **Part 3: Supporting Literature for the Use of Momentum Filtering**
>
> While a rigorous theoretical demonstration of the superiority of momentum filtering over gradient filtering **in fine-tuning performance** is difficult, we attempt to present an initial analysis with supporting literature.
>
> * **Temporal denoising.** In modern LLM fine-tuning, each parameter update uses a mini-batch stochastic gradient; such gradients are noisy. Momentum (an exponential moving average of past gradients) aggregates information **of the fine-tuning task** across steps, yielding a more history-aware update with a higher signal-to-noise ratio [1].
> * **Stability and variance reduction.** Classical and modern results show that momentum damps oscillations and reduces variance in stochastic optimization; momentum filtering can inherit these benefits, whereas a gradient-only filter may not [2,3,4].
>
>
> ### **Reference:**
>
> [1] Mandt, Stephan, Matthew D. Hoffman, and David M. Blei. "Stochastic gradient descent as approximate bayesian inference." Journal of Machine Learning Research 18.134 (2017): 1-35.
>
> [2] Polyak, Boris T. "Some methods of speeding up the convergence of iteration methods." Ussr computational mathematics and mathematical physics 4.5 (1964): 1-17.
>
> [3] Polyak, Boris T., and Anatoli B. Juditsky. "Acceleration of stochastic approximation by averaging." SIAM journal on control and optimization 30.4 (1992): 838-855.
>
> [4] Sutskever, Ilya, James Martens, George Dahl, and Geoffrey Hinton. "On the importance of initialization and momentum in deep learning." In International conference on machine learning, pp. 1139-1147. pmlr, 2013.

---

> ### Author Response · Authors · 2025-08-15
> **Response to W3**
>
> ## Comment:
> *The forgetting mitigation in Subsection 3.2 involves the specifical cases of learning functions. It is more convincing to provide the theoretical analysis in general cases.*
>
> ## Response:
>
> We sincerely appreciate the reviewer’s insightful comment regarding the theoretical generality of our forgetting mitigation example. We agree that extending the analysis beyond the specific function class studied here would further strengthen the work. We view our study as a preliminary step toward a less-discussed aspect of BCD (Block Coordinate Descent): a tendency toward proximal minima of the fine-tuning loss, which may has implications for forgetting mitigation. We note that Reviewer NmJ8 comment that this example (Theorem 2) is interesting and could be used to characterize difference between MoFO and Adam.
>
> Much of the existing BCD literature focuses predominantly on:
> - (a) Convergence guarantees [1,2,3];
> - (b) Algorithmic extensions (e.g., block‑selection rules, message‑passing strategies [4]);
> - \(c) Complexity analysis [5,6].
>
> To our knowledge, no prior work has addressed BCD's geometric preference for minima. Though simplified, our synthetic example is designed to capture two characteristics of neural network loss landscapes (described in Remark 1 of the manuscript): nonconvexity and degenerate structures of minima. This abstraction provides an ilustrative example for observing proximity bias.
>
>
> While our current analysis focuses on a specific example, we propose two potential next-step extension for future works:
> - Single-layer transformer analysis: Investigating BCD dynamics in attention-based loss landscapes, where analytically tractable minima structures may enable a preliminary characterization of the tendency;
> - Infinite-width NTK regimes: Investigating whether BCD's geometric convergence properties might be formally derived via neural tangent kernel theory in overparameterized networks.
>
> **Reference:**
>
> [1] Tseng, Paul. "Convergence of a block coordinate descent method for nondifferentiable minimization." Journal of optimization theory and applications 109.3 (2001): 475-494.
>
> [2] Beck, Amir, and Luba Tetruashvili. "On the convergence of block coordinate descent type methods." SIAM journal on Optimization 23.4 (2013): 2037-2060.
>
> [3] Razaviyayn, Meisam, Mingyi Hong, and Zhi-Quan Luo. "A unified convergence analysis of block successive minimization methods for nonsmooth optimization." SIAM Journal on Optimization 23.2 (2013): 1126-1153.
>
> [4] Nutini, Julie, Issam Laradji, and Mark Schmidt. "Let's Make Block Coordinate Descent Converge Faster: Faster Greedy Rules, Message-Passing, Active-Set Complexity, and Superlinear Convergence." arXiv preprint arXiv:1712.08859 (2017).
>
> [5] Sun, Ruoyu, and Mingyi Hong. "Improved iteration complexity bounds of cyclic block coordinate descent for convex problems." Advances in Neural Information Processing Systems, vol. 28, 2015.
>
> [6] Hong, Mingyi, et al. "Iteration complexity analysis of block coordinate descent methods." Mathematical Programming 163.1 (2017): 85-114.

---

> ### Author Response · Authors · 2025-08-15
> **Response to Requested Change 1**
>
> ## Comment:
> *The caption of Figure 2 states that "Given that LLaMA2’s original training data is not publicly available, we use the RedPajama dataset (Computer, 2023) as a comparable alternative". It is confusing why RedPajama is used here.*
>
> ## Response:
>
> Thank you for the helpful comment. We have added the following clarification in **Appendix B.1** of the revised manuscript:
>
> **Reason for using RedPajama to approximate LLaMA-2’s training data.**
>
> We note that original LLaMA-2 training dataset has not been publicly released. Thus, we can only rely on public datasets to approximate LLaMA-2's original training data.
>
> RedPajama project was explicitly designed as an open-source reproduction of the LLaMA training dataset [1,2]. It closely mirrors the data sources outlined in the original LLaMA paper and adopts similar strategies for data collection, mixture, and preprocessing. We believe it serves as a reasonable proxy for approximating LLaMA-2’s training dataset.
>
> **Reference:**
>
> [1] https://www.together.ai/blog/redpajama
>
> [2] Weber, et al. RedPajama: An Open Dataset for Training Large Language Models. NeurIPS 2024.

---

> ### Author Response · Authors · 2025-08-15
> **Response to Requested Change 2**
>
> ## Comment:
> *It states that "Figure 2(b) indicates a strong negative correlation with MMLU accuracy". However, it seems that MoFO involves a strong positive correlation. This phenomenon needs more explanations.*
>
> ## Response:
> Thanks for your sincere comments. Our response is twofold: (i) explanation and exploration of the phenomenon you noted; (ii) Evidence of the negative correlation across different optimizers.
>
> We now present a detailed explanation, which is also included in **Appendix B.3** of the revised manuscript.
>
>
> ### **Part (i): Explanation and exploration of the phenomenon you noted.**
>
> We augment Figure 2(b) by probing the relationship between a model’s parameter distance from its pre-trained state and evaluation accuracy under additional training budgets and optimizers. Concretely, under the same settings as Figure 2(b), we add runs at 0.1 and 0.2 epochs and extend training beyond 3 epochs, using Adam and MoFO. We report results on MMLU (as in the main text, measuring preservation of factual knowledge) and newly include HumanEval (measuring preservation of code-generation ability). Since the fine-tuning task is math, both can serve as forgetting mitigation metrics. The corresponding scatter plots are shown in [[Figure 8]](https://anonymous.4open.science/r/MoFO-TMLR-Rebuttal-C117/MMLU-dist-cor.png) (MMLU) and [[Figure 9]](https://anonymous.4open.science/r/MoFO-TMLR-Rebuttal-C117/HumanEval-dist-cor.png) (HumanEval). When examining the points **for each optimizer separately**, we make the following observations:
>
>
> - **Observation 1 (sufficient training).** Once training exceeds approximately 1 epoch, MMLU and HumanEval scores show a consistent strong negative correlation with the parameter distance to the pre-trained state.
> - **Observation 2 (early training).** For Adam or MoFO at less than 1 epoch, we may observe a mild correlation with the parameter distance to the pre-trained state. The correlation may be unstable and can be positive or negative.
>
>
> We speculate that this short-lived positive trend may be related to benchmark alignment. The MMLU benchmark (used to measure the preservation of factual knowledge) might share partial overlap with the patterns of our math fine-tuning task (measured by GSM8K benchmark); in the early training steps, the model might incidentally acquire features that also benefit MMLU, resulting in a temporary gain. By contrast, as for HumanEval, which measures code generation and differs from our math fine-tuning in both domain and output format, it may exhibit an unstable correlation whose sign can be either negative or positive. Another possible factor could stochasticity, since at less than 1 epoch the dataset has not yet been fully traversed.
>
> Overall, the negative correlation becomes clear and consistent after sufficient training; while the early training presents a mild, benchmark-dependent relationship.
>
> ### **Part (ii): Evidence of the negative correlation across different optimizers.**
>
> In addition, we emphasize an empirical point: the negative relationship between parameter distance and the preservation of pre-trained knowledge is evident **across optimizers**. In Figure 2(b), [[Figure 8]](https://anonymous.4open.science/r/MoFO-TMLR-Rebuttal-C117/MMLU-dist-cor.png), and [[Figure 9]](https://anonymous.4open.science/r/MoFO-TMLR-Rebuttal-C117/HumanEval-dist-cor.png), the parameter distances roughly follow the ordering **Lion > Adam > MoFO**, whereas the scores measuring preservation of pre-trained knowledge follow the inverse ordering **MoFO > Adam > Lion**. For a broader comparison, we evaluate five optimizers—MoFO, NAdam [1], Adam, RMSProp [2], and Lion. After two epochs of training, we report (i) their parameter distance to the pre-trained state and (ii) their forgetting-mitigation performance on MMLU and HumanEval, shown in [[Figure 10]](https://anonymous.4open.science/r/MoFO-TMLR-Rebuttal-C117/opt-compare.png). The results show a consistent negative correlation between distance and performance. Notably, MoFO remains closer to the pre-trained state and achieves higher scores compared with the other optimizers.
>
> **Therefore, this cross-optimizer rank-order correlation is sufficient to motivate our algorithm design**: favor optimizers that converge closer to the pre-trained state, so as to better preserve the pre-trained knowledge.
>
>
> ### **Reference:**
>
> [1] Dozat, Timothy. "Incorporating nesterov momentum into adam." (2016).
>
> [2] Tieleman, Tijmen. "Lecture 6.5‐rmsprop: Divide the gradient by a running average of its recent magnitude." (2012).

---

> ### Author Response · Authors · 2025-08-15
> **Response to Requested Change 3**
>
> ## Comment:
> *It is unclear how all model parameters can be partitioned into B blocks in Algorithm 1.*
>
> ## Response:
>
> We thank the reviewer for the helpful comment. We kindly note that we have explained how to partition the model parameters in the main text and in Appendix C.4 of the manuscript, and we would like to further explain it more clearly as follows.
>
> ### **The main text reads as follows, with a few additional sentences added for clarity (in bold):**
>
> For the parameter partitioning, we note that the network architecture is naturally composed of different modules (e.g., weight matrices, and bias terms). In the PyTorch implementation, the parameters of different modules (along with their gradients and momenta) are naturally stored in separate data tensors. Therefore, we adopt the default partitioning of model parameters as implemented in PyTorch. **For Transformers, this means that parameters such as query (Q), key (K), value (V) weights in the attention layers, as well as feed-forward network (FFN) weights, are grouped into distinct partitions following PyTorch’s default scheme.** This allows us to select and update the top-$\alpha$ parameters in each block without introducing much implementation overhead. See Appendix C.4 for further explanation of the partitioning.
>
>
> ### **The relevant content from Appendix C.4 is as follows:**
>
> **Partitioning.** We use the default partitioning scheme in PyTorch's Transformer implementation. Different types of parameters within the Transformer, such as query (Q), key (K), value (V) weights for attention heads, and feed-forward network (FFN) weights, are divided into separate partitions. Notably, in the default PyTorch implementation, within a layer, the query (Q) weights of all attention heads are grouped into a single partition. The same applies to the key (K) and value (V) weights. Our momentum-based filtering mechanism is applied to each partition individually.

---

> ### Author Response · Authors · 2025-08-15
> **Response to Requested Change 4**
>
> ## Comment:
> *The running efficiency of MoFo can be analyzed compared to baselines.*
>
> ## Response:
>
> We sincerely thank the reviewer for this comment. If we understand correctly, the comment concerns time efficiency. If that is the case, we would like to clarify that our original manuscript has included an efficiency analysis in Section 4.5 of the main text, with further details in Appendix F.2. Our discussions in the main text and the appendix are as follows:
>
> ### **Section 4.5**
>
> **Efficiency Analysis**. We provide an efficiency analysis on MoFO in Appendix F.2. The results show that MoFO requires only around 4%–5% additional training time compared with Default FT throughout the entire training process.
>
>
> ### **Appendix F.2**
>
> We claim that MoFO does not lead to significant reduced fine-tuning efficiency. We provide an efficiency analysis by comparing the total training time between MoFO and Default FT on three LLMs with different sizes. The parameter update fraction of MoFO is set as 10\%.
>
> The experimental results show that although MoFO requires computing a filter to select parameters with the largest momentum magnitudes, the additional computational overhead is minor. As shown in Table 1, the additional training time incurred by MoFO is approximately 4\%--5\%, which is relatively minor and manageable in practical fine-tuning scenarios.
>
> **Table 1:** Comparison of total training time for Default FT and MoFO on various LLaMA models and datasets.
>
>
> | Model         | Default FT | MoFO        | Additional Training Time |
> |---------------|------------|-------------|---------------------------|
> | LLaMA3.2-1B   | 49m22s     | 51m24s      | 4.1%                      |
> | LLaMA3.2-3B   | 1h30m18s   | 1h34m24s    | 4.5%                      |
> | LLaMA3-8B     | 5h2m0.69s  | 5h17m34.01s | 5.0%                      |

---

### Review · Reviewer_NmJ8 · 2025-08-03

**Summary Of Contributions:**

This paper studies the problem of preventing overfitting in fine-tuning, which is a common problem that practitioners encounter in the usage of large pretrained transformer models. Unlike prior work that often utilizes additional regularizations to prevent overfitting, this paper introduces a momentum filtering mechanism, called MoFO. The idea is to truncate the largest $\alpha$ percentage of the momentum updates to zero before applying the update. This idea is similar to the shrinkage estimator in statistics where large weights are shrinked relatively to reduce overfitting.

The proposed algorithm is validated through both theoretical analysis of its convergence and empirical validations on several LLM fine-tuning experiments. The experiments clearly demonstrate that the proposed algorithm leads to a convergent minimum that stays close to the pretrained model, while delivering strong empirical performance compared with two momentum methods, i.e., Adam and Lion.

Overall, I think the main idea of this paper meets the bar for acceptance to TMLR. I have several comments and suggestions below that I hope will further improve the quality of this work.

**Audience:**

Yes

**Claims And Evidence:**

Yes

**Requested Changes:**

List of comments to address:

- In the appendix, it would be useful to add a roadmap for quick reference to the proofs.

- Some of the figures are a bit difficult to understand, and I think they could use some extra polish in a revision.

    - For example, in Figure 1, it wasn't clear what the three axes are.

- The proofs in the Appendix could use some extra polish in the revision. Some of the notations and expositions can be improved.

    - In the proof of Proposition 1, page 21, what is the definition of $S'_k$ and $S''_k$? It would be good to clarify.

    - At the top of page 22, first equation, the norms are lacking a right-side bracket.

    - Appendix A.2, it would be helpful to recall the definitions of various notations from time to time so that would be easier to keep track.

    - Lemma 2: The notation $\alpha$% in C should be without the percentage symbol and instead define $\alpha$ as the percentage value directly.

- There are several earlier and recent references on distance-based regularization and regularization related to instruction fine-tuning. It would be good to add these to the related work discussion. For instance, it's plausible that the momentum filtering also leads to better stability of the loss surface (measured in terms of noise additions).

    - Li, D. and Zhang, H., 2021. Improved regularization and robustness for fine-tuning in neural networks. Advances in Neural Information Processing Systems, 34, pp.27249-27262.

    - Zhang, H.R., Li, D. and Ju, H., 2024. Noise Stability Optimization for Finding Flat Minima: A Hessian-based Regularization Approach. Transactions on Machine Learning Research.

- I wonder how difficult it would be to extend the convergence analysis to nonsmooth functions. It would be useful to add a discussion.

**Strengths And Weaknesses:**

Strengths:
- The paper includes a thorough theoretical analysis of the convergence behavior of MoFO.

- The proposed momentum filter is conceptually simple and may have implications for other types of optimization algorithms and settings.

- The proof of Theorem 1 is interesting and relies on a key norm condition on the momentum filter.

- Theorem 2 is interesting. I think it would be useful to run some synthetic experiments on top of Example 1 to validate the claimed relation between MoFO and Adam.

Weaknesses and Major Comments:
- The additional technical contribution on top of the prior paper by Shi et al. (2021) can be more clearly spelled out. Specifically, I think the technical exposition in Section 3.1 can probably be further fleshed out given the extra page space in TMLR format. For instance, from equation (1) to equation (2), this is interesting and I think you can connect this step with the proofs of the appendix more clearly.

- In Theorem 1, I wonder if it's possible to derive a bound in terms of the $\ell_2$ norm of the gradient? Asking because from the infinity norm to the $\ell_2$ norm one would incur an additional factor of $\sqrt d$, which would trickle into the convergence rate.

---

> ### Author Response · Authors · 2025-08-15
> **Response to Strength 4**
>
> ## Comment:
>  *Theorem 2 is interesting. I think it would be useful to run some synthetic experiments on top of Example 1 to validate the claimed relation between MoFO and Adam.*
>
> ## Response:
> Thanks sincerely for your comments. In **Appendix B.5** of the revised manuscript, we have added a synthetic experiment based on Example 1 to further examine the relationship between MoFO and Adam. For ease of reading, we include the corresponding section below:
>
> We conduct a synthetic experiment to provide a more concrete illustration of Example 1. Specifically, we set the parameter dimension to $d=10$, with  parameters $\theta = (\theta_1, \dots, \theta_{10})$. The pretraining loss is defined as the squared $L_2$ norm of the parameters: $\mathcal{L}\_{\text{pretrain}}(\theta) = \frac{1}{2} \Vert \theta \Vert_2^2$, with the pre-trained model given by $\theta_{\rm pretrain} = (0, 0, \dots, 0)$. Starting from the pre-trained model, we optimize the parameters with respect to the fine-tuning loss $\mathcal{L}(\theta) = \prod_{i=1}^d (a_i \theta_i - b_i)^2$, where $a_i, b_i > 0$ for any $1 \leq i \leq d$. The coefficients $a_i$ and $b_i$ are sampled from a standard normal distribution; to ensure positivity, we take their absolute values and add 0.3 and 0.1, respectively.
>
> In this experiment, we compare two optimizers: Adam and MoFO. For each, we perform a grid search for the optimal learning rate over the set $\{10^{-2}, 10^{-3}, 10^{-4}\}$. We consider the fine-tuning process to have converged to a minimum when the fine-tuning loss drops below $10^{-8}$ within 10000 iterations. At convergence, we record two metrics: the Euclidean distance from the fine-tuned model to the original pre-trained model, and the value of the pretraining loss. The entire experiment is repeated across three different random seeds for robustness.
>
>
> As presented in Table 1, the results indicate that the minimum found by MoFO is at roughly half the distance from the pre-trained model compared to the one found by Adam. Concurrently, MoFO achieves a lower pre-training loss. These findings provide strong evidence that MoFO can converge to minima closer to the pre-trained model in Example 1, thereby supporting Theorem 2.
>
>
>
> **Table 1:** The Euclidean distance from the fine-tuned model to the original pre-trained model, and the pretraining loss of parameters after optimizing the fine-tuning loss using Adam and MoFO. The results show that MoFO finds a fine-tuning minimum that is closer to the pre-trained model compared to the one found by Adam.
>
> |                          | Adam  | MoFO  |
> |--------------------------|-------|-------|
> | **Distance to pre-trained model** | 0.549 | 0.275 |
> | **Pre-training loss**    | 0.151 | 0.038 |

---

> ### Author Response · Authors · 2025-08-15
> **Response to Weakness 1**
>
> ## Comment:
> *The additional technical contribution on top of the prior paper by Shi et al. (2021) can be more clearly spelled out. Specifically, I think the technical exposition in Section 3.1 can probably be further fleshed out given the extra page space in TMLR format. For instance, from equation (1) to equation (2), this is interesting and I think you can connect this step with the proofs of the appendix more clearly.*
>
> ## Response:
> Thank you for your thoughtful comments. Following your suggestions, we have rewritten the proof sketch in the revised manuscript to make it more structured and detailed. In particular, we now specify the additional technical contribution we introduce beyond the prior work of Shi et al. (2021), with the key parts highlighted in blue.
>
> We note that **the step from Equation (1) to Equation (2) in the original manuscript** is actually **part of the proof strategy developed by Shi et al. (2021)**. In the revised manuscript, we have provided a more detailed explanation of this step in Part (i) of the proof sketch.

---

> ### Author Response · Authors · 2025-08-15
> **Response to Weakness 2**
>
> ## Comment:
> *In Theorem 1, I wonder if it's possible to derive a bound in terms of the l2 norm of the gradient? Asking because from the infinity norm to the l2 norm one would incur an additional factor of $\sqrt{d}$, which would trickle into the convergence rate.*
>
> ## Response:
>
> Thanks for the insightful comment.
>
> 1. Yes—we can derive an $L_2$ bound. In the revised manuscript, Theorem 1 has been extended to deliver $L_p$ bounds for all $p\in[1,\infty]$, which of course includes the $L_2$ case.
>
> 2. In fact, in our updated version, obtaining an $\ell_2$ bound in our proof does **not** introduce an additional $\sqrt{d}$ factor. In the derivation of Theorem 1 we first establish
>
> $\min_{1\leq t \le T} \Vert g_t\Vert_{1,\mathrm{top}\text{-}\alpha}=\min_{1\le t\le T} \Vert g_t \odot FLT_{\alpha} (g_t) \Vert_{1} = \mathcal{O} \left(\frac{\log T}{T}\right),$
>
> where $\texttt{FLT}_\alpha(\cdot)$ keeps the largest $\alpha$-fraction of entries (by magnitude) and $\odot$ is the Hadamard product.
> Then, by Lemma 2 (Appendix A.1), for any vector $z$,
>
> $\alpha\Vert z\Vert_{p} \le \Vert z \Vert_{1,\mathrm{top}\text{-}\alpha} \le \Vert z \odot FLT_\alpha(z) \Vert_{1}.$
>
> Applying this with $z=g_t$ yields
>
> $
> \Vert g_t \Vert_{p} \le \frac{1}{\alpha} \Vert g_t \Vert_{1,\mathrm{top}\text{-}\alpha},
> $
>
> and further
>
> $
> \min_{1\leq t \le T} \Vert g_t \Vert_{p} \le \frac{1}{\alpha} \min_{1\leq t \le T} \Vert g_t \Vert_{1,\mathrm{top}\text{-}\alpha}.
> $
>
> So the only additional factor is $1/\alpha$, not $\sqrt{d}$.

---

> ### Author Response · Authors · 2025-08-15
> **Response to Requested Change 1**
>
> ## Comment:
> *In the appendix, it would be useful to add a roadmap for quick reference to the proofs.*
>
> ## Response:
>
> We sincerely appreciate your thoughtful suggestion. In the revised manuscript, we have included a roadmap at the beginning of Appendix A. This roadmap offers a brief overview of the subsections (Appendix A.1-A.4), highlights their key points, and clarifies the roles of the propositions and lemmas, aiming to make it easier for readers to navigate the proofs.

---

> ### Author Response · Authors · 2025-08-15
> **Response to Requested Change 2**
>
> ## Comment:
> *Some of the figures are a bit difficult to understand, and I think they could use some extra polish in a revision.*
>
> - For example, in Figure 1, it wasn't clear what the three axes are.
>
>
>
> ## Response:
>
> Thanks for your comments. In the revised manuscript, we have added the explanations to the captions of the loss landscape visualization figures (**Figure 1, 4, 7, 12, 15**). We have extracted the key explanatory parts in Figure 1 for example:
>
> We visualize a 2D weight-space plane spanned by the vector from the pre-trained model to the Lion-tuned model (x-axis) and to the Adam-tuned model (y-axis). Axes are normalized so that one unit equals the length of the pre-trained $\to$ Adam vector. The color bar indicates the loss value—(a) fine-tuning loss and (b) pre-training loss.

---

> ### Author Response · Authors · 2025-08-15
> **Response to Requested Change 3**
>
> ## Comments:
>
> The proofs in the Appendix could use some extra polish in the revision. Some of the notations and expositions can be improved.
>
> - In the proof of Proposition 1, page 21, what is the definition of $S_k'$ and $S_k''$? It would be good to clarify.
> - At the top of page 22, first equation, the norms are lacking a right-side bracket.
> - Appendix A.2, it would be helpful to recall the definitions of various notations from time to time so that would be easier to keep track.
> - Lemma 2: The notation $\alpha$ \% in C should be without the percentage symbol and instead define $\alpha$ as the percentage value directly.
>
> ## Response:
>
> Thank you for your detailed comments. We have revised the manuscript accordingly and marked the changes in blue in the updated version. Our revisions are as follows:
>
> * On Page 26 of the updated manuscript (Page 22 of the original manuscript), we have clarified the definitions of $S_k'$ and $S_k''$ and highlighted them in blue.
>
> * We have added the missing right-side brackets in the first equation on page 22. In addition, we have checked and corrected several other typos in the theoretical proofs in Appendix A.
>
> * To help readers follow the proofs in Appendix A.2, we have listed the key variable definitions at the beginning of this section Moreover, we have repeatedly recalled these definitions at key points in the proofs to improve readability.
>
> * For better academic writing style and notational consistency, we have replaced all instances of $\alpha$% in the revised manuscript with $\alpha$.

---

> ### Author Response · Authors · 2025-08-15
> **Response to Requested Change 4**
>
> ## Comment:
>
> *There are several earlier and recent references on distance-based regularization and regularization related to instruction fine-tuning. It would be good to add these to the related work discussion. For instance, it's plausible that the momentum filtering also leads to better stability of the loss surface (measured in terms of noise additions).*
>
> - Li, D. and Zhang, H., 2021. Improved regularization and robustness for fine-tuning in neural networks. Advances in Neural Information Processing Systems, 34, pp.27249-27262.
>
> - Zhang, H.R., Li, D. and Ju, H., 2024. Noise Stability Optimization for Finding Flat Minima: A Hessian-based Regularization Approach. Transactions on Machine Learning Research.
>
>
> ## Response:
>
> Thanks for your comments. We have added the following discussion into the related work section of the updated manuscipt.
>
> To improve generalization and robustness to noise after instruction-tuning, several studies introduce explicit regularization [1,2]. In particular, [2] proposes a Hessian-based penalty that encourages convergence to flatter minima. It is an interesting direction for future research to evaluate whether MoFO's momentum filtering mechanism implicitly favors more stable minima, thereby further enhancing generalization and noise robustness.
>
> **Reference:**
>
> [1] Li, D. and Zhang, H., 2021. Improved regularization and robustness for fine-tuning in neural networks. Advances in Neural Information Processing Systems, 34, pp.27249-27262.
>
> [2] Zhang, H.R., Li, D. and Ju, H., 2024. Noise Stability Optimization for Finding Flat Minima: A Hessian-based Regularization Approach. Transactions on Machine Learning Research.

---

> ### Author Response · Authors · 2025-08-15
> **Response to Requested Change 5**
>
> ## Comment:
>
> *I wonder how difficult it would be to extend the convergence analysis to nonsmooth functions. It would be useful to add a discussion.*
>
> ## Response:
>
> Thank you for the thoughtful suggestion. We agree that extending our convergence analysis to non-smooth objectives is both valuable and practically relevant. Below we clarify why this extension is non-trivial under our current proof technique and outline several potential directions. The following discussion has been added as a new subsection (**Appendix A.4**) in the revised manuscript.
>
> **Why the extension is challenging.**
>
> Our Theorem 1 relies on the standard **$L$-smoothness** assumption—i.e., the gradient of $\mathcal{L}$ is $L$-Lipschitz—to invoke the descent lemma and to derive an upper bound on $\min_{0 \le t \le T-1}\Vert \nabla \mathcal{L}(\theta_t)\Vert$. When $\mathcal{L}$ is non-smooth,
> - (i) $\nabla \mathcal{L}(\theta)$ may not exist at singular points, which requires working with generalized gradients (e.g., subgradients) rather than classical gradients in our derivation.
> - (ii) Our upper bound of $\min_{0 \le t \le T-1}\Vert \nabla \mathcal{L}(\theta_t)\Vert$ in the proof of Theorem 1 (Appendix A.2) essentially scales with the smoothness constant $L$. When $L$ blows up, these inequalities become non-informative—the bound itself can blow up.
>
> **Possible extensions.**
> 1. **Subgradient analysis.** Work with subgradients rather than classical gradients and assess convergence via a subgradient-based stationarity criterion.
> 2. **Smoothing.** A natural analysis path could be to introduce a family of smoothed surrogates $\{\mathcal{L}_\mu\}$, analyze the MoFO algorithm under these surrogates to obtain $\mu$-dependent bounds, and then let $\mu \downarrow 0$ to derive the convergence result on the original loss $\cal{L}$.
> 3. **Algorithmic modifications.** Incorporate techniques such as gradient clipping or stochastic subgradient methods [1], and then demonstrate the convergence of the modified algorithm.
>
> **Reference:**
>
> [1] Xiao, Nachuan, et al. "Adam-family methods for nonsmooth optimization with convergence guarantees." _Journal of Machine Learning Research_ 25.48 (2024): 1-53.

---

> > ### Comment · Reviewer_NmJ8 · 2025-08-24
> >
> > Thanks for your detailed response. The revised exposition in Section 3 is now much clearer. The improved convergence rate in Theorem 1 is also very interesting.

---

### Review · Reviewer_ChzC · 2025-08-04

**Summary Of Contributions:**

This paper describes a new variant of the Adam optimizer called Momentum-Filtered Optimization (MoFO) that is designed to allow language models to finetune to new tasks while retaining more of their original capabilities, that is, while avoiding forgetting. The optimizer works by selecting a small percentage (typically between 5 and 15%) of parameters in each parameter block to be updated. Parameters are selected based on which have the largest Adam momentum values. Blocks can be practically defined to correspond to natural parameter groupings in popular implementations. The authors provide a proof that this optimization converges, adapting a recent proof for the Adam optimizer. They also provide experiments that show the method outperforms reasonable baselines drawn from optimization (half fine-tuning), architecture (LORA), regularization (L1 or L2 toward the original model), a host of continual learning techniques (GEM, EWC, Replay), and a number of block coordinate descent variants (Random, Grad). The experiments are on fairly small models (Llama-2-7B, Gemma-2B, TinyLlama-1.1B), but cover a nice collection of tasks. Results are convincing overall.

**Audience:**

Yes

**Broader Impact Concerns:**

None. An optimization paper such as this has no ethical concerns beyond those inherent to machine learning.

**Claims And Evidence:**

Yes

**Requested Changes:**

None. The paper seems fine to me.

**Strengths And Weaknesses:**

Strengths:

- The proposed method is described well. It makes sense to have tried it, and it sounds easy to re-implement.
- It is nice that the authors targeted something uniquely suited to Adam, by far the most popular LLM optimizer.
- The authors provide a proof of convergence.
- Experiments are thorough and convincing. The method has a wide range of competitors, and many of those are represented in at least one experiment.

Weaknesses:

- The motivation section is nice, but it spends a lot of time talking about how crucial it is for a finetuned model to find a local optimum close to the original model parameters. Once we reach the experiments section, and see that both L1 and L2 regularization (which explicitly optimize for proximity to the original model) are pareto-dominated by the proposed technique, it would be nice to spend some amount of time speculating as to why that should be the case. Did FoMO actually do a better job of optimizing the L2-regularized objective? Or is it possible to be too close to the original parameters? Honestly, this isn’t much of a weakness.

---

> ### Author Response · Authors · 2025-08-16
> **Response to the Weakness**
>
> Thank you for the thoughtful comments. Here we clarify that our motivation is twofold—to preserve pretrained knowledge and to achieve strong fine-tuning performance—rather than to focus solely on preserving knowledge by minimizing the distance to the pretrained weights. **We have made some modifications in the Introduction to emphasize this clarification.**
>
> Our view is that an effective forgetting-mitigation fine-tuning method should meet two goals:
> - **(G1)** mitigate forgetting of pretrained knowledge, which we think is strongly correlated with staying close to the original parameters;
> - **(G2)** achieve strong fine-tuning performance.
>
> **MoFO is designed to advance both (G1) and (G2):** it preserves proximity to the pretrained model **(G1)** by constraining updates at each step to a fraction of parameters via momentum filtering, and it achieves fine-tuning performance **(G2)** by optimizing the original fine-tuning loss without penalty terms.
>
> By contrast, penalty methods (e.g., L1/L2) promote (G1) by explicitly encouraging the model converges closer, but they also alter the training objective and may bias the optimizer, potentially harming (G2). **In other words, although L1/L2 regularization may keep the model close to the pre-trained state, it is less likely to actually reach a local minimum of the original fine-tuning loss.** We argue that this **might be one of the reasons why the MoFO line Pareto-dominates the L1/L2 lines** in Figure 5.
>
> To further explore this point, we add **Appendix F.5** in the revised manuscript. Using the Llama-2-7B/MetaMathQA checkpoints underlying Figure 5, we compute the (unregularized) fine-tuning loss and the $\ell_2$ norm of the gradient of the unregularized fine-tuning loss.
>
> As shown in [[the figure]](https://anonymous.4open.science/r/MoFO-TMLR-Rebuttal-C117/mofo-l1l2-loss-gradnorm.png), for comparable CR scores, MoFO checkpoints generally exhibit smaller fine-tuning loss and smaller gradient norms than those trained with $L_1$ or $L_2$, indicating better convergence under the original task loss. A plausible explanation is that while penalty methods keep parameters close, they also modify the objective/gradient field and can bias optimization, thereby hindering convergence to a local minimum of the original fine-tuning loss. MoFO instead constrains how updates are applied (via momentum filtering) while continuing to optimize the original loss, letting the optimizer follow task-relevant directions more effectively. These observations are consistent with the Pareto-front gap we observe in Figure 5 (see Appendix F.5 for the plot and details).

---

### Comment · Editors_In_Chief · 2026-01-30

On January 29, at the request of the authors, the EiCs replaced the camera ready version of this paper with an updated version. This includes an acknowledgment which was originally omitted.

---

### Decision · Action_Editor_bFLR · 2025-09-08

**Recommendation:** Accept as is

**Audience:**

Yes

**Audience Explanation:**

This addresses a critical practical problem in LLM deployment - catastrophic forgetting during fine-tuning - that affects researchers, practitioners, and companies working with large language models. The proposed solution is implementable without access to pre-training data, making it broadly applicable to the TMLR audience working on optimization, continual learning, and LLM applications.

**Claims And Evidence:**

Yes

**Claims Explanation:**

The claims are well-supported by rigorous theoretical analysis with convergence proofs, comprehensive experiments across multiple models and tasks, and thorough comparisons with relevant baselines. All three reviewers confirmed the evidence is convincing and accurate.